# Off-Policy Evaluation Under Nonignorable Missing Data

**Han Wang** [1]   **Yang Xu** [1]   **Wenbin Lu** [1]   **Rui Song** [1]

## Abstract

Off-Policy Evaluation (OPE) aims to estimate the value of a target policy using offline data collected from potentially different policies. In real-world applications, however, logged data often suffers from missingness. While OPE has been extensively studied in the literature, a theoretical understanding of how missing data affects OPE results remains unclear. In this paper, we investigate OPE in the presence of monotone missingness and theoretically demonstrate that the value estimates remain unbiased under ignorable missingness but can be biased under nonignorable (informative) missingness. To retain the consistency of value estimation, we propose an inverse probability weighting value estimator and conduct statistical inference to quantify the uncertainty of the estimates. Through a series of numerical experiments, we empirically demonstrate that our proposed estimator yields a more reliable value inference under missing data.

## 1. Introduction

Reinforcement learning (RL) has demonstrated many successes in various domains such as game playing (Mnih et al., 2013; Silver et al., 2016), robotic control (Kober et al., 2013), bidding (Jin et al., 2018; Xu et al., 2023), and ridesharing (Xu et al., 2018). These successes often rely on simulators to generate large amounts of interaction data for training. However, in real-world applications, direct access to the environment is usually limited, making it difficult to deploy and evaluate new policies in practice, especially in safety-critical fields like healthcare and autonomous driving.

Off-Policy Evaluation (OPE) is a critical step in offline RL to estimate the value of a target policy using offline data that may have been collected under different policies (Prasad et al., 2017; Raghu et al., 2017b; Wang et al., 2018).

[1]Department of Statistics, North Carolina State University, USA. Correspondence to: Rui Song <songray@gmail.com>.

*Proceedings of the 42$^{nd}$ International Conference on Machine Learning*, Vancouver, Canada. PMLR 267, 2025. Copyright 2025 by the author(s).

In recent years, there has been growing interest in high-confidence off-policy evaluation (HCOPE), which not only estimates the policy's value but also provides statistical inference to quantify the confidence in these estimates (Thomas et al., 2015; Luckett et al., 2019; Shi et al., 2021b). However, one overlooked aspect in this literature is that offline data is often incomplete due to different types of missingness. For example, the reward and next-state may be absent following some actions, resulting in incomplete transition tuples.

Missing data mechanisms are generally categorized as ignorable or nonignorable (informative) missingness. Ignorable missingness assumes that the pattern of missing data can be fully explained by observed variables. In contrast, nonignorable missingness occurs when the missing data depends on the outcome or reward of interest, making it a more complex and challenging issue to address. Both types of missingness frequently arise in real-world RL applications, including online advertising (Tewari & Murphy, 2017), healthcare (Smith et al., 2023; Yu et al.), robotics (Wang et al., 2019), and more. Failing to account for the underlying missingness pattern, particularly nonignorable missingness (as discussed in later sections), can lead to biased evaluations of policy performance and, consequently, flawed decision-making.

For example, in movie recommendation systems like Movielens (Harper & Konstan, 2015), platforms aim to determine the best strategy for recommending personalized movie genres using historical user rating data. However, it is well known that users are more likely to rate movies they prefer, leading to nonignorable missingness. Imagine we want to assess user preferences for two genres, comedy and horror. If users only rate movies they enjoy and avoid providing ratings for those they dislike, the dataset might consist solely of high ratings (e.g., 5-star ratings) while lower ratings remain completely unobserved. Without accounting for this nonignorable missingness, the resulting recommendation strategy might mistakenly treat comedy and horror as equally preferred genres and recommend them at random to all users, which is totally ineffective in evaluating user preferences.

A similar challenge arises in survival analysis. Healthcare data often suffers from missingness caused by factors like early discharge (nonignorable when health status is the outcome) or missed follow-ups (ignorable, as it occurs randomly and is unrelated to health status). For example, in

the sepsis dataset (Komorowski et al., 2018), nonignorable missingness is evident in the shorter trajectories of deceased patients, who typically exhibit higher SOFA scores (Sepsis-related Organ Failure Assessment) (Vincent et al., 1996), indicating more severe conditions. As shown in Figure 1, the average SOFA score for deceased patients is higher than that of surviving patients. Ignoring this mortality-driven missingness pattern can result in underestimating the effect of treatments on SOFA scores, potentially leading to misguided decisions about treatment effectiveness.

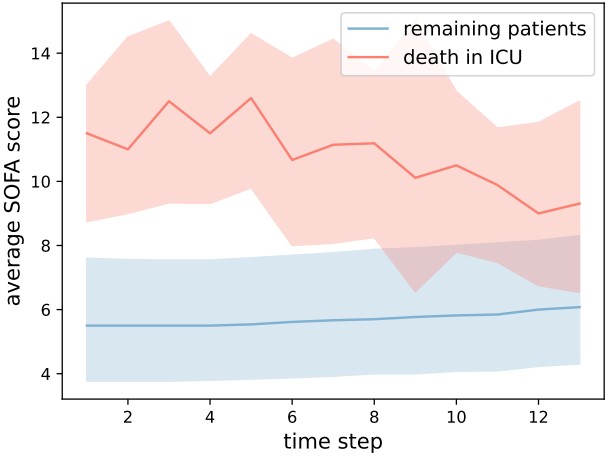

*Figure 1.* The average SOFA scores for patients remaining in the dataset (blue) and patients who died during ICU stay (red). The shadow represents the 25% to 75% quantile.

In the RL literature, missingness is sometimes addressed by manually defining it as a special event within the reward framework. For example, in the Gridworld environment, "missingness" might be represented by the agent hitting a wall. In such cases, RL algorithms typically assign a large negative value (e.g., $-10$ or $-100$) to discourage the agent from taking actions that lead to these boundary states. However, the choice of penalty severity can significantly affect the evaluation of the value function. Other naive approaches might treat "missingness" as an additional constraint, transforming the problem into a constrained RL task (Chu et al., 2023). This leads to challenges such as balancing the original reward with the penalty for missingness. Moreover, these simplified solutions do not truly treat "missingness" as a distinct issue, which risks diluting the focus of the problem:

"*What is the expected value function under the target policy, assuming the trajectories were not subject to missingness?*"

Building on the extensive literature on missingness in survival analysis (Goldberg & Kosorok, 2012; Dong et al., 2020; Zhao & Ma, 2022; Miao et al., 2024), we aim to propose a systematic framework for off-policy evaluation

in the presence of missing data. Unlike the simplifications described above, we treat missingness as a distinct node in the causal diagram, which can be causally influenced by both historical information and the rewards. This allows us to explicitly model its impact on policy evaluation. Specifically, we theoretically demonstrate that the original value estimator remains valid under ignorable missingness but becomes biased when the missing mechanism is nonignorable. To mitigate the bias, we propose a novel Inverse Probability Weighting (IPW) value estimator that is shown to be consistent under nonignorable missingness. Furthermore, we conducted statistical inference on the proposed value estimator and provide the associated confidence interval to quantify the uncertainty in value estimation. The effectiveness of the proposed estimator is empirically demonstrated through a simulation study and a real application to MIMIC-III data.

We highlight our contributions as follows:

- Under MAR, we are the first to provide theoretical justification for traditional OPE estimators that do not explicitly account for missingness. Specifically, we identify the key conditions required to ensure the consistency and validity of these estimators.

- Under MNAR, we show that traditional OPE methods lead to biased estimates. To address this, we propose a novel value estimator to ensure the consistency, with a flexible framework that supports both parametric and semi-parametric estimation.

- We are also the first to thoroughly study the asymptotic properties of OPE under MNAR, providing uncertainty quantification to the value estimate. The effectiveness of our approach has been validated through extensive simulations and real-world data applications.

## 2. Related Work

**Off-Policy Evaluation.** OPE has been extensively studied in the literature. Existing approaches can be categorized into three classes. The first category is the Direct Method (DM), where the value is estimated by learning the transition model or fitting the Q-function via model-free function approximation (Bradtke & Barto, 1996; Lagoudakis & Parr, 2003; Le et al., 2019). The second category is the Importance Sampling-based (IS) method (Precup, 2000; Liu et al., 2018; Nachum et al., 2019), which re-weights the observed rewards to correct the mismatch of data distributions between the target policy and the behavior policy. The third category is the Doubly Robust (DR) method (Jiang & Li, 2016; Tang et al., 2019; Kallus & Uehara, 2022), which combines these two methods for more robust and efficient value evaluation. For a comparison of the empirical performance of various OPE approaches, we refer the readers to

Voloshin et al. (2019). However, the performance of those OPE methods under missing data is seldom explored.

**High-Confidence OPE.** In addition to obtaining point estimates of value, many applications would benefit from quantifying the uncertainty in Off-Policy Evaluation (OPE) estimates. This type of OPE method is referred to as High-Confidence Off-Policy Evaluation (HCOPE) (Thomas et al., 2015). Dai et al. (2020) estimated the value confidence interval (CI) using the empirical likelihood approach under the assumption of i.i.d. transitions, which is often violated in practice (Shi et al., 2021a). Recently, Luckett et al. (2019) and Shi et al. (2021b) advanced the asymptotic theory under Markov Decision Processes (MDPs), with the former focusing on policy learning and the latter on value inference. Despite the extensive literature on OPE, HCOPE remains theoretically more challenging and, as a result, has been less explored in the literature.

**Missing Data.** For ignorable missingness, the IPW technique has been studied in finite-horizon decision-making (Goldberg & Kosorok, 2012; Dong et al., 2020). However, these methods depend on backward recursion, making them susceptible to model misspecification as the horizon length increases. To the best of our knowledge, there is a lack of theoretical support that guarantees the performance of classical OPE approaches in infinite horizon with ignorable missingness, such as under the widely studied MDP framework (Puterman, 1994).

For nonignorable missingness, the problem becomes more challenging to solve. In the absence of auxiliary variables, it has been proved that identifying the observed likelihood is impossible, even within a parametric framework (Wang et al., 2014). Existing research has made some progress in identifying and estimating dropout patterns using auxiliary variables, such as *shadow* variables (Zhao & Ma, 2022; Miao et al., 2024) or *instrumental* variables (Chen et al., 2009; Wang et al., 2014; Shao & Wang, 2016; Sun et al., 2018; Tchetgen Tchetgen & Wirth, 2017). However, these approaches are limited to single-stage decision-making and do not provide the necessary insights for value estimation in sequential decision-making, which is crucial in many real-world applications, as outlined in the introduction.

Among these works, Wang et al. (2014); Tchetgen Tchetgen & Wirth (2017) focus on parametric estimation, while Kim & Yu (2011); Shao & Wang (2016); Sun et al. (2018) emphasize semi-parametric estimation, which offers greater flexibility in specific contexts. Notably, Miao et al. (2024) advances the field by providing a non-parametric identification framework that integrates and generalizes the findings of Wang et al. (2014); Shao & Wang (2016); Miao et al. (2016). This comprehensive framework serves as an excellent foundation for addressing the general problem of MNAR in more complex multi-stage settings such as MDPs.

## 3. Preliminaries

Assume the data follows an MDP defined by a tuple $(\mathcal{S}, \mathcal{A}, p, r, \gamma)$, where $\mathcal{S}$ is the state space, $\mathcal{A}$ is the action space, $p : \mathcal{S} \times \mathcal{A} \to \mathcal{S}$ is the Markov transition kernel that characterizes the environment dynamics, $r : \mathcal{S} \times \mathcal{A} \to \mathbb{R}$ is the reward function where larger positive rewards are preferable, and $\gamma \in (0, 1)$ is a discount factor that trades off long-term rewards for immediate rewards. In this work, we assume the state space $\mathcal{S}$ is continuous with $d$-dimensional state variables, and the action space $\mathcal{A} = \{1, \ldots, m\}$ is discrete with $m$ distinct actions. Consider a trajectory $\{(S_t, A_t, R_{t+1})\}_{t \geq 0}$ generated from the MDP model, where $(S_t, A_t, R_{t+1})$ denotes the triplet of state, action, and immediate reward. Here, we denote the reward as $R_{t+1}$ instead of $R_t$ to emphasize that the reward $R_{t+1}$ and next state $S_{t+1}$ are jointly determined. The following assumptions are commonly imposed in infinite-horizon RL problems.

**Assumption 3.1** (Time-homogeneous Markov Assumption)**.** The transition probability satisfies $P(S_{t+1}|S_t = s, A_t = a, \{S_j, A_j, R_{j+1}\}_{0 \leq j < t}) = p(S_{t+1}|S_t = s, A_t = a) = p(S_1|S_0 = s, A_0 = a)$, where $p$ is the transition function.

**Assumption 3.2** (Conditional Mean Independence Assumption)**.** $\mathbb{E}(R_{t+1}|S_t = s, A_t = a, \{S_j, A_j, R_{j+1}\}_{0 \leq j < t}) = \mathbb{E}(R_{t+1}|S_t = s, A_t = a) := r(s, a)$, where $r$ is the reward function.

Assumptions 3.1-3.2 guarantee the existence of an optimal stationary policy (Puterman, 1994). A stationary policy $\pi$ is a mapping from the state space $\mathcal{S}$ to a probability mass function over the action space $\mathcal{A}$. For discounted infinite-horizon MDPs, the state value function of policy $\pi$ is defined as $V^\pi(s) = \mathbb{E}_\pi\{\sum_{t=0}^\infty \gamma^t R_{t+1}|S_0 = s\}$, where $\mathbb{E}_\pi$ denotes the expectation with respect to the trajectory distribution following policy $\pi$. The policy value is defined as $V_{\mathbb{G}}^\pi = \mathbb{E}_{s \sim \mathbb{G}} V^\pi(s) = \int_{s \in \mathcal{S}} V^\pi(s)\mathbb{G}(ds)$, where $\mathbb{G}$ is some reference state distribution over which the policy is evaluated, a common choice is the initial state distribution. This integrated value $V_{\mathbb{G}}^\pi$ quantifies the overall performance of a policy and hence is a key concept in policy evaluation. Similarly, the state-action value function (better known as the Q-function) is defined as $Q^\pi(s, a) = \mathbb{E}_\pi\{\sum_{t=0}^\infty \gamma^t R_{t+1}|S_0 = s, A_0 = a\}$. Under Assumption 3.1 and 3.2, $Q^\pi$ satisfies the Bellman equation

$$\mathbb{E}\Big\{R_{t+1} + \gamma V^\pi(S_{t+1}) - Q^\pi(S_t, A_t) \mid S_t, A_t\Big\} = 0, \quad (1)$$

where $V^\pi(S_{t+1}) = \sum_{a' \in \mathcal{A}} Q^\pi(S_{t+1}, a')\pi(a'|S_{t+1})$. This equation plays a critical role in estimating the Q-function in many RL algorithms.

## 4. Off-Policy Evaluation with Missing Data

In this section, we systematically study OPE under missing data. Specifically, we focus on monotone missingness,

which is a common pattern in longitudinal data analysis (Birmingham et al., 2003; Zhou & Kim, 2012; Linero & Daniels, 2015). Monotone missingness assumes that if an observation is missing at a given time point in a trajectory, it will remain missing at all subsequent time points. This pattern is often seen in multi-stage missing data settings, particularly when subjects drop out before completing the entire follow-up period. Before proceeding, we introduce some necessary definitions. Let $\mathcal{D} = \{\tau_i\}_{1 \leq i \leq n}$ denote the observed data consisting of $n$ independent and identically distributed trajectories. For complete data, each trajectory can be expressed as $\tau_i = \{(S_{i,t}, A_{i,t}, R_{i,t+1}, S_{i,t+1})\}_{0 \leq t < T}$, where $T$ is the termination time. Throughout this paper, we assume that the immediate rewards are uniformly bounded.

For incomplete data, define $\boldsymbol{\eta} = (\eta_0, \eta_1, \eta_2, \ldots, \eta_T)^{\top}$ as a vector of binary response indicators. $\eta_{i,t}$ is a sample of $\eta_t$ that represents the response status of subject $i$ at time $t$: $\eta_{i,t} = 1$ if subject $i$ is still in the study at time $t$ and we observe $(R_{i,t}, S_{i,t}, A_{i,t})$, otherwise $\eta_{i,t} = 0$. Assume the baseline covariates and initial treatment assignment are always observable, i.e., $\eta_{i,0} = 1$. We consider a general setting where the reward $R_{t+1}$ depends on $(S_t, A_t, S_{t+1})$. If $S_{t+1}$ is unobserved, then $R_{t+1}$ is missing as well. Therefore, an observed trajectory can be represented as $\tau_i = \{(\eta_{i,t}S_{i,t}, \eta_{i,t}A_{i,t}, \eta_{i,t+1}R_{i,t+1}, \eta_{i,t+1}S_{i,t+1})\}_{0 \leq t < T}$. Under monotone missingness, $\boldsymbol{\eta}_i$ is a decreasing sequence: if $\eta_{i,t} = 0$, then $\eta_{i,s} = 0$ for all $s > t$. To describe the lengths of observed trajectories, we also define the dropout time $C$: $C = t$ if the subject dropped out right after action $A_t$, which corresponds to $(\eta_t, \eta_{t+1}) = (1, 0)$. If the trajectory is fully observed, $C$ is set to $T$.

Given the offline data $\mathcal{D}$, our goal is to estimate and conduct statistical inference on the value function under target policy $\pi$. Although this work focuses on monotone missingness where dropout occurs after the action is observed, the proposed framework and theoretical results are applicable to more general dropout patterns. A discussion on these broader scenarios, including dropout before action occurs and intermittent missingness, is provided in Appendix F.1.

## 4.1. Missing Data Mechanisms

There are two major types of missing data mechanisms: ignorable and nonignorable missingness. Ignorable missingness refers to the case where the missingness can be fully explained by the observed information, which is also referred to as Missing-At-Random (MAR). An example of MAR would be dropout in a clinical trial due to recorded side effects and lack of efficacy, or other known baseline characteristics. The term "randomness" in MAR implies that once one has conditioned on all the available data, any remaining missingness is completely random (Graham et al., 2009). On the other hand, if the missingness depends on

unobserved components, the missing data mechanism is referred to as nonignorable, or Missing-Not-At-Random (MNAR). Dropout in a clinical trial due to the unobserved current health status or other latent variables is an example of an MNAR. Formal definitions of the two missing mechanisms are given as follows.

**Definition 4.1** (Ignorable Missingness, MAR)**.** The missingness can be fully accounted for by the observed information, i.e., $\eta_{t+1} \perp\!\!\!\perp (R_{t+1}, S_{t+1}) \mid (S_t, A_t, \{(S_j, A_j, R_{j+1})\}_{0 \leq j < t}, \eta_t)$, for $t = 0, \ldots, T - 1$.

**Definition 4.2** (Nonignorable Missingness, MNAR)**.** The missingness depends on the next state regardless of whether it is observed or not, i.e., $\eta_{t+1} \not\perp\!\!\!\perp (R_{t+1}, S_{t+1}) \mid (S_t, A_t, \{(S_j, A_j, R_{j+1})\}_{0 \leq j < t}, \eta_t)$, for $t = 0, \ldots, T - 1$.

*Remark* 4.3. In the special case where $S_{t+1}$ and $R_{t+1}$ are fully determined by $S_t$ and $A_t$, nonignorable missingness reduces to ignorable missingness. However, in practice, we cannot evaluate whether the missing data mechanism is ignorable or nonignorable solely based on the observed data. Instead, this distinction need to be justified using contextual information and subject-matter knowledge.

We define the dropout propensity $\lambda(\cdot)$, i.e. the probability of the subject dropping out right after time $t$, as a function of all historical data observed until time $t$ (Diggle & Kenward, 1994). That is, $P(C = t \mid \{(S_j, A_j, R_{j+1}, S_{j+1})\}_{0 \leq j < t+1}, C \geq t) = P(\eta_{t+1} = 0 \mid \{(S_j, A_j, R_{j+1}, S_{j+1})\}_{0 \leq j < t+1}, \eta_t = 1)$. Similar to the Markov assumptions in MDPs imposed on $R_t$ and $S_{t+1}$ (see Assumptions 3.1-3.2), we assume that $\eta_{t+1}$ exhibits a similar dependency on existing data, as stated below.

**Assumption 4.4.** The dropout propensity satisfies $P(\eta_{t+1} = 0 \mid \{S_j, A_j, R_{j+1}\}_{0 \leq j \leq t}, S_{t+1}, \eta_t = 1) = P(\eta_{t+1} = 0 \mid S_t, A_t, R_{t+1}, S_{t+1}, \eta_t = 1)$, denoted by $\lambda(S_t, A_t, R_{t+1}, S_{t+1})$.

This assumption states that whether a subject will drop out or not right after receiving $A_t$ depends on the history only through the current data tuple $(S_t, A_t, R_{t+1}, S_{t+1})$. In practical applications, if the missing probability (or $R_{t+1}$ and $S_{t+1}$ as specified in Assumption 3.1-3.2) is suspected to depend on multiple past steps, this assumption can still be satisfied by directly incorporating such information into the current state variable (Shi et al., 2020).

## 4.2. Value Estimation and Inference Under Missingness

Given incomplete data, the response indicator $\eta_t$ must be accounted for. One common approach is to estimate using Equation (1) with only the observed data where $\eta_{i,t} = 1$. However, we will demonstrate later in this section that this straightforward solution does not always yield an unbiased estimate under general missing data mechanisms, such as MNAR. In this section, we investigate the scenarios intro-

duced in Section 4.1, propose consistent estimators for these cases, and conduct a comprehensive statistical analysis.

Suppose that the base algorithm approximates the Q-function $Q^\pi$ with linear sieves (Shi et al., 2021b), $Q^\pi(s, a) \approx \Phi_L^\top(s)\boldsymbol{\beta}_{\pi,a}$, where $\Phi_L(\cdot) = \{\boldsymbol{\phi}_{L,1}(\cdot), \cdots, \boldsymbol{\phi}_{L,L}(\cdot)\}^\top$ denotes a vector of $L$ sieve basis functions. One can use splines (De Boor, 1976) or wavelet basis (Huang et al., 1998), and the number of basis functions $L$ is allowed to grow with the sample size to reduce the approximation error[1]. For ease of notation, we first define $\boldsymbol{\beta}_\pi = (\boldsymbol{\beta}_{\pi,1}, \ldots, \boldsymbol{\beta}_{\pi,m})^\top \in \mathbb{R}^{mL}$, $\boldsymbol{\xi}(s, a) = \{\Phi_L^\top(s)\mathbb{1}(a = 1), \cdots, \Phi_L^\top(s)\mathbb{1}(a = m)\}^\top$, and $\boldsymbol{U}_\pi(s) = \{\Phi_L^\top(s)\pi(1|s), \cdots, \Phi_L^\top(s)\pi(m|s)\}^\top$. Then, after some calculations, the Q-function can be expressed as $Q^\pi(s, a) = \boldsymbol{\xi}(s, a)^\top \boldsymbol{\beta}_\pi$, and the value function is $V^\pi(s) = \mathbb{E}_{a \sim \pi(\cdot|s)}[Q^\pi(s, a)] = \boldsymbol{U}_\pi(s)^\top \boldsymbol{\beta}_\pi$.

When there is no missingness, one can follow the standard Bellman equation in (1), substitute the definitions above, and derive the following estimating equation: $\mathbb{E}\{\boldsymbol{M}_t(\boldsymbol{\beta}_\pi)\} = \mathbf{0}$, where $\boldsymbol{M}_t(\boldsymbol{\beta}_\pi) = \boldsymbol{\xi}_t\{R_{t+1} + \gamma V^\pi(S_{t+1}) - Q^\pi(S_t, A_t)\} = \boldsymbol{\xi}_t\{R_{t+1} - (\boldsymbol{\xi}_t - \gamma \boldsymbol{U}_{\pi,t+1})^\top \boldsymbol{\beta}_\pi\}$. The problem of value function estimation thus reduces to a linear regression. The true parameter $\boldsymbol{\beta}_\pi^*$ can be estimated by solving the estimating equations $\mathbb{E}_{nT}\{\boldsymbol{M}_t(\boldsymbol{\beta}_\pi)\} = \mathbf{0}$, where $\mathbb{E}_{nT}(\cdot)$ denotes the empirical average over $nT$ transition pairs of $(S_{i,t}, A_{i,t}, R_{i,t+1}, S_{i,t+1})$.

However, when dealing with MNAR, adjustments are necessary to obtain unbiased estimates. We define the propensity score as the probability of observing the data at time $t$, denoted by $1 - \lambda_t(\cdot)$. To ensure consistency in value estimates, an Inverse Probability Weighting (IPW) term, $\frac{\eta_{t+1}}{1 - \lambda_{t+1}(\cdot)}$, is incorporated into the estimating equation. Specifically, we consider the following estimating equation:

$$\mathbb{E}_{nT}\left\{\frac{\eta_{t+1}}{1 - \lambda_{t+1}(\boldsymbol{\psi})}\boldsymbol{M}_t(\boldsymbol{\beta}_\pi)\right\} = \mathbf{0}, \quad (2)$$

where $\lambda_{t+1}(\boldsymbol{\psi}) := \lambda(S_t, A_t, R_{t+1}, S_{t+1}; \boldsymbol{\psi})$ is the dropout propensity model parameterized by $\boldsymbol{\psi} \in \mathbb{R}^q$. As we focus here on presenting the general framework, the estimation of $\boldsymbol{\psi}$, denoted by $\widehat{\boldsymbol{\psi}}_{nT}$, will be detailed in Section 4.3. Based on Equation (2), an IPW estimator for $\boldsymbol{\beta}_\pi$ is obtained as

$$\widehat{\boldsymbol{\beta}}_{\pi,\text{IPW}} =$$
$$\underbrace{\left\{\frac{1}{nT}\sum_{i=1}^n\sum_{t=0}^{T-1}\frac{\eta_{i,t+1}}{1 - \lambda_{i,t+1}(\widehat{\boldsymbol{\psi}}_{nT})}\boldsymbol{\xi}_{i,t}(\boldsymbol{\xi}_{i,t} - \gamma\boldsymbol{U}_{\pi,i,t+1})^\top\right\}^{-1}}_{\widehat{\boldsymbol{\Sigma}}_{\pi,\text{IPW}}}$$
$$\left(\frac{1}{nT}\sum_{i=1}^n\sum_{t=0}^{T-1}\frac{\eta_{i,t+1}}{1 - \lambda_{i,t+1}(\widehat{\boldsymbol{\psi}}_{nT})}\boldsymbol{\xi}_{i,t}R_{i,t+1}\right).$$
$$(3)$$

---

[1] For a brief introduction to the definitions of B-splines and wavelet bases, please refer to Appendix C.

The estimated Q-function and value function are given by $\widehat{Q}_{\text{IPW}}^\pi(s, a) = \boldsymbol{\xi}^\top(s, a)\widehat{\boldsymbol{\beta}}_{\pi,\text{IPW}}$, $\widehat{V}_{\text{IPW}}^\pi(s) = \boldsymbol{U}_\pi^\top(s)\widehat{\boldsymbol{\beta}}_{\pi,\text{IPW}}$. Given a reference distribution $\mathbb{G}$ on state space $\mathcal{S}$, the integrated value can be estimated as $\widehat{V}_{\text{IPW}}^\pi(\mathbb{G}) = \int_{s\in\mathcal{S}}\widehat{V}_{\text{IPW}}^\pi(s)\mathbb{G}(ds) = \{\int_{s\in\mathcal{S}}\boldsymbol{U}_\pi(s)\mathbb{G}(ds)\}^\top\widehat{\boldsymbol{\beta}}_{\pi,\text{IPW}}$. In practice, the integration $\int_{s\in\mathcal{S}}\boldsymbol{U}_\pi(s)\mathbb{G}(ds)$ can be approximated with sample average of $\boldsymbol{U}_\pi(s)$ using the reference distribution $\mathbb{G}$.

Notice that if we disregard the weighting term $\{1 - \lambda_{i,t+1}(\widehat{\boldsymbol{\psi}}_{nT})\}$ in the denominator, the estimating equation simplifies to $\mathbb{E}_{nT}\{\eta_{i,t+1}\boldsymbol{M}_t(\boldsymbol{\beta}_\pi)\} = \mathbf{0}$, which represents a straightforward adaptation of the classical estimator that only incorporates observed examples into parameter estimation. For simplicity of reference in later sections, we refer to this approach as the Complete Case (CC) estimator. Following the same estimation logic as IPW, we derive

$$\widehat{\boldsymbol{\beta}}_{\pi,\text{CC}} = \underbrace{\left\{\frac{1}{nT}\sum_{i=1}^n\sum_{t=0}^{T-1}\eta_{i,t+1}\boldsymbol{\xi}_{i,t}(\boldsymbol{\xi}_{i,t} - \gamma\boldsymbol{U}_{\pi,i,t+1})^\top\right\}^{-1}}_{\widehat{\boldsymbol{\Sigma}}_{\pi,\text{CC}}}$$
$$\left(\frac{1}{nT}\sum_{i=1}^n\sum_{t=0}^{T-1}\eta_{i,t+1}\boldsymbol{\xi}_{i,t}R_{i,t+1}\right).$$

The estimated value function and integrated value can be similarly obtained by $\widehat{V}_{\text{CC}}^\pi(s) = \boldsymbol{U}_\pi^\top(s)\widehat{\boldsymbol{\beta}}_{\pi,\text{CC}}$, and $\widehat{V}_{\text{CC}}^\pi(\mathbb{G}) = \{\int_{s\in\mathcal{S}}\boldsymbol{U}_\pi(s)\mathbb{G}(ds)\}^\top\widehat{\boldsymbol{\beta}}_{\pi,\text{CC}}$.

When there are no missing data, both the IPW and CC estimators reduce to the base algorithm, and the value estimator is shown to be asymptotically normal as either $n \to \infty$ or $T \to \infty$ under regularity conditions (see Theorem 1 of Shi et al. (2021b)). However, when dealing with incomplete data, the solution to $\mathbb{E}_{nT}\{\eta_{t+1}\boldsymbol{M}_t(\boldsymbol{\beta}_\pi)\} = \mathbf{0}$ may differ from the solution to $\mathbb{E}_{nT}\{\boldsymbol{M}_t(\boldsymbol{\beta}_\pi)\} = \mathbf{0}$. The following theorem outlines when the complete-case value estimator remains valid and when it may not.

**Theorem 4.5.** *Suppose Assumption A.1 holds. $\widehat{V}_{CC}^\pi(\mathbb{G})$ is a consistent estimator if the missing mechanism is ignorable (MAR). However, if the missing mechanism is nonignorable (MNAR), $\widehat{V}_{CC}^\pi(\mathbb{G})$ can be biased.*

Assumption A.1 establishes the necessary conditions that ensure the consistency and asymptotic distribution of value estimation when there is no missing data. To the best of our knowledge, this is the first result establishing the validity of OPE methods in the presence of missing data. As per Theorem 4.5, the complete-case value estimator remains valid if the missing data mechanism is ignorable. However, for nonignorable missingness, further adjustments based on IPW are required to retrieve consistency, as we will show in Theorem 4.6 and 4.7.

**Theorem 4.6** (Bidirectional Consistency). *Assume conditions A.1 and A.2 hold. $\widehat{V}_{IPW}^\pi(\mathbb{G})$ is a consistent estimator*

*for value, i.e., $\widehat{V}_{IPW}^{\pi}(\mathbb{G}) \xrightarrow{p} V^{\pi}(\mathbb{G})$ as either $n \to \infty$ or $T \to \infty$.*

**Theorem 4.7** (Bidirectional Asymptotics). *Assume conditions A.1-A.2 hold. As either $n \to \infty$ or $T \to \infty$, we have*

$$\sqrt{nT}\widehat{\sigma}_{\pi,\text{IPW}}^{-1}(\mathbb{G})\{\widehat{V}_{\text{IPW}}^{\pi}(\mathbb{G}) - V^{\pi}(\mathbb{G})\} \xrightarrow{d} \mathcal{N}(0,1),$$

*where $\widehat{\sigma}_{\pi,IPW}^{2}(\mathbb{G})$ is given by (34) in Appendix G.3.*

The proofs for Theorems 4.5-4.7 can be found in Appendix G.1-G.3. Different from the asymptotic result presented in Shi et al. (2021b), we now take into account the response indicator $\eta_t$ and the uncertainty associated with dropout propensity estimation. Intuitively, observations with higher dropout propensities are assigned higher weights to adjust the data distribution so as to retain the consistency of value function. As such, a two-sided Confidence Interval (CI) for $V^{\pi}(\mathbb{G})$ with significance level $\alpha$ can be constructed as

$$\left[\widehat{V}_{\text{IPW}}^{\pi}(\mathbb{G}) - z_{\alpha/2} \cdot \frac{\widehat{\sigma}_{\pi,\text{IPW}}(\mathbb{G})}{\sqrt{nT}}, \widehat{V}_{\text{IPW}}^{\pi}(\mathbb{G}) + z_{\alpha/2} \cdot \frac{\widehat{\sigma}_{\pi,\text{IPW}}(\mathbb{G})}{\sqrt{nT}}\right],$$

where $z_{\alpha/2}$ is the $(1 - \alpha/2)$-quantile of the standard normal distribution. The complete algorithm is outlined in Algorithm 1 of Appendix E.1.

### 4.3. Dropout Propensity Estimation

Unlike a classical classification problem, modeling $\lambda(\psi)$ under nonignorable missingness is a challenging task and imposes an extra layer of complexity to the problem. It has been proved that if no extra information is provided, it is impossible to identify the observed likelihood, even when both the dropout propensity $\lambda(S_t, A_t, R_{t+1}, S_{t+1})$ and the conditional density function $f(R_{t+1}, S_{t+1}|S_t, A_t)$ are completely parametric (Wang et al., 2014). As introduced in Section 2, the problem of MNAR has been extensively explored through an auxiliary variable called *shadow* variable (Wang et al., 2014; Shao & Wang, 2016; Miao et al., 2016; Zhao & Ma, 2022; Miao et al., 2024), as defined below.

**Definition 4.8.** (Shadow Variable) A variable $Z_t$ is called a shadow variable if
(a) $Z_t \perp\!\!\!\perp \eta_{t+1} \mid (S_t, A_t, R_{t+1}, S_{t+1})$, and
(b) $Z_t \not\perp\!\!\!\perp (R_{t+1}, S_{t+1}) \mid (\eta_{t+1} = 1, S_t, A_t)$.

Typically, shadow variable $Z_t$ excludes the conditional dependency on missingness given $(S_t, A_t, R_{t+1}, S_{t+1})$ (part (a)), while ensuring that it can partially explain the unobserved outcome $(R_{t+1}, S_{t+1})$ for the observed data (part (b)), serving as a "nonresponse instrument". This additional information about the outcome $(R_{t+1}, S_{t+1})$ is crucial for ensuring the identifiability of the observed data likelihood (see Miao et al. (2024) for non-parametric identification), and therefore plays a key role in estimating the value function. A detailed discussion on the practicality of shadow variables in real applications, along with a guide on how to identify a shadow variable, is provided in Appendix B.3.

To estimate the parameter $\psi$ of the dropout propensity model $\lambda(S_t, A_t, R_{t+1}, S_{t+1}; \psi)$, we solve the following estimating equation:

$$\mathbb{E}[\boldsymbol{m}(Z_t, S_t, A_t, R_{t+1}, S_{t+1}; \psi)] = 0, \quad (4)$$

where $\boldsymbol{m}(\psi) = \left\{\frac{\eta_{t+1}}{1-\lambda(\psi)} - 1\right\} \cdot \boldsymbol{h}(S_t, A_t, Z_t)$, and $\boldsymbol{h}(S_t, A_t, Z_t)$ is a vector function of dimension $q$ (same as $\psi$) that can be flexibly determined by the users. A discussion on the selection of the $\boldsymbol{h}$-function is provided in Appendix E.2.3. For flexible estimation options, we adapt both parametric and semi-parametric estimation for the dropout propensity $\lambda(\cdot)$ in Equation (4), with details provided in Appendix B. As such, after obtaining an estimate of $\widehat{\psi}_{nT}$ by Equation (4), the value estimate can be obtained from Equation (3). Thus, the value estimation and inference process is complete.

### 4.4. Extension to Other Off-Policy Evaluation Methods

For valid inference, a potential limitation of our work is that the state dimension cannot be too high due to basis approximation. However, it is worth noting that the idea of using IPW to correct the bias caused by MNAR can be naturally integrated with other OPE methods, such as Fitted Q Evaluation (FQE) (Le et al., 2019). In this case, the Q-function can be approximated with any function class $\mathcal{F}$ and iteratively estimated by minimizing $\ell(Q) = (1/nT)\sum_{i=1}^{n}\sum_{t=0}^{T}\{Q(S_{i,t}, A_{i,t}) - (R_{i,t+1} + \gamma\sum_{a\in\mathcal{A}}\pi(a|S_{i,t+1})Q_{k-1}(S_{i,t+1}, a))\}^2$, where $Q_{k-1}$ is the estimated Q-function obtained from the last iteration. To handle nonignorable missingness, we can incorporate the inverse weighting term into the loss function as follows

$$\widetilde{\ell}(Q) = \frac{1}{nT}\sum_{i=1}^{n}\sum_{t=0}^{T-1}\frac{\eta_{i,t+1}}{1-\lambda_{i,t+1}(\widehat{\psi}_{nT})}\left\{Q(S_{i,t}, A_{i,t}) - \left(R_{i,t+1} + \gamma\sum_{a\in\mathcal{A}}\pi(a|S_{i,t+1})Q_{k-1}(S_{i,t+1}, a)\right)\right\}^2.$$

Other potential extensions beyond direct methods, such as the extension to a Marginalized Importance Sampling (MIS)-based method with theoretical guarantees, are provided in Appendix F.2.

## 5. Simulation

In this section, we demonstrate the performance of our proposed IPW estimator and the theoretical properties established in Section 4.2. To the best of our knowledge, no other method has ever considered OPE under nonignorable missingness, so we mainly focus on the comparison between the IPW and CC estimator. All supplementary code is available at our Github repository.

In our simulation environment, which is based on the setups used in Luckett et al. (2019) and Shi et al. (2021b), the state

variable is a 2-dimensional vector $S_t = (S_t^{(1)}, S_t^{(2)})^\top$, and the action $A_t$ is binary, taking values in $\{0, 1\}$. We evaluate the value function under target policy $\pi(a = 1|s) = \mathbb{1}\{s^{(1)} + s^{(2)} > 0\}$, a deterministic policy characterized by a discontinuous function with respect to the state. For each target policy, the true policy values are estimated with $100,000$ Monte Carlo approximations.

Assume $S_t^{(2)}$ is a shadow variable such that it is correlated with $(R_{t+1}, S_{t+1})$ but uncorrelated with dropout propensity. We consider the MNAR dropout model $\lambda_1(S_t, A_t, R_{t+1}, S_{t+1}) = \{1 + \exp(2.2 + 0.15S_t^{(1)} - 0.3R_{t+1})\}^{-1}$ and the MAR model $\lambda_2(R_t, S_t, A_t) = \{1 + \exp(2.2 + 0.15S_t^{(1)} - 0.3R_t)\}^{-1}$. The difference between the two models is that $\lambda_1$ relies on the next state $S_{t+1}$ through reward $R_{t+1}$ while $\lambda_2$ does not. In this setting, higher reward leads to higher dropout propensity, so the distribution of the observed data is biased towards the low-reward region. More details about data generation are provided in Appendix E.2.

*Table 1.* Results of value estimates and 95% confidence intervals under policy $\pi$. The average bias, MSE values, and ECP are reported for each estimator (with standard error in parenthesis). IPW (P) refers to the IPW estimator with parametric estimation, while IPW (SP) refers to the semi-parametric version.

| $n$ | Dropout | Method | Bias | MSE | ECP |
|---|---|---|---|---|---|
| | no dropout | CC | 0.013 ( 0.602 ) | 1.592 | 0.968 |
| | MAR | CC | -0.028 ( 0.807 ) | 2.198 | 0.972 |
| 500 | MAR | IPW | -0.025 ( 0.806 ) | 2.207 | 0.980 |
| | MNAR | CC | -0.598 ( 0.823 ) | 2.658 | 0.904 |
| | MNAR | IPW (P) | -0.016 ( 0.851 ) | 2.315 | 0.976 |
| | MNAR | IPW (SP) | 0.015 ( 0.861 ) | 2.340 | 0.960 |
| | no dropout | CC | -0.04 ( 0.443 ) | 1.235 | 0.968 |
| | MAR | CC | -0.029 ( 0.58 ) | 1.538 | 0.944 |
| 1000 | MAR | IPW | -0.03 ( 0.582 ) | 1.545 | 0.956 |
| | MNAR | CC | -0.614 ( 0.587 ) | 2.013 | 0.820 |
| | MNAR | IPW (P) | -0.023 ( 0.602 ) | 1.591 | 0.940 |
| | MNAR | IPW (SP) | 0.003 ( 0.608 ) | 1.596 | 0.932 |

Four different combinations of $n$ and $T$ are considered in our experiment, which are $(500, 10)$, $(1000, 10)$, $(500, 25)$, $(1000, 25)$. For each setting, we run 250 experiments. In each experiment, we generate a new dataset, estimate the value, and compute its confidence interval. The Empirical Coverage Probability (ECP) is then calculated as the proportion of intervals, out of 250, that contain the true value of the target policy. We present the results for $T = 10$ in the main paper, the complete set of results are provided in Appendix D.

According to Table 1, the CC estimator remains consistent under MAR, which in line with our theoretical findings. The associated confidence intervals also achieve satisfactory coverage, indicating that no further adjustment is necessary in this scenario. However, when it comes to MNAR, the

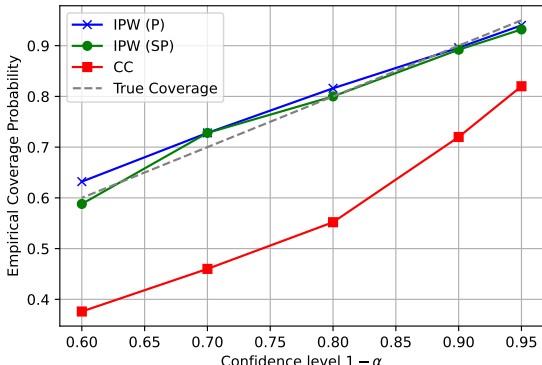

*Figure 2.* Empirical coverage probability with respect to different values of $\alpha$ under $(n, T) = (1000, 10)$ and target policy $\pi$.

CC estimator exhibits high bias, resulting in poor coverage probability of the associated confidence intervals. This under-coverage issue gets worse as the sample size grows. In contrast, the proposed IPW estimator (the last two lines, with a 'P' or 'SP' in parentheses to distinguish between the parametric or semi-parametric model used to estimate the dropout propensity $\lambda(\cdot)$, as detailed in Appendix B) effectively reduces bias and yields more accurate confidence intervals compared to CC, in line with our theoretical results.

Figure 2 visualizes the ECP of the estimated confidence intervals at different values of $\alpha$. As can be seen, our proposed IPW estimator (under both parametric and semi-parametric dropout function estimations) achieves a coverage probability that closely matches the true value of $(1 - \alpha)$. In contrast, the CC estimator exhibits significantly lower coverage. These results further support the findings in Theorems 4.5 to 4.7.

## 6. Real Data

We now demonstrate the accuracy and stability of our value estimates by comparing them with existing baselines using a real-world sepsis dataset from the Medical Information Mart for Intensive Care (MIMIC-III v1.4) database (Johnson et al., 2016). The dataset is processed according to the cohort and inclusion/exclusion criteria outlined by Komorowski et al. (2018). Note that the construction of states, actions, rewards, and the dropout model is based on our limited medical research and is solely for data demonstration purposes. Further validation by domain experts may be necessary for healthcare applications.

Sepsis is a life-threatening condition that arises when the body's response to infection causes damage to its tissues and organs (Singer et al., 2016). For each patient, we have

information on relevant physiological features, including demographics, lab values, vital, and treatment administration information. In order to capture the early phase of sepsis management, data is included from the diagnosis of sepsis and until 48 hours following the onset of sepsis. Intravenous fluids (IV) and vasopressors (VASO) are two commonly administered interventions to correct hypotension caused by infection. Our goal is to evaluate different IV and VASO management policies using this offline dataset.

After close examination of the dataset, we find that around 72% patients did not have complete treatment trajectories during the observational time window either due to early discharge from ICU or mortality. Here we only focus on missingness due to early discharge because mortality within 48 hours of sepsis onset only takes up a very small proportion in our data. Application of offline RL or OPE to MIMIC-III dataset has been studied in several works (Raghu et al., 2017b; 2018; Peng et al., 2018; Li et al., 2020; Sonabend et al., 2020). To the best of our knowledge, none of these works consider the monotone missingness issue.

We construct a 14-dimensional state feature vector aggregated over a time resolution of 4 hours (see details in Appendix E.3). The action space is discretized into three bins: no intravenous fluids and no vasopressors, intravenous fluids only, and vasopressors. We adopt a reward function similar to Raghu et al. (2017a). Specifically,

$$r(\boldsymbol{S}_t, A_t, \boldsymbol{S}_{t+1}) = C_0 \cdot \mathbb{1}(S_{t+1}^{\text{SOFA}} = S_t^{\text{SOFA}} \& S_{t+1}^{\text{SOFA}} > 0)$$
$$+ C_1 \cdot (S_{t+1}^{\text{SOFA}} - S_t^{\text{SOFA}})$$
$$+ C_2 \cdot \tanh(S_{t+1}^{\text{Lactate}} - S_t^{\text{Lactate}}) + C_3,$$

where $C_0 = -5, C_1 = -2.5, C_2 = -10, C_3 = 10$. A higher reward indicates that the patient is in better health. The decision of discharge from ICU is often made based on the current status of the patients, so it is reasonable to assume the missing mechanism is nonignorable, that is, the missingness cannot to be fully accounted for by the observed information in the data.

To handle nonignorable missingness, we incorporate dropout information into value estimation. Glasgow Coma Scale (GCS) measures a person's level of consciousness, which is shown to be an important factor for early discharge prediction (Knight, 2003; Kramer & Zimmerman, 2010; McWilliams et al., 2019), so we include the GCS score $S_t^{\text{GCS}}$ in our dropout model. Besides, we also add the Fraction of inspired oxygen (FiO2), Heart Rate (HR), and Respiratory Rate (RR) at the previous time window into the dropout model. Previous GCS score $S_{t-1}^{\text{GCS}}$ is used as a shadow variable [2]. The target policies to be evaluated include a fitted behavior policy and optimal policies trained from Deep

Q-Network (Mnih et al., 2015), Dueling Double Deep Q-Network (Wang et al., 2016) and Batch-Constrained Deep Q-Learning (BCQ) (Fujimoto et al., 2019). Note that these methods are only used for learning the target policy, while the entire estimation and inference process relies entirely on either the CC or IPW method, as discussed earlier in Section 4. In our implementation, the dataset is split into two parts, with the first part used for learning the optimal policy and the second part for policy evaluation.

Table 2. Results of value estimates and confidence intervals using the original sepsis dataset. In the $\hat{V}^\pi$ column, the number within the parenthesis stands for the standard error. For clarity in comparing CC and IPW, IPW refers specifically to IPW (SP) in the table.

| Policy | Method | $\widehat{V}^\pi$ | CI |
|---|---|---|---|
| Behavior | CC | 2.356 (0.334) | (1.702,3.010) |
| | IPW | 2.377 (0.338) | (1.716,3.039) |
| DQN | CC | 4.459 (0.505) | (3.470,5.448) |
| | IPW | 4.542 (0.506) | (3.551,5.534) |
| Dueling DQN | CC | 4.823 (0.557) | (3.731,5.915) |
| | IPW | 4.925 (0.557) | (3.833,6.016) |
| BCQ | CC | 5.002 (0.609) | (3.808,6.195) |
| | IPW | 5.115 (0.608) | (3.924,6.306) |

Table 2 presents the value estimation results. Since the dropout model $\lambda$ is often unknown in real applications, we adapt the semi-parametric estimation, i.e. IPW (SP) (see Appendix B) in this section and omit the 'SP' when there is no confusion. In most cases, the IPW estimator yields higher value estimates than the CC estimator. This matches our intuition, as patients who had an early discharge were believed to be in better condition than those who did not.

However, for this real-world dataset, the true dropout mechanism is unknown and is hard to verify. To better illustrate the effectiveness of the proposed IPW adjustment, we also build a synthetic dataset consists of 4,490 complete trajectories from the whole sepsis dataset and design a custom dropout hazard model as

$$\lambda(\cdot) = \left[ 1 + \exp\left\{ 4.5 - 0.8 \cdot \mathbb{1}(S_t^{\text{FiO2}} \le 0.6) \right. \right.$$
$$- 0.8 \cdot \mathbb{1}(60 \le S_t^{\text{HR}} \le 100) - 0.6 \cdot \mathbb{1}(10 \le S_t^{\text{RR}} \le 30)$$
$$\left. \left. - 1.5 \cdot \mathbb{1}(S_{t+1}^{\text{GCS}} \ge 14) \right\} \right]^{-1},$$

which is used as the ground truth. Then we apply this dropout model to the complete data and generate a synthetic dataset with nonignorable missingness. The OPE results are summarized in Table 3, we also include the CC estimates calculated from the complete data as a baseline. Table 3 shows that the CC estimator tends to underestimate the value, while our proposed IPW estimator can effectively reduce the bias with respect to the baseline. This pseudo-

---

[2]The rationale for selecting the previous GCS score as a shadow variable is detailed in Appendix B.3.

*Table 3.* Results of value estimates and confidence intervals using the synthetic sepsis dataset. In the $\hat{V}^\pi$ column, the number within the parenthesis stands for the standard error.

| Policy | Dropout | Method | $\hat{V}^\pi$ | CI |
|---|---|---|---|---|
| Behavior | no dropout | CC | 1.075 (0.299) | (0.487,1.662) |
| | MNAR | CC | 0.504 (0.467) | (-0.412,1.420) |
| | MNAR | IPW | 0.988 (0.501) | (0.005,1.971) |
| DQN | no dropout | CC | 2.160 (0.448) | (1.282,3.039) |
| | MNAR | CC | 1.437 (0.727) | (0.011,2.863) |
| | MNAR | IPW | 2.227 (0.742) | (0.772,3.682) |
| Dueling DQN | no dropout | CC | 2.368 (0.472) | (1.442,3.293) |
| | MNAR | CC | 1.672 (0.750) | (0.202,3.142) |
| | MNAR | IPW | 2.450 (0.758) | (0.962,3.938) |
| BCQ | no dropout | CC | 2.085 (0.470) | (1.163,3.008) |
| | MNAR | CC | 1.403 (0.779) | (-0.124,2.931) |
| | MNAR | IPW | 2.147 (0.795) | (0.589,3.706) |

real data further illustrates the effectiveness of our approach in correcting bias introduced by MNAR.

## 7. Summary

In this work, we comprehensively studied the problem of missingness in off-policy evaluation, providing theoretical guarantees with statistical inference on the value estimates. One limitation of our method results from the difficulty of justifying the missingness mechanism, which often require context and subject-matter knowledge. In fact, handling nonignorable missingness still remains an active research area in the field of missing data methodology, and we leave the integration of these evolving advancements for future exploration. Additionally, we believe that extending this work to learning optimal policies from offline data with nonignorable missingness presents an intriguing direction for future research.

## Acknowledgments

This research was conducted with the support of NSF Grant DMS-2113637. The authors gratefully acknowledge this funding, which made it possible to carry out the simulation studies and real data analysis.

## Impact Statement

To the best of our knowledge, this paper is the first to address the missing data problem, particularly missing not at random (MNAR), within the context of off-policy evaluation. Our approach has broad applicability to real-world domains such as healthcare, robotics, recommendation systems, and beyond. We believe that this work, along with the potential extensions discussed in Section 4.4 and Appendices, will inspire future research aimed at improving evaluation methods and decision-making in various machine learning applications.

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

## A. Assumptions

In this section, we provide the assumptions for the theoretical results. The following assumption is introduced by Shi et al. (2021b) to ensure consistency and asymptotic distribution of value estimation when there is no missing data.

**Assumption A.1.** The following conditions hold.

(a) The transition kernel $p(\cdot|s, a)$ is absolutely continuous with respect to the Lebesgue measure, then there exists some transition density function $\tilde{q}$ such that $p(ds'|s, a) = \tilde{q}(s'|s, a)\, ds'$. Let $\Lambda(p, c)$ denotes the class of $p$-smooth functions as follows

$$\Lambda(p, c) = \left\{ h : \sup_{\|\alpha\|_1 \leq \lfloor p \rfloor} \sup_{s \in \mathcal{S}} |D^\alpha h(s)| \leq c, \sup_{\|\alpha\|_1 = \lfloor p \rfloor} \sup_{\substack{s, s' \in \mathcal{S} \\ s \neq s'}} \frac{|D^\alpha h(s) - D^\alpha h(s')|}{\|s - s'\|_2^{p - \lfloor p \rfloor}} \leq c \right\},$$

where $D^\alpha$ denotes the differential operator $D^\alpha h(s) = \frac{\partial^{\|\alpha\|_1} h(s)}{\partial s_1^{\alpha_1} \cdots \partial s_d^{\alpha_d}}$, $s_j$ denotes the $j$-th element of $s$, $\lfloor p \rfloor$ denote the largest integer that is smaller than $p$. Assume there exist some $p, c > 0$ such that $r(\cdot, a), \tilde{q}(s'|\cdot, a) \in \Lambda(p, c)$ for any $a \in \mathcal{A}, s' \in \mathcal{S}$.

(b) Let $\mathrm{BSpl}(L, r)$ denote a tensor-product B-spline basis of degree $r$ and dimension $L$ on $[0, 1]^d$, and let $Wav(L, r)$ denote a tensor-product Wavelet basis of regularity $r$ and dimension $L$ on $[0, 1]^d$. The sieve $\Phi_L$ is either $\mathrm{BSpl}(L, r)$ or $\mathrm{Wav}(L, r)$ with $r > \max(p, 1)$.

(c) Assume the Markov chain has a unique invariant distribution with some density function $\mu(\cdot)$ on $\mathcal{S}$, the probability density function of $S_0$ is denoted as $\nu_0$. The density functions $\mu$ and $\nu_0$ are uniformly bounded away from $0$ and $\infty$ on $\mathcal{S}$.

(d) Suppose (i) and (ii) hold when $T \to \infty$ and (iii) holds when $T$ is bounded.

   (i) $\lambda_{\min}\left[ \int_{s \in \mathcal{S}} \sum_{a \in \mathcal{A}} \left\{ \boldsymbol{\xi}(s, a)\boldsymbol{\xi}^\top(s, a) - \gamma^2 \boldsymbol{u}_\pi(s, a)\boldsymbol{u}_\pi^\top(s, a) \right\} b(a|s)\mu(s)ds \right] \geq \bar{c}$ for some constant $\bar{c} > 0$, where $\boldsymbol{u}_\pi(s, a) = \mathbb{E}\left\{ \boldsymbol{U}_\pi(S_1)|S_0 = s, A_0 = a \right\}$ and $\lambda_{\min}(K)$ denotes the minimum eigenvalue of a matrix $K$.

   (ii) The Markov chain $\{S_t\}_{t \geq 0}$ is geometrically ergodic, i.e, there exists some function $M(\cdot)$ on $\mathcal{S}$ and some constant $\rho < 1$ such that $\int_{s \in \mathcal{S}} M(s)\mu(s)ds < +\infty$ and $\|p_S^t(\cdot \mid s) - \mu(\cdot)\|_{TV} \leq M(s)\rho^t$ for any $t \geq 0$, where $\|\cdot\|_{TV}$ denotes the total variation norm, $p_S^t(\mathcal{B}|s) = P(S_t \in \mathcal{B}|S_0 = s)$ is the $t$-step transition kernel.

   (iii) $\lambda_{\min}\left[ \sum_{t=0}^{T-1} \mathbb{E}\left\{ \boldsymbol{\xi}_t\boldsymbol{\xi}_t^\top - \gamma^2 \boldsymbol{u}_\pi(S_t, A_t)\boldsymbol{u}_\pi^\top(S_t, A_t) \right\} \right] \geq T\bar{c}$ for some constant $\bar{c} > 0$.

(e) The number of basis $L$ satisfies $L = o\{\sqrt{nT}/\log(nT)\}$, $L^{2p/d} \gg nT\{1 + \|\int_s \Phi_L(s)\mathbb{G}(ds)\|_2^{-2}\}$.

(f) There exists some constant $c_0 \geq 1$ such that

$$\delta_\pi(s, a) = \mathbb{E}\left[ \left\{ R_1 + \gamma \sum_{a \in \mathcal{A}} \pi(a|S_1)Q^\pi(S_1, a) - Q^\pi(S_0, A_0) \right\}^2 \middle| S_0 = s, A_0 = a \right] \geq c_0^{-1}$$

for any $s \in \mathcal{S}, a \in \mathcal{A}$, and $P(\max_t |R_t| \leq c_0) = 1$.

These assumptions together guarantee consistent value estimation under complete data. Assumption A.1(a) basically assumes the smoothness of the reward function $r$ and the transition density function $\tilde{q}$ with respect to the current state $s$, this allows us to establish the smoothness of the Q-function, which is critical when deriving inference for the value function. Assumption A.1(b) specifies the types of sieve $\Phi_L$ to approximate the Q-function, which is more of a claim or explanation rather than a strict assumption. Here we consider two types of basis functions, which are standard choices for such problems and serve to simplify the analysis while ensuring general applicability.

Assumption A.1(c) is a mild condition on the marginal distribution over states, when $\nu_0 = \mu$, $\{S_t\}_{t \geq 0}$ is stationary. Assumption A.1(d) is imposed to guarantee the invertiblility of $\widehat{\Sigma}_\pi$. The geometric ergodicity condition in Assumption A.1(d)(ii) ensures that $\{S_t\}_{t \geq 0}$ is exponentially $\beta$-mixing (see Theorem 3.7 of Bradley (2005)). We remark that the geometric ergodicity condition is less restrictive than the independence assumption imposed in some existing reinforcement learning literature (e.g., Dai et al. (2020)). For Assumption A.1(e), the constraint on the number of basis functions $L$

controls the smoothness of the basis function, which in turn determines how closely the linear sieve basis function can approximate the true $Q$ function. In the proof of asymptotic normality, we rely on this condition on $L$ to establish that $\sup_{s \in \mathcal{S}, a \in \mathcal{A}} |Q(\pi; s, a) - \Phi_L^\top(s)\beta_{\pi,a}^*| \leq O(L^{-p/d})$, which guarantees the consistency and asymptotic behavior of $\widehat{\beta}$ and the value estimates. Assumption A.1(f) is a mild condition on the randomness of observed the reward $R_{t+1}$ around $r(S_t, A_t)$ and the uniform boundedness of the observed reward.

Besides the assumptions on Q-function approximation, we also make the following assumption regarding the dropout propensity.

**Assumption A.2.** We assume the following conditions hold.

(a) The dropout propensity model is correctly specified such that $\lambda(S_t, A_t, R_{t+1}, S_{t+1}) = \lambda(S_t, A_t, R_{t+1}, S_{t+1}; \psi^*)$ for some $\psi^*$.

(b) There exist some $c_\lambda > 0$ such that $1 - \lambda(S_t, A_t, R_{t+1}, S_{t+1}) \geq c_\lambda$.

(c) There exits a *shadow* variable $Z_t$ at each stage, such that

$$Z_t \perp\!\!\!\perp \eta_{t+1} \mid (S_t, A_t, R_{t+1}, S_{t+1}), \quad \text{and} \quad Z_t \not\perp\!\!\!\perp (R_{t+1}, S_{t+1}) \mid (\eta_{t+1} = 1, S_t, A_t)$$

Assumption A.2(a) ensures that, for either the parametric or semi-parametric model specified in Appendix B, the chosen model class should include the true model of $\lambda(\cdot)$. This assumption is essential for guaranteeing the statistical properties of the parameter estimate for $\psi$. Assumption A.2(b) is similar to the positivity assumption described by Rosenbaum & Rubin (1983) in causal inference. It requires that there is always a non-zero probability, $c_\lambda$, of observing the data, avoiding cases of pure dropout. Assumption A.2(c), as explained in Section 4.3, ensures there is an auxiliary variable (i.e. a *shadow* variable) that helps identify the model, which is necessary in handling nonignorable missingness (Shao & Wang, 2016; Wang et al., 2014; Zhao & Ma, 2022; Miao et al., 2024).

## B. Estimation Details of the Dropout Propensity Model

To solve for $\beta_\pi$, the first step is to fit the dropout propensity model. In this section, we detail two approaches for estimating the dropout propensity, which can be incorporated into our IPW-based value estimation framework.

For ignorable missingness, the dropout propensity function can be simplified to $\lambda(S_t, A_t)$, since $\eta_{t+1}$ is conditionally independent of $S_{t+1}$ and $R_{t+1}$. In such cases, the propensity can be modeled with any binary classification method.

Unlike ignorable missingness, modeling the nonignorable missingness is much more challenging. The difficulty lies in that if both the dropout propensity $\lambda(S_t, A_t, R_{t+1}, S_{t+1})$ and the conditional density function $f(R_{t+1}, S_{t+1}|S_t, A_t)$ are completely unknown, the joint distribution of $(\eta_{t+1}, R_{t+1}, S_{t+1})$ given $(S_t, A_t)$ is non-identifiable (Rotnitzky & Robins, 1997). As discussed in Section 4.3 of the main paper, our goal is to solve the following estimating equation:

$$\mathbb{E}[\boldsymbol{m}(Z_t, S_t, A_t, R_{t+1}, S_{t+1}; \boldsymbol{\psi})] = 0,$$

$$\text{where } \boldsymbol{m}(Z_t, S_t, A_t, R_{t+1}, S_{t+1}; \boldsymbol{\psi}) = \left\{ \frac{\eta_{t+1}}{1 - \lambda(S_t, A_t, R_{t+1}, S_{t+1}; \boldsymbol{\psi})} - 1 \right\} \cdot \boldsymbol{h}(S_t, A_t, Z_t), \tag{5}$$

with $\boldsymbol{h}(S_t, A_t, Z_t)$ denoting a user-specified vector function of dimension $q$, same as $\boldsymbol{\psi}$.

For simplicity, we denote $\boldsymbol{m}(Z_t, S_t, A_t, R_{t+1}, S_{t+1}; \boldsymbol{\psi})$ as $\boldsymbol{m}_t(\boldsymbol{\psi})$, and $\boldsymbol{h}(S_t, A_t, Z_t) = \boldsymbol{h}_t$ when there is no confusion. That is, $\mathbb{E}[\boldsymbol{m}_t(\boldsymbol{\psi})] = 0$. Specifically, we adapt both parametric and semi-parametric approaches from the single-stage survival analysis literature that address MNAR, which are explained in the following two subsections.

### B.1. Parametric Estimation for $\lambda(\psi)$ under MNAR

If the parametric form of the dropout model $\lambda(\cdot)$ is known, the simplest approach is to directly substitute this parametric form into Equation (5) and solve for $\psi$ using the Method of Moments (MoM) or gradient-based algorithms, such as gradient descent (GD). In our simulation studies, we found that a generalized MoM (GMM) approach, where the weights are determined by the inverse of the covariance matrix, tends to outperform the classical MoM, especially when $\psi$ is relatively high-dimensional.

**B.2. Semi-Parametric Estimation for $\lambda(\psi)$ under MNAR**

Inspired by the recent development of the semiparametric framework to model nonignorable missing data (Kim & Yu, 2011; Shao & Wang, 2016), we consider a semiparametric exponential tilting model for the dropout propensity as $\lambda(S_t, A_t, R_{t+1}, S_{t+1}; \psi) = \{1 + \exp[g(S_t, A_t) + V_{t+1}^\top \psi]\}^{-1}$, where $\psi \in \mathbb{R}^q$ is an unknown tilting parameter to learn, $V_{t+1} \in \mathbb{R}^q$ are features mapped from $(R_{t+1}, S_{t+1})$, $g(\cdot)$ is a non-parametric function of observed variables $(S_t, A_t)$. For succinctness, we suppress the data arguments in $\lambda(S_t, A_t, R_{t+1}, S_{t+1}; \psi)$ and write it as $\lambda_{t+1}(\psi)$.

Based on the definition of the shadow variable (which is the same as *instrumental* variable originally introduced in Shao & Wang (2016) [3]), $Z_t$ can be removed from the modeling process of the dropout propensity function. For clarity, we will refer to it as the shadow variable throughout the remainder of this work. Denote the non-shadow part of $(S_t, A_t)$ as $\mathcal{U}_t$, the exponential tilting model can be rewritten as

$$\lambda_{t+1}(\psi) = \left\{ 1 + \exp[g(\mathcal{U}_t) + \psi^\top V_{t+1}] \right\}^{-1}. \tag{6}$$

According to the definition, a shadow variable $Z_t$ is a covariate in $(S_t, A_t)$ that is related to the outcome $(R_{t+1}, S_{t+1})$ but not related to the dropout propensity given other covariates. With the shadow variable, multiple estimating equations can be constructed to estimate the parameters of interest, $g$ and $\psi$. If the shadow variable $Z_t$ is originally discrete with $\widetilde{L}$ levels, the $\widetilde{L}$ estimating equations can be constructed as $\mathbb{E}_{nT}\{\mathbb{1}(Z_t = l)(\eta_{t+1}(1 - \lambda_{t+1}(\psi))^{-1} - 1)\} = 0, l \in \{1, \ldots, \widetilde{L}\}$. Here we use the notation $\widetilde{L}$ to differentiate it from the notation $L$, which represents the number of basis functions. In the case of $Z_t$ being a continuous variable, it can first be discretized into $\widetilde{L}$ bins. In order to estimate $\psi$, the non-parametric component $g$ is first profiled with a kernel estimator. The remaining $\widetilde{L} - 1$ estimating equations are used to solve for $\psi$ using the Generalized Method of Moments (GMM) (Hansen, 1982). Notice that this semiparametric approach to estimating $\lambda(\cdot)$ typically involves a specialized form of the estimating equation (5), where each dimension of $h(S_t, A_t, Z_t)$ is set to $\mathbb{1}(Z_t = l)$ for $l \in 1, \ldots, \widetilde{L}$. Instead of directly specifying a parametric form for $\lambda(\cdot)$, which could potentially be misspecified, we combine a non-parametric function $g(\cdot)$ with a parametric component to solve for $\psi$ and thus allows for greater flexibility.

**B.3. How to Find the Shadow Variable**

Identifying a suitable shadow variable remains a challenging issue in the literature on MNAR, even without accounting for sequential decision-making scenarios like MDPs. As defined in Definition 4.8, a shadow variable is an auxiliary variable that has a causal relationship with the outcome of interest while remaining conditionally independent of the dropout pattern. This assumption is often plausible in various empirical settings. For example, in our real-world application, certain baseline measurements taken before treatment, such as the previous GCS score, can act as shadow variables. This is because measurements like the previous GCS score typically help explain the patient's health status at the current stage, but do not offer additional information about dropout when conditioned on the outcome (the patient's health status).

A comparable scenario arises in healthcare, as described in Miao et al. (2024). In a study investigating the mental health of children in Connecticut (Zahner et al., 1992), researchers sought to assess the prevalence of students exhibiting abnormal psychopathological traits based on teacher evaluations, which were prone to missing data. The likelihood of a teacher providing an assessment might depend on their evaluation of the student but is unlikely to influence a separate parent report, provided the teacher's evaluation and all observed covariates are accounted for. Additionally, the parent report is expected to strongly correlate with the teacher's assessment, making the parent's evaluation a valid shadow variable in this context. Other illustrative examples of shadow variables can be found in Wang et al. (2014); Zhao & Shao (2015); Zhao & Ma (2018).

While the above examples provide some insight into understanding and identifying shadow variables in real-world applications, the practical task of identifying shadow variables can still be challenging. In cases where definitive domain knowledge is lacking or the conditions for a chosen shadow variable are only partially satisfied, the observed likelihood becomes non-identifiable, and the resulting estimating equation is likely to fail. In such situations, it is advisable to perform a sensitivity analysis (Robins et al., 2000) on the selected shadow variable to evaluate how variations in the choice impact the results.

---

[3]To avoid confusion with existing literature on instrumental variables for missing data (Tchetgen Tchetgen & Wirth, 2017; Sun et al., 2018), we will use the term shadow variable exclusively in the following discussion, avoiding the term instrumental variable, as the identification framework presented here differs slightly from that in other studies.

## B.4. Dropout Estimation Accuracy

In this section, we briefly evaluate the accuracy of dropout propensity estimation for both parametric and semi-parametric models. Accurate estimation of dropout propensity is crucial in MNAR settings, as it appears in the denominator of the IPW estimator and directly influences the estimation accuracy of $\widehat{\psi}$ and the value function.

Figures 3 is generated based on the simulation setting described in Section 5, which provides an overall assessment of the estimation accuracy of $\lambda(\widehat{\psi})$. As shown in the MSE distribution plot, both parametric and semi-parametric models perform adequately well. As expected, the semi-parametric model yields slightly higher MSE than the parametric model due to its greater flexibility and potential for model misspecification.

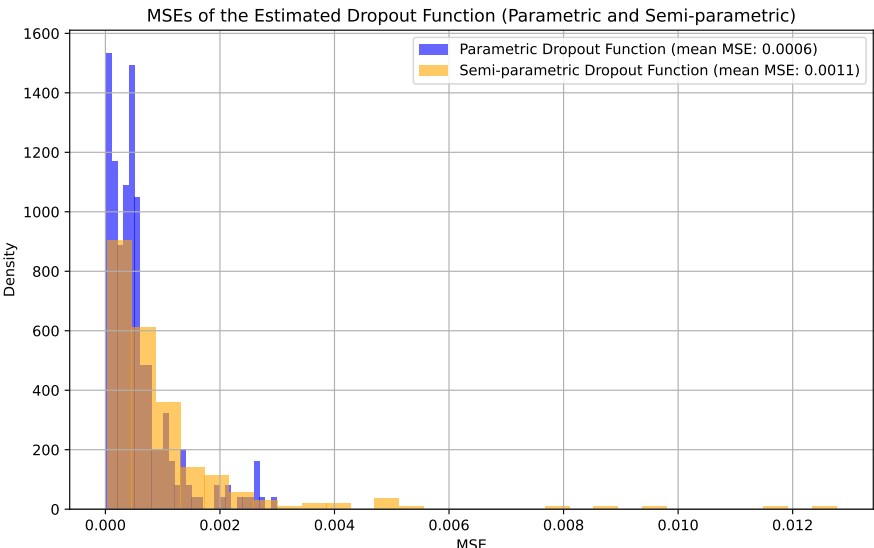

*Figure 3.* Mean Squared Errors (MSEs) for both parametric and semi-parametric dropout models are close to zero, demonstrating the effectiveness of the dropout propensity estimation.

# C. A Brief Introduction to B-Spline and Wavelet Basis Functions

## C.1. B-Spline Basis Function

A B-spline basis function $B_{i,k}(x)$ is a collection of piecewise polynomial functions defined over a partition of the domain (called knots), which are used to approximate smooth functions. For a non-decreasing sequence of knots $\{t_i\}_{i=0}^{n+k}$, the B-spline basis of degree $k$ is defined recursively via the Cox–de Boor recursion formula. When degree $k = 0$,

$$B_{i,0}(x) = \begin{cases} 1 & \text{if } t_i \leq x < t_{i+1}, \\ 0 & \text{otherwise.} \end{cases}$$

When degree $k > 0$,

$$B_{i,k}(x) = \frac{x - t_i}{t_{i+k} - t_i} B_{i,k-1}(x) + \frac{t_{i+k+1} - x}{t_{i+k+1} - t_{i+1}} B_{i+1,k-1}(x),$$

where terms with zero denominators are treated as zero.

By adjusting the number and placement of knots, B-splines can flexibly combine piecewise polynomial segments to create smooth functions that approximate complex functional forms.

## C.2. Wavelet Basis Function

Wavelet basis functions are localized in both time (or space) and frequency, making them suitable for representing functions with local irregularities. A wavelet basis is constructed from a single *mother wavelet* $\psi(x)$ via dilations and translations:

$$\psi_{j,k}(x) = 2^{j/2}\psi(2^j x - k),$$

where $j \in \mathbb{Z}$ denotes the *scale* (frequency) and $k \in \mathbb{Z}$ the *translation* (location).

Commonly used wavelets such as Haar and Daubechies form orthonormal bases in $L^2(\mathbb{R})$ and are particularly useful in nonparametric regression, signal processing, and capturing complex functions with discontinuities.

## D. More on Simulations

In this section, we present additional simulation results to further support our study. We provide comprehensive results for two dropout models to evaluate the performance of different value estimators. The MNAR dropout model is defined as $\lambda_1(S_t, A_t, R_{t+1}, S_{t+1}) = \{1 + \exp(\psi_1 + \psi_2 S_t^{(1)} + \psi_3 R_{t+1})\}^{-1}$, and the MAR model as $\lambda_2(R_t, S_t, A_t) = \{1 + \exp(\psi_1 + \psi_2 S_t^{(1)} + \psi_3 R_t)\}^{-1}$. The true parameter values of $\psi = [\psi_1, \psi_2, \psi_3]^T$ are set to $[2, 0.08, -0.15]^T$ in Setting 1 and $[2, 0.15, -0.3]^T$ in Setting 2 (results for Setting 2 are presented in the main paper). The full set of comparisons, covering four combinations of $(n, T)$ with values $(500, 10)$, $(1000, 10)$, $(500, 25)$, and $(1000, 25)$, are provided in Tables 4-5.

*Table 4.* Results of value estimates and 95% confidence intervals for Setting 1 in the 2D-Linear environment. The average bias, MSE values, and ECP are reported for each estimator (with standard error in parenthesis).

| $T$ | $n$ | DROPOUT | METHOD | BIAS | MSE | ECP |
|---|---|---|---|---|---|---|
| 10 | 500 | NO DROPOUT | CC | 0.013 ( 0.602 ) | 1.592 | 0.968 |
| | | MAR | CC | 0.009 ( 0.779 ) | 2.095 | 0.964 |
| | | MAR | IPW | 0.012 ( 0.781 ) | 2.097 | 0.964 |
| | | MNAR | CC | -0.246 ( 0.806 ) | 2.205 | 0.948 |
| | | MNAR | IPW (P) | 0.007 ( 0.819 ) | 2.158 | 0.948 |
| | | MNAR | IPW (SP) | 0.013 ( 0.837 ) | 2.188 | 0.948 |
| | 1000 | NO DROPOUT | CC | -0.04 ( 0.443 ) | 1.235 | 0.968 |
| | | MAR | CC | -0.05 ( 0.566 ) | 1.471 | 0.952 |
| | | MAR | IPW | -0.051 ( 0.566 ) | 1.472 | 0.952 |
| | | MNAR | CC | -0.307 ( 0.555 ) | 1.588 | 0.920 |
| | | MNAR | IPW (P) | -0.05 ( 0.563 ) | 1.487 | 0.940 |
| | | MNAR | IPW (SP) | -0.056 ( 0.558 ) | 1.482 | 0.944 |
| 25 | 500 | NO DROPOUT | CC | -0.027 ( 0.413 ) | 1.168 | 0.944 |
| | | MAR | CC | -0.022 ( 0.653 ) | 1.748 | 0.964 |
| | | MAR | IPW | -0.02 ( 0.651 ) | 1.742 | 0.976 |
| | | MNAR | CC | -0.296 ( 0.702 ) | 1.923 | 0.920 |
| | | MNAR | IPW (P) | -0.012 ( 0.712 ) | 1.837 | 0.960 |
| | | MNAR | IPW (SP) | -0.045 ( 0.718 ) | 1.841 | 0.936 |
| | 1000 | NO DROPOUT | CC | -0.033 ( 0.245 ) | 0.985 | 0.976 |
| | | MAR | CC | -0.052 ( 0.46 ) | 1.291 | 0.960 |
| | | MAR | IPW | -0.052 ( 0.46 ) | 1.289 | 0.964 |
| | | MNAR | CC | -0.32 ( 0.47 ) | 1.439 | 0.896 |
| | | MNAR | IPW (P) | -0.068 ( 0.474 ) | 1.322 | 0.964 |
| | | MNAR | IPW (SP) | -0.062 ( 0.475 ) | 1.322 | 0.952 |

The key takeaways from the results in the tables are as follows:

1. When the missing data is ignorable (i.e., under MAR), both the CC and IPW estimators perform well in terms of estimation accuracy and empirical coverage.

2. In the case of nonignorable missingness (i.e., MNAR), the CC estimator suffers from significant bias, higher MSE, and lower coverage, which negatively impacts the performance of value estimation. In contrast, our proposed IPW-based

estimator, whether using a parametric or semi-parametric dropout model, remains stable in terms of both bias and coverage.

3. Across different values of $(n, T)$, we observe that increasing the total number of data points, i.e., $nT$, leads to slightly more stable estimates, particularly when examining the standard error in the Bias column and the MSE in the MSE column.

*Table 5.* Results of value estimates and 95% confidence intervals for Setting 2 in the 2D-Linear environment. The average bias, MSE values, and ECP are reported for each estimator (with standard error in parenthesis).

| $T$ | $n$ | DROPOUT | METHOD | BIAS | MSE | ECP |
|---|---|---|---|---|---|---|
| 10 | 500 | NO DROPOUT | CC | 0.013 ( 0.602 ) | 1.592 | 0.968 |
| | | MAR | CC | -0.028 ( 0.807 ) | 2.198 | 0.972 |
| | | MAR | IPW | -0.025 ( 0.806 ) | 2.207 | 0.980 |
| | | MNAR | CC | -0.598 ( 0.823 ) | 2.658 | 0.904 |
| | | MNAR | IPW (P) | -0.016 ( 0.851 ) | 2.315 | 0.976 |
| | | MNAR | IPW (SP) | 0.015 ( 0.861 ) | 2.340 | 0.960 |
| | 1000 | NO DROPOUT | CC | -0.04 ( 0.443 ) | 1.235 | 0.968 |
| | | MAR | CC | -0.029 ( 0.58 ) | 1.538 | 0.944 |
| | | MAR | IPW | -0.03 ( 0.582 ) | 1.545 | 0.956 |
| | | MNAR | CC | -0.614 ( 0.587 ) | 2.013 | 0.820 |
| | | MNAR | IPW (P) | -0.023 ( 0.602 ) | 1.591 | 0.940 |
| | | MNAR | IPW (SP) | 0.003 ( 0.608 ) | 1.596 | 0.932 |
| 25 | 500 | NO DROPOUT | CC | -0.027 ( 0.413 ) | 1.168 | 0.944 |
| | | MAR | CC | -0.021 ( 0.723 ) | 1.925 | 0.952 |
| | | MAR | IPW | -0.016 ( 0.723 ) | 1.927 | 0.956 |
| | | MNAR | CC | -0.655 ( 0.743 ) | 2.476 | 0.864 |
| | | MNAR | IPW (P) | -0.072 ( 0.76 ) | 2.025 | 0.944 |
| | | MNAR | IPW (SP) | -0.041 ( 0.764 ) | 2.031 | 0.964 |
| | 1000 | NO DROPOUT | CC | -0.033 ( 0.245 ) | 0.985 | 0.976 |
| | | MAR | CC | -0.034 ( 0.485 ) | 1.361 | 0.972 |
| | | MAR | IPW | -0.034 ( 0.486 ) | 1.361 | 0.976 |
| | | MNAR | CC | -0.648 ( 0.509 ) | 1.919 | 0.780 |
| | | MNAR | IPW (P) | -0.057 ( 0.523 ) | 1.443 | 0.972 |
| | | MNAR | IPW (SP) | -0.033 ( 0.523 ) | 1.439 | 0.956 |

# E. Additional Experimental Details

In this section, we provide more details on the experiments and our implementation.

## E.1. Algorithm

The outline for the proposed estimator is presented in Algorithm 1. In Line 5, when employing a semi-parametric model to estimate the dropout propensity function (see Appendix B.2), an approximation of the estimated standard deviation $\widetilde{\sigma}_{\pi,\text{IPW}}(\mathbb{G})$, as given in Equation (35), can be used as a substitute for the exact solution $\widehat{\sigma}_{\pi,\text{IPW}}(\mathbb{G})$. Accordingly, the confidence interval can be obtained as $\left[ \widehat{V}_{\text{IPW}}^{\pi}(\mathbb{G}) \pm z_{\alpha/2}(nT)^{-1/2}\widetilde{\sigma}_{\pi,\text{IPW}}(\mathbb{G}) \right]$.

## E.2. Simulation Settings

### E.2.1. DATA GENERATION

The initial states are generated from the standard bivariate normal distribution $\mathcal{N}(\mathbf{0}_2, \mathbf{I}_2)$. For $t \geq 0$, we slightly modify the original dynamics and consider the following transition: $S_{t+1}^{(1)} = (2A_t - 1)S_t^{(1)} + \varepsilon_t^{(1)}$, $S_{t+1}^{(2)} = (1 - 2A_t)S_t^{(2)} + \varepsilon_t^{(2)}$, where $\varepsilon_t^{(1)}$ and $\varepsilon_t^{(2)}$ are independent $\mathcal{N}(0, 0.25)$ random variables. The immediate reward is designed as $R_{t+1} = 2S_{t+1}^{(1)} + S_{t+1}^{(2)} + 0.5S_t^{(2)} - 0.25(2A_t - 1) + \varepsilon_t^{(3)}$, where $\varepsilon_t^{(3)} \sim \mathcal{N}(0, 10^{-4})$. The behavior policy follows a Bernoulli distribution with a mean of 0.5. Throughout the simulation studies, we set the discount factor $\gamma$ to 0.9.

---

**Algorithm 1** Off-Policy Evaluation with Nonignorable Monotone Missingness

1: **Input:** Observed dataset $\mathcal{D} = \{\tau_i\}_{i=1}^n$, target policy $\pi$, discount factor $\gamma$, number of basis $L$
2: Fit dropout propensity model (6) using the semiparametric approach
3: Construct a set of basis $\Phi_L(s)$ from state variables and estimate $\widehat{\boldsymbol{\beta}}_{\pi,\text{IPW}}$ by (3)
4: Estimate value $\widehat{V}_{\text{IPW}}^\pi(s) = \boldsymbol{U}_\pi^\top(s)\widehat{\boldsymbol{\beta}}_{\pi,\text{IPW}}^\pi$, $\widehat{V}_{\text{IPW}}^\pi(\mathbb{G}) = \int_{s \in \mathcal{S}} \widehat{V}_{\text{IPW}}^\pi(s)\mathbb{G}(ds)$
5: Calculate the asymptotic variance $\widehat{\sigma}_{\pi,\text{IPW}}^2(\mathbb{G})$ given by Equation (34)
6: **Return:** the CI of $V^\pi(\mathbb{G})$: $\left[\widehat{V}_{\text{IPW}}^\pi(\mathbb{G}) \pm z_{\alpha/2}(nT)^{-1/2}\widehat{\sigma}_{\pi,\text{IPW}}(\mathbb{G})\right]$

---

To generate complete data with $n$ trajectories, we first sample $n$ initial states from the reference distribution $\mathbb{G}$, and then generate the action, next state and reward following the generative model described above. This process is repeated until reaching the maximum horizon $T$. To generate incomplete data, we first calculate the dropout probability $\lambda_{i,t}$ at each step using the dropout model $\lambda_j(\cdot)$ defined in Section 5, this corresponds to the probability of subject $i$ dropping out after taking action $A_t$. Given the dropout probability, we sample the response indicator $\eta_{i,t+1}$ from a Bernoulli distribution with mean $(1 - \lambda_{i,t})$. To control the overall missing rate, we also set a no-dropout period of two steps, i.e., $\eta_{i,0} = \eta_{i,1} = 1$. After the second step, the dropout probability is applied and a trajectory will terminate when the response indicator $\eta_{i,t}$ turns 0.

### E.2.2. COMPLETE-CASE (CC) ESTIMATOR

The OPE step of Shi et al. (2021b) approximates the Q-function with linear sieves, $Q^\pi(s, a) \approx \Phi_L^\top(s)\boldsymbol{\beta}_{\pi,a}$, where $\Phi_L(\cdot) = \left\{\boldsymbol{\phi}_{L,1}(\cdot), \cdots, \boldsymbol{\phi}_{L,L}(\cdot)\right\}^\top$ is a vector consisting of $L$ spline bases. In our implementation, we first scale the state variables onto $[0, 1]$ and then construct 6 cubic B-spline bases for each dimension, where the knots are placed at equally-spaced quantiles of the transformed state variables. To avoid extrapolation of the basis function, three repeated knots are placed on the boundary. The tensor product of the basis for each dimension is used to construct the final basis, hence $L = 36$. The number of basis functions $L$ is allowed to grow with the sample size to reduce the approximation error. For a fair comparison, here we fix $L = 36$ throughout the experiments despite the sample sizes. The CC estimator of the Q-function parameter $\boldsymbol{\beta}_\pi^*$ is given in (4.2). The matrix inversion of $\widehat{\boldsymbol{\Sigma}}_\pi \in \mathbb{R}^{mL \times mL}$ tends to be unstable when $mL$ is large, so we add a small ridge penalty with weight $10^{-5}$ to improve the stability. Given $\widehat{\boldsymbol{\beta}}_\pi$, the value function can be calculated as $\widehat{V}^\pi(s) = \boldsymbol{U}_\pi^\top(s)\widehat{\boldsymbol{\beta}}_\pi$. We approximate the integrated value $\widehat{V}^\pi(\mathbb{G}) = \int_{\boldsymbol{s} \in \mathcal{S}} \widehat{V}^\pi(\boldsymbol{s})\mathbb{G}(d\boldsymbol{s})$ by sampling 10,000 states from the reference distribution $\mathbb{G}$ and take the average of the estimated value for each state.

### E.2.3. SOLVING THE ESTIMATING EQUATION (4)

To calculate the IPW estimator, we need to estimate the dropout probability from the data first. For ignorable missingness (MAR), we fit a logistic regression with the correctly specified model to predict the dropout probability. For nonignorable missingness (MNAR), we adopt both parametric (Miao et al., 2024) and semiparametric methods (Shao & Wang, 2016) for flexible dropout model estimation.

In the parametric estimation for the simulation study, we select $\boldsymbol{h}(S_t, A_t, Z_t) = [1, S_t^{(1)}, Z_t]^T$ for simplicity when learning $\psi$. Theoretically, it has been established (see Miao et al. (2024)) that any function of $(S_t, A_t, Z_t)$ can be chosen as the $\boldsymbol{h}$-function in the estimating equation while maintaining estimation consistency. Furthermore, Miao et al. (2024) demonstrated, by leveraging doubly robust estimation derived from the efficient influence function of $\psi$, that there exists a specific function $h^{\text{eff}}(S_t, A_t, Z_t)$ capable of minimizing the estimation variance. However, given that the proposed value estimator in this work is based on IPW and to simplify practical implementation, we recommend users choose $h$ to be as simple as possible in real applications. For instance, incorporating an intercept term, using the original form of the non-shadow variables rather than higher-order polynomial transformations, and including the shadow variable in the construction of $h$ to ensure the identifiability of $\psi$ can help reduce estimation variance and potentially improve the overall accuracy of the estimate.

In the semi-parametric estimation for the simulation study, the shadow variable $S_t^{(2)}$ is discretized into 4 bins based on the quartiles. The nonparametric part is approximated using Gaussian kernel with bandwidth $h_l = c \cdot \sigma_l n_l^{-1/3}$, where $\sigma_l$'s and $n_l$'s are the estimated standard deviation and the sample size for samples with $S_2 = l \in \{1, 2, 3, 4\}$. We pick $c = 7.5$ in the bandwidth formula based on an inspection of the objective function curve. The parametric part, i.e. $\psi$, is estimated using Equation (5) by setting $\boldsymbol{h}(S_t, A_t, Z_t) = [\mathbb{1}(Z_t = 1), \dots, \mathbb{1}(Z_t = 3)]^T$. In the minimization step of GMM, we use the

limited-memory BFGS algorithm (Liu & Nocedal, 1989) for both parametric and semi-parametric estimation with several initial values to avoid local minimum. In the semi-parametric estimation procedure, we recommend users construct the $h$-function using the discretized shadow variable, which has been empirically shown to perform well in our synthetic studies.

### E.2.4. IPW ESTIMATOR

After getting an estimate of the parameter $\widehat{\psi}_{nT}$ for the dropout model, we plug in the estimated probability to calculate $\widehat{\beta}_{\pi,\mathrm{IPW}}$, which is given in (3). To avoid extremely large inverse weight, we bound the missing propensity below at 0.01. After obtaining $\widehat{\beta}_{\pi,\mathrm{IPW}}$, we calculate the integrated value $\widehat{V}_{\mathrm{IPW}}^{\pi}(\mathbb{G})$ in a similar way to the CC estimator. Finally, we can construct the confidence interval for the proposed IPW estimator based on the theoretical form of the asymptotic variance, as provided in Equation (34). Note that when using a semi-parametric model for dropout estimation, the asymptotic variance of $\psi$ has a complex form (as discussed in Shao & Wang (2016)), making it difficult to compute due to the non-parametric kernel $g(\mathcal{U}_t)$. To simplify the computation, we suggest using an approximation of $\widehat{\sigma}_{\pi,\mathrm{IPW}}^2(\mathbb{G})$ given by (35). In all the empirical experiments presented in the main paper and appendices, the confidence intervals based on this approximation are very close to those obtained through bootstrapping and provide stable and satisfactory coverage. Therefore, we use this approximation in our implementation.

## E.3. Real Data Application

**Data description**   The sepsis dataset is extracted from the MIMIC-III v1.4 database (Johnson et al., 2016). We follow the data processing procedure described in Komorowski et al. (2018) and use a pure-python re-implementation available at https://github.com/microsoft/mimic_sepsis. Data is included from the diagnosis of sepsis and until 48 hours following the onset of sepsis to capture treatment management at the early phase. We exclude mortality cases within this time window and only focus on early discharged patients.

**Model features**   We consider a 14-dimensional state feature vector to represent important features clinicians would examine when deciding treatment and dosage for patients. The following physiological features are used in our model:

- Demographics: Age

- Lab values: Arterial pH, Chloride, Hemoglobin, INR-International Normalized Ratio, PT-Prothrombin Time, Arterial Blood Gas, Ionised Calcium, Calcium, Arterial Lactate

- Vital signs: SpO2, Temperature, Heart Rate

- Other: Sequential Organ Failure Assessment (SOFA) score

The features are aggregated over a time resolution of 4 hours, we carry the last value forward if no record is available in the current time window.

**Target policies**   In our experiments, we evaluate a fitted behavior policy and optimal policies learned via Deep Q-Network (Mnih et al., 2015), Dueling Double Deep Q-Network (Wang et al., 2016) and a discrete version of Batch-Constrained Deep Q-Learning (BCQ) (Fujimoto et al., 2019). The behavior policy is fitted with a random forest with 250 trees. For the other three types of Q-learning algorithms, we run for $2 \times 10^5$ iterations with minibatch size 256 and learning rate $1 \times 10^{-3}$.

**More details about implementation**   Similar to the simple environment, we first scale the state variables onto $[0, 1]$ and construct 4 cubic B-spline basis for each dimension. We do not use tensor product here due to high-dimensionality concern, so there are $L = 56$ bases in total. To fit the dropout model, we use the previous GCS score as a shadow variable, it is discretized into 4 bins based on quantiles. We consider the model in (6) with $\mathcal{U}_t = \{\mathbb{1}(S_t^{\mathrm{FiO2}} \leq 0.6), \mathbb{1}(60 \leq S_t^{\mathrm{HR}} \leq 100), \mathbb{1}(10 \leq S_t^{\mathrm{RR}} \leq 30)\}$, $Z_{t+1} = \mathbb{1}(S_{t+1}^{\mathrm{GCS}} \geq 14)$, and a bandwidth of $h_l = 10\sigma_l n_l^{-1/3}$. For large dataset, the kernel estimator used in this semiparametric method can be a bottleneck in computation. To accelerate the algorithm, we apply the downsample technique where we repeatedly sample random subsets from the whole dataset and aggregate the value estimation results by taking average.

# F. Generalizability of the Proposed Framework

In the main paper, we focus on the dropout scenario that occurs after the action is observed but before the reward and the next state are observed. Additionally, we use Least-Squares Temporal Difference Learning (LSTDQ) as the base OPE algorithm to investigate the effect of missing data. In this section, we will expand our scope to more general dropout patterns and other OPE methods.

## F.1. More General Dropout Patterns

The proposed framework is universally applicable to a broader class of dropout patterns. Specifically, the theoretical results for our IPW estimator are valid when dropout occurs after the observed action, regardless of whether the reward is observed or not. This is because the key idea behind the proposed IPW estimator is to assign weights to each transition based on the inverse probability of observing the complete transition quadruple $(S_t, A_t, R_{t+1}, S_{t+1})$ given observed $(S_t, A_t)$. On the other hand, when dropout occurs after an observed state but before an observed action, the proposed framework also applies. The distinction lies in that MAR and MNAR are now defined with respect to $S_t$ instead of $(S_t, A_t)$. If the missingness of the current action only depends on the current state and not on the action itself, it is considered ignorable. In such cases, the CC estimator remains valid, and no further adjustment is required. This can be seen from the decomposition $\mathbb{E}\{\eta_{t+1} M_t(\beta_\pi^*)\} = \mathbb{E}\{\mathbb{E}(\eta_{t+1} \mid S_t, \eta_t) \mathbb{E}(M_t(\beta_\pi^*) \mid S_t)\} = \mathbf{0}$. Here, $\mathbb{E}(M_t(\beta_\pi^*) \mid S_t) = \mathbf{0}$ follows from the law of total probability together with equation $\mathbb{E}\{R_{t+1} + \gamma V^\pi(S_{t+1}) - Q^\pi(S_t, A_t)|S_t, A_t\} = 0$. In the case of nonignorable missingness where the dropout is dependent on the potential action, the CC estimator can be biased, and the proposed IPW estimator can still be used to mitigate such bias.

Moreover, the idea of IPW adjustment can potentially be extended to handle intermittent missingness. The key distinction from monotone missingness lies in estimating the dropout propensity, which should be determined on a case-by-case basis and sometimes requires additional assumptions regarding the missing pattern. We leave this for future investigation.

## F.2. More General OPE Methods

In this section, we discuss how the proposed framework can be extended to Marginalized Importance Sampling-based (MIS) methods. We first introduce some additional notations and review the MIS value estimator.

Given the discount factor $\gamma$ and reference distribution of initial states $\mathbb{G}$, the discounted state-action visitation probability density for policy $\pi$ is defined as $d_\pi(s, a) = (1 - \gamma)\pi(a \mid s) \sum_{t=0}^\infty \gamma^t P(S_t = s \mid \pi)$. It satisfies the backward Bellman recursion

$$d_\pi(s', a') = (1 - \gamma)\mathbb{G}(s')\pi(a'|s') + \gamma \cdot \pi(a'|s') \int_{s \in \mathcal{S}} \sum_{a \in \mathcal{A}} d_\pi(s, a) p(s'|s, a) ds'. \tag{7}$$

With the notation of $d_\pi(s, a)$, the policy value can also be expressed as

$$V_\mathbb{G}^\pi = \frac{1}{1 - \gamma} \mathbb{E}_{(S_t, A_t) \sim d_\pi, R_{t+1} \sim r(S_t, A_t)} \{R_{t+1}\}.$$

In offline settings, the data may be collected from potentially different policies than the target policy $\pi$, denote the state-action visitation probability density under behavior policies as $d_\mathcal{D}$. To estimate the value using off-policy data, define the marginalized density ratio under the target policy $\pi$ as

$$\omega_\pi(s, a) := \frac{d_\pi(s, a)}{d_\mathcal{D}(s, a)}.$$

Then the policy value can be equivalently expressed as

$$V_\mathbb{G}^\pi = \frac{1}{1 - \gamma} \mathbb{E}_{(S_t, A_t) \sim d_\mathcal{D}, R_{t+1} \sim r(S_t, A_t)} \{\omega_\pi(S_t, A_t) \cdot R_{t+1}\},$$

which leads to the MIS value estimator (Liu et al., 2018; Xie et al., 2019). Compared with trajectory-based importance sampling methods, such a marginalized density ratio plays a critical role in breaking the curse of horizon. To handle unknown behavior policies, it is preferred to model the density ratio $\omega_\pi(s, a)$ directly, and plug in $\widehat{\omega}_\pi(s, a)$ to obtain the

final value estimate as follows,

$$\widehat{V}_{\mathbb{G}}^{\pi} = \frac{1}{1-\gamma} \frac{1}{nT} \sum_{i=1}^{n} \sum_{t=0}^{T-1} \widehat{\omega}_{\pi}(S_{i,t}, A_{i,t}) R_{i,t+1}.$$

The key for estimating $\omega_{\pi}(s, a)$ is by noting the following result derived from (7),

$$\mathbb{E}_{d_{\mathcal{D}}} \left\{ \omega_{\pi}(S_t, A_t) \left( f(S_t, A_t) - \gamma \mathbb{E}_{S_{t+1} \sim p(\cdot|S_t, A_t), a' \sim \pi(\cdot|S_{t+1})} [f(S_{t+1}, a')] \right) \right\}$$
$$= (1-\gamma) \mathbb{E}_{S_0 \sim \mathbb{G}, a \sim \pi(\cdot|S_0)} [f(S_0, a)].$$

A special case is to replace with $f(s, a)$ with $Q^{\pi}(s, a)$, leading to the following equation

$$\mathbb{E}_{d_{\mathcal{D}}} \left\{ \omega_{\pi}(S_t, A_t) \left( Q^{\pi}(S_t, A_t) - \gamma \mathbb{E}_{S_{t+1} \sim p(\cdot|S_t, A_t), a' \sim \pi(\cdot|S_{t+1})} [Q^{\pi}(S_{t+1}, a')] \right) \right\}$$
$$= (1-\gamma) \mathbb{E}_{S_0 \sim \mathbb{G}, a \sim \pi(\cdot|S_0)} [Q^{\pi}(S_0, a)]. \tag{8}$$

Another way to derive (8) is by noting the equivalence between two expressions of the policy value

$$\mathbb{E}_{d_{\mathcal{D}}} [\omega_{\pi}(S_t, A_t) \cdot r(S_t, A_t)] = (1-\gamma) \mathbb{E}_{S_0 \sim \mathbb{G}, a \sim \pi(\cdot|S_0)} [Q^{\pi}(S_0, a)],$$

and then replacing $r(S_t, A_t)$ with $Q^{\pi}(S_t, A_t) - \gamma \mathbb{E}_{S_{t+1} \sim p(\cdot|S_t, A_t), a' \sim \pi(\cdot|S_{t+1})} [Q^{\pi}(S_{t+1}, a')]$ using the Bellman equation.

Based on this equation, several methods have been developed to estimate the density ratio $\omega_{\pi}(s, a)$ (Nachum et al., 2019; Uehara et al., 2020). These methods typically learn $\omega_{\pi}(s, a)$ by minimizing the difference between the two sides of equation (8) within the chosen function classes for $Q^{\pi}(s, a)$ and $\omega_{\pi}(s, a)$. Denote the function class for $Q^{\pi}(s, a)$ as $\mathcal{Q}$ and the function class for $\omega_{\pi}(s, a)$ as $\Omega$. To illustrate the estimation process, we use Minimax Weight Learning (MWL) (Uehara et al., 2020) as an example, which estimates $\omega_{\pi}$ by solving $\widehat{\omega}_{\pi,nT}(s, a) = \underset{\omega_{\pi} \in \Omega}{\operatorname{argmin}} \underset{Q^{\pi} \in \mathcal{Q}}{\sup} \mathcal{L}_{nT}(\omega_{\pi}, Q^{\pi})^2$, where $\mathcal{L}_{nT}(\omega_{\pi}, Q^{\pi})$ is defined as follows

$$\mathcal{L}_{nT}(\omega_{\pi}, Q^{\pi}) = \frac{1}{nT} \sum_{i=1}^{n} \sum_{t=0}^{T-1} \eta_{i,t+1} \omega_{\pi}(S_{i,t}, A_{i,t}) \left( \gamma \sum_{a' \in \mathcal{A}} \pi(a'|S_{i,t+1}) Q^{\pi}(S_{i,t+1}, a') - Q^{\pi}(S_{i,t}, A_{i,t}) \right)$$
$$+ (1-\gamma) \cdot \mathbb{E}_{S_0 \sim \mathbb{G}} \left\{ \sum_{a \in \mathcal{A}} \pi(a|S_0) Q^{\pi}(S_0, a) \right\}. \tag{9}$$

The complete-case MIS value estimator can then be obtained by plugging in $\widehat{\omega}_{\pi,nT}(s, a)$ as follows,

$$\widehat{V}_{\text{CC}}^{\pi}(\mathbb{G}) = \frac{1}{1-\gamma} \frac{1}{nT} \sum_{i=1}^{n} \sum_{t=0}^{T-1} \eta_{i,t+1} \widehat{\omega}_{\pi,nT}(S_{i,t}, A_{i,t}) R_{i,t+1}. \tag{10}$$

Next, we present the consistency results under the two missingness mechanisms.

**Theorem F.1.** *Assume conditions 3.1-4.4 and A.1(a)(f) hold. Let $\omega_{\pi}(s, a)$ denote the true density ratio under missing data and $\widehat{\omega}_{\pi}(s, a)$ denote the estimated density ratio from the observed data. Further assume*

(a) *There exists a constant $c_{\omega} > 0$ such that $\sup_{s,a} |\omega_{\pi}(s, a)| \leq c_{\omega}$ and the function class $\Omega$ satisfies $\|\omega\|_{\infty} \leq c_{\omega}$ for all $\omega \in \Omega$.*

(b) *$\mathcal{L}_{nT}(\widehat{\omega}_{\pi}, Q^{\pi}) = o_p(1)$, where $Q^{\pi}$ represents the true Q-function.*

*Under ignorable missingness (MAR), the value estimate (10) remains consistent. On the other hand, if the missingness is nonignorable (MNAR), the value estimator (10) can be biased.*

In Assumptions (a), the boundedness of marginalized state-action density ratio $\omega_{\pi}$ can be ensured if the enumerator $d_{\pi}$ is bounded above and the denominator $d_{\mathcal{D}}$ is bounded away from 0. Such an assumption is commonly made in the literature related to importance sampling or inverse weighting. Additionally, the boundedness of function class $\Omega$ can be guaranteed

through a truncation argument. Assumption (b) states that $\widehat{\omega}_\pi$ ensures equation (8) approximately holds when substituting $f(s, a)$ with the true Q-function $Q^\pi(s, a)$. This assumption can be achieved when the function class $\mathcal{Q}$ captures the true Q-function, i.e., $Q^\pi \in \mathcal{Q}$, and the OPE algorithm minimizes $\sup_{Q^\pi \in \mathcal{Q}} \mathcal{L}_{nT}(\omega_\pi, Q^\pi)^2$ sufficiently close to 0.

The proof for Theorem F.1 can be found in Appendix G.4. It is noteworthy that the statement in Theorem F.1 can also be viewed from a special case of MWL, where $\omega_\pi(s, a)$ and $Q^\pi(s, a)$ are modeled with the same set of basis functions, i.e., $\omega_\pi(s, a) = \Phi_L(s)^\top \boldsymbol{\alpha}_{\pi,a}$ and $Q^\pi(s, a) = \Phi_L(s)^\top \boldsymbol{\beta}_{\pi,a}$. The corresponding value estimator can be shown to be

$$\widehat{V}_{\mathrm{CC}}^\pi(\mathbb{G}) =$$

$$\left\{ \int_s \boldsymbol{U}_\pi(s)\mathbb{G}(ds) \right\} \left\{ \frac{1}{nT} \sum_{i=1}^n \sum_{t=0}^{T-1} \eta_{i,t+1}\boldsymbol{\xi}_{i,t} \left( \boldsymbol{\xi}_{i,t} - \gamma \boldsymbol{U}_{\pi,i,t+1} \right)^\top \right\}^{-1} \left( \frac{1}{nT} \sum_{i=1}^n \sum_{t=0}^{T-1} \eta_{i,t+1}\boldsymbol{\xi}_{i,t} R_{i,t+1} \right).$$

which is identical to the complete-case (CC) estimator discussed in Section 4.2; the derivation is similar to Appendix A.3 of Uehara et al. (2020). Consequently, these two estimators share the same theoretical properties described in Theorem 4.5. In the case of nonignorable missingness, the IPW adjustment discussed in Section 4.2 can be applied to this estimator as well.

## G. Consistency and Asymptotic Results

In this section, we provide the proofs for Theorem 4.5, 4.6 and 4.7. For simplicity, we will omit the subscript $\pi$ in $\boldsymbol{\Sigma}_\pi$, $\widehat{\boldsymbol{\Sigma}}_\pi, \boldsymbol{\beta}_\pi, \widehat{\boldsymbol{\beta}}_\pi, \sigma_\pi, \widehat{\sigma}_\pi$. We first introduce the following lemmas from Shi et al. (2021b), the proofs can be found in Section E of their paper.

**Lemma G.1.** *Under Assumption A.1(a), there exists some constant $c' > 0$ such that $Q^\pi(s, a) \in \Lambda(p, c')$ for any policy $\pi$ and $a \in \mathcal{A}$.*

**Lemma G.2.** *Under Assumption A.1(b), there exists some constant $c^* \geq 1$ such that*

$$(c^*)^{-1} \leq \lambda_{\min} \left\{ \int_{s \in \mathcal{S}} \Phi_L(s)\Phi_L^\top(s)ds \right\} \leq \lambda_{\max} \left\{ \int_{s \in \mathcal{S}} \Phi_L(s)\Phi_L^\top(s)ds \right\} \leq c^*$$

*and $\sup_{s \in \mathcal{S}} \|\Phi_L(s)\|_2 \leq c^*\sqrt{L}$.*

**Lemma G.3.** *Suppose Assumption A.1 holds. Define $\boldsymbol{\Sigma} = \mathbb{E}\widehat{\boldsymbol{\Sigma}}$, we have $\|\boldsymbol{\Sigma}^{-1}\|_2 \leq 3\bar{c}^{-1}, \|\boldsymbol{\Sigma}\|_2 = O(1), \|\widehat{\boldsymbol{\Sigma}} - \boldsymbol{\Sigma}\|_2 = O_p\left\{ L^{1/2}(nT)^{-1/2} \log(nT) \right\}, \|\widehat{\boldsymbol{\Sigma}}^{-1} - \boldsymbol{\Sigma}^{-1}\|_2 = O_p\left\{ L^{1/2}(nT)^{-1/2} \log(nT) \right\}$ and $\|\widehat{\boldsymbol{\Sigma}}^{-1}\|_2 \leq 6\bar{c}^{-1}$ with probability approaching 1, as either $n \to \infty$ or $T \to \infty$.*

**Lemma G.4.** *Suppose Assumption A.1 holds. As either $n \to \infty$ or $T \to \infty$, we have $\lambda_{\max}(T^{-1} \sum_{t=0}^{T-1} \mathbb{E}\boldsymbol{\xi}_t\boldsymbol{\xi}_t^\top) = O_p(1), \lambda_{\max}\{(nT)^{-1} \sum_{i=1}^n \sum_{t=0}^{T-1} \boldsymbol{\xi}_{i,t}\boldsymbol{\xi}_{i,t}^\top\} = O_p(1), \lambda_{\min}(T^{-1} \sum_{t=0}^{T-1} \mathbb{E}\boldsymbol{\xi}_t\boldsymbol{\xi}_t^\top) \geq \bar{c}/2$ and $\lambda_{\min}\{(nT)^{-1} \sum_{i=1}^n \sum_{t=0}^{T-1} \boldsymbol{\xi}_{i,t}\boldsymbol{\xi}_{i,t}^\top\} \geq \bar{c}/3$ with probability approaching 1.*

**Lemma G.5.** *$\left\| \int_s \boldsymbol{U}(s)\mathbb{G}(ds) \right\|_2 \geq m^{-1/2} \left\| \int_s \Phi_L(s)\mathbb{G}(ds) \right\|_2$, where $m$ is the number of actions in the action space.*

**Lemma G.6.** *Define $\boldsymbol{\Sigma}^* = \int_{s \in \mathcal{S}} \sum_{a \in \mathcal{A}} \boldsymbol{\xi}(s, a)\{\boldsymbol{\xi}(s, a) - \gamma\boldsymbol{u}(s, a)\}^\top b(a \mid s)\mu(s)ds$. Suppose $T \to \infty$. Under the given conditions in Lemma G.3, we have $\|\boldsymbol{\Sigma} - \boldsymbol{\Sigma}^*\|_2 \preceq T^{-1/2}$.*

*Remark G.7.* The notation $a_n \preceq b_n$ means that there exists some constant $C > 0$ such that $a_n \leq C \cdot b_n$ for any $n$. The notation $a_n \preceq 1$ means $a_n = O_p(1)$.

Next, we will go through the proof for the consistency and asymptotic result for the proposed IPW estimator. The big idea is similar to the proof of Theorem 1 in Shi et al. (2021b) but with additional components to handle inverse weights and associated uncertainty.

### G.1. Proof of Theorem 4.5

**Theorem** Suppose Assumption A.1 holds. $\widehat{V}_{\mathrm{CC}}^\pi(\mathbb{G})$ is a consistent estimator if the missing mechanism is ignorable (MAR). However, if the missing mechanism is nonignorable (MNAR), $\widehat{V}_{\mathrm{CC}}^\pi(\mathbb{G})$ can be biased.

*Proof.* We first provide a sketch of the big idea. Assume the true Q-function is $Q^\pi(s, a) = \Phi_L^\top(s)\boldsymbol{\beta}_\pi^*$ and the true parameter $\boldsymbol{\beta}^*$ satisfies $\mathbb{E}\{\boldsymbol{M}_t(\boldsymbol{\beta}_\pi^*)\} = \boldsymbol{0}$, where $\boldsymbol{M}_t(\boldsymbol{\beta}_\pi) = \boldsymbol{\xi}_t\{R_{t+1} - (\boldsymbol{\xi}_t - \gamma\boldsymbol{U}_{\pi,t+1})^\top\boldsymbol{\beta}_\pi\}$. Under incomplete data, the equation

becomes $\mathbb{E}\{\eta_{t+1}\boldsymbol{M}_t(\boldsymbol{\beta}_\pi)\} = \boldsymbol{0}$. Using the condition of MAR (Definition 4.1), we apply the conditional independence between $\eta_{t+1}$ and $(S_{t+1}, R_{t+1})$ to separate the $\eta_{t+1}$ and $\boldsymbol{M}_t$ term as follows

$$
\begin{aligned}
\mathbb{E}\{\eta_{t+1}\boldsymbol{M}_t(\boldsymbol{\beta}_\pi)\} &= \mathbb{E}\{\mathbb{E}(\eta_{t+1}\boldsymbol{M}_t(\boldsymbol{\beta}_\pi) \mid S_t, A_t, \eta_t)\} \\
&= \mathbb{E}\{\mathbb{E}(\eta_{t+1} \mid S_t, A_t, \eta_t)\,\mathbb{E}(\boldsymbol{M}_t(\boldsymbol{\beta}_\pi) \mid S_t, A_t)\}.
\end{aligned}
$$

It follows from $\mathbb{E}\{R_{t+1} + \gamma \sum_{a'\in\mathcal{A}} Q^\pi(S_{t+1}, a')\pi(a'|S_{t+1}) - Q^\pi(S_t, A_t)|S_t, A_t\} = 0$ that $\mathbb{E}\{\boldsymbol{M}_t(\boldsymbol{\beta}_\pi^*)|S_t, A_t\} = \boldsymbol{0}$. Therefore, $\mathbb{E}\{\eta_{t+1}\boldsymbol{M}_t(\boldsymbol{\beta}_\pi^*)\} = \boldsymbol{0}$, then $\boldsymbol{\beta}_\pi^*$ is still the solution to $\mathbb{E}\{\eta_{t+1}\boldsymbol{M}_t(\boldsymbol{\beta}_\pi)\} = \boldsymbol{0}$. As a result, the corresponding value estimator is still unbiased under some regularity conditions. However, for nonignorable missingness (MNAR), $\mathbb{E}\{\eta_{t+1}\boldsymbol{M}_t(\boldsymbol{\beta}_\pi^*)\} = \boldsymbol{0}$ no longer holds because $\eta_{t+1}$ and $\boldsymbol{M}_t$ cannot be separated using the conditional independence. Thus the complete-case estimator $\widehat{\boldsymbol{\beta}}_{\pi,\mathrm{CC}}$ will be biased from $\boldsymbol{\beta}_\pi^*$ unless the probability $P(\eta_{t+1} = 1 \mid S_t, A_t, R_{t+1}, S_{t+1}, \eta_t)$ is a constant.

Next, we provide a more rigorous proof that takes into account the approximation error.

By Condition A.1(a)(b)(e), the number of basis $L$ for the Q-function satisfies $L^{2p/d} \gg nT\{1 + \|\int_s \Phi_L(s)\mathbb{G}(ds)\|_2^{-2}\}$, it follows from Lemma G.5 that $L^{2p/d} \gg nT\{1 + \|\int_s \boldsymbol{U}(s)\mathbb{G}(ds)\|_2^{-2}\}$. By Lemma G.1 and Condition A.1(b), there exist a set of vectors $\{\beta_a^*\}$ that satisfy

$$
\sup_{s\in\mathcal{S}, a\in\mathcal{A}} \left| Q^\pi(s, a) - \Phi_L^\top(s)\beta_a^* \right| \le CL^{-p/d}, \tag{11}
$$

for some constant $C > 0$ (Huang et al., 1998). Let $\boldsymbol{\beta}^* = (\beta_1^{*\top}, \ldots, \beta_m^{*\top})^\top$, define

$$
\begin{aligned}
r_{i,t} &= \gamma \sum_{a\in\mathcal{A}} \left\{ \Phi_L^\top(S_{i,t+1})\beta_a^* - Q^\pi(S_{i,t+1}, a) \right\} \pi(a|S_{i,t+1}) - \left\{ \Phi_L^\top(S_{i,t})\beta_{A_{i,t}}^* - Q^\pi(S_{i,t}, A_{i,t}) \right\}, \\
\varepsilon_{i,t} &= R_{i,t+1} + \gamma \sum_{a\in\mathcal{A}} Q^\pi(S_{i,t+1}, a)\pi(a|S_{i,t+1}) - Q^\pi(S_{i,t}, A_{i,t}).
\end{aligned} \tag{12}
$$

The condition $P(\max_t |R_t| \le c_0) = 1$ in Assumption A.1(f) implies that $R_{i,t} \le c_0, \forall i, t$, almost surely. By Lemma G.1, we have $|Q^\pi(s, a)| \le c'$ for any $\pi, s, a$. Therefore, the error term $\varepsilon_{i,t}$ can be bounded as follows

$$
\max_{0\le t < T, 1\le i\le n} |\varepsilon_{i,t}| \le c_0 + (\gamma + 1)c' \le c_0 + 2c', \text{ almost surely.} \tag{13}
$$

In addition, it follows from (11) that

$$
\max_{0\le t < T, 1\le i\le n} |r_{i,t}| \le 2 \sup_{s\in\mathcal{S}, a\in\mathcal{A}} \left| Q^\pi(s, a) - \Phi_L^\top(s)\beta_a^* \right| \le 2CL^{-p/d}. \tag{14}
$$

For incomplete data, we can only leverage the observed samples for inference. With the response indicator $\eta_t$ defined in Section 4, the estimating equations can be written as $\mathbb{E}\{\boldsymbol{M}_t(\boldsymbol{\beta}^*)|\eta_{t+1} = 1\} = \boldsymbol{0}$, or equivalently, $\mathbb{E}\{\eta_{t+1}\boldsymbol{M}_t(\boldsymbol{\beta}^*)\} = \boldsymbol{0}$. The estimator for $\boldsymbol{\beta}^*$ is given by

$$
\widehat{\boldsymbol{\beta}}_{\mathrm{CC}} = \underbrace{\left\{ \frac{1}{nT} \sum_{i=1}^n \sum_{t=0}^{T-1} \eta_{i,t+1} \boldsymbol{\xi}_{i,t} \left( \boldsymbol{\xi}_{i,t} - \gamma \boldsymbol{U}_{i,t+1} \right)^\top \right\}^{-1}}_{\widehat{\boldsymbol{\Sigma}}_{\mathrm{CC}}} \left( \frac{1}{nT} \sum_{i=1}^n \sum_{t=0}^{T-1} \eta_{i,t+1} \boldsymbol{\xi}_{i,t} R_{i,t+1} \right).
$$

Let $\boldsymbol{\Sigma}_{\mathrm{CC}} = \mathbb{E}\widehat{\boldsymbol{\Sigma}}_{\mathrm{CC}}$. By definition,

$$
\begin{aligned}
\widehat{\boldsymbol{\beta}}_{\mathrm{CC}} - \boldsymbol{\beta}^* &= \widehat{\boldsymbol{\Sigma}}_{\mathrm{CC}}^{-1} \left[ \frac{1}{nT} \sum_{i=1}^{n} \sum_{t=0}^{T-1} \eta_{i,t+1} \boldsymbol{\xi}_{i,t} \left\{ R_{i,t+1} - \left( \boldsymbol{\xi}_{i,t} - \gamma \boldsymbol{U}_{i,t+1} \right)^{\top} \boldsymbol{\beta}^* \right\} \right] \\
&= \widehat{\boldsymbol{\Sigma}}_{\mathrm{CC}}^{-1} \Bigg[ \frac{1}{nT} \sum_{i=1}^{n} \sum_{t=0}^{T-1} \eta_{i,t+1} \boldsymbol{\xi}_{i,t} \times \\
&\qquad \left\{ R_{i,t+1} - \Phi_L^{\top} \left( S_{i,t} \right) \beta_{A_{i,t}}^* + \gamma \sum_{a \in \mathcal{A}} \Phi_L^{\top} \left( S_{i,t+1} \right) \beta_a^* \pi \left( a \mid S_{i,t+1} \right) \right\} \Bigg] \\
&= \widehat{\boldsymbol{\Sigma}}_{\mathrm{CC}}^{-1} \left\{ \frac{1}{nT} \sum_{i=1}^{n} \sum_{t=0}^{T-1} \eta_{i,t+1} \boldsymbol{\xi}_{i,t} \left( \varepsilon_{i,t} + r_{i,t} \right) \right\} \\
&= \underbrace{\boldsymbol{\Sigma}_{\mathrm{CC}}^{-1} \left( \frac{1}{nT} \sum_{i=1}^{n} \sum_{t=0}^{T-1} \eta_{i,t+1} \boldsymbol{\xi}_{i,t} \varepsilon_{i,t} \right)}_{\zeta_1} + \underbrace{\left( \widehat{\boldsymbol{\Sigma}}_{\mathrm{CC}}^{-1} - \boldsymbol{\Sigma}_{\mathrm{CC}}^{-1} \right) \left( \frac{1}{nT} \sum_{i=1}^{n} \sum_{t=0}^{T-1} \eta_{i,t+1} \boldsymbol{\xi}_{i,t} \varepsilon_{i,t} \right)}_{\zeta_2} \\
&\quad + \underbrace{\widehat{\boldsymbol{\Sigma}}_{\mathrm{CC}}^{-1} \left( \frac{1}{nT} \sum_{i=1}^{n} \sum_{t=0}^{T-1} \eta_{i,t+1} \boldsymbol{\xi}_{i,t} r_{i,t} \right)}_{\zeta_3} .
\end{aligned}
$$

It suffices to derive the error bounds for $\|\zeta_1\|_2$, $\|\zeta_2\|_2$, and $\|\zeta_3\|_2$.

**Error bound for $\|\zeta_3\|_2$.** For any $\boldsymbol{a} \in \mathbb{R}^{mL}$,

$$
\begin{aligned}
\left| \boldsymbol{a}^{\top} \left( \frac{1}{nT} \sum_{i=1}^{n} \sum_{t=0}^{T-1} \eta_{i,t+1} \boldsymbol{\xi}_{i,t} r_{i,t} \right) \right| &\leq \frac{1}{nT} \sum_{i=1}^{n} \sum_{t=0}^{T-1} \left| \boldsymbol{a}^{\top} \boldsymbol{\xi}_{i,t} \right| \left| r_{i,t} \eta_{i,t+1} \right| \leq \max_{i,t} \left| r_{i,t} \right| \left( \frac{1}{nT} \sum_{i=1}^{n} \sum_{t=0}^{T-1} \left| \boldsymbol{a}^{\top} \boldsymbol{\xi}_{i,t} \right| \right) \\
&\leq 2CL^{-p/d} \left( \frac{1}{nT} \sum_{i=1}^{n} \sum_{t=0}^{T-1} \left| \boldsymbol{a}^{\top} \boldsymbol{\xi}_{i,t} \right| \right) \leq 2CL^{-p/d} \left( \frac{1}{nT} \sum_{i=1}^{n} \sum_{t=0}^{T-1} \boldsymbol{a}^{\top} \boldsymbol{\xi}_{i,t} \boldsymbol{\xi}_{i,t}^{\top} \boldsymbol{a} \right)^{1/2} .
\end{aligned} \tag{15}
$$

The second inequality uses the bound of binary $\eta_t$ that $|\eta_t| \leq 1$, the third inequality follows from (14), and the fourth inequality applies the Cauchy-Schwarz inequality. Then we obtain

$$
\left\| \frac{1}{nT} \sum_{i=1}^{n} \sum_{t=0}^{T-1} \eta_{i,t+1} \boldsymbol{\xi}_{i,t} r_{i,t} \right\|_2 \leq 2CL^{-p/d} \lambda_{\max}^{1/2} \left( \frac{1}{nT} \sum_{i=1}^{n} \sum_{t=0}^{T-1} \boldsymbol{\xi}_{i,t} \boldsymbol{\xi}_{i,t}^{\top} \right) .
$$

By Lemma G.3 and Lemma G.4, we have

$$
\|\zeta_3\|_2 \leq \left\| \widehat{\boldsymbol{\Sigma}}_{\mathrm{CC}}^{-1} \right\|_2 \left\| \frac{1}{nT} \sum_{i=1}^{n} \sum_{t=0}^{T-1} \eta_{i,t+1} \boldsymbol{\xi}_{i,t} r_{i,t} \right\|_2 = O_p(1) O_p \left( L^{-p/d} \right) = O_p \left( L^{-p/d} \right), \tag{16}
$$

which indicates that $\zeta_3$ is driven by the approximation error of the Q-function, and can be controlled by increasing the number of basis functions.

The main difference between ignorable missingness (MAR) and nonignorable missingness (MNAR) lies in $(\zeta_1 + \zeta_2)$. In the following steps, we will show that the complete-case value estimator $\widehat{V}_{\mathrm{CC}}^{\pi}(\mathbb{G})$ is still consistent under ignorable missingness (MAR) but becomes biased under nonignorable missingness (MNAR).

- **MAR**

To show $\|\widehat{\boldsymbol{\beta}}_{\mathrm{CC}} - \boldsymbol{\beta}^*\|_2 = o_p(1)$, we need to derive the error bound for $\|\zeta_1\|_2$, $\|\zeta_2\|_2$ and show they are $o_p(1)$.

**Error bound for $\|\zeta_2\|_2$.** We first derive the error bound for $\|\zeta_2\|_2$. By Markov Assumption, Conditional Mean Independence Assumption and Bellman equation, $\mathbb{E}\left(\varepsilon_t | \mathcal{F}_t\right) = \mathbb{E}\left(\varepsilon_t | S_t, A_t\right) = 0$, where $\mathcal{F}_t = \{(S_j, A_j, R_{j+1})\}_{0 \leq j < t} \cup \{S_t, A_t\}$ denotes the past information up to time $t$. Together with the conditional independence of $\eta_{t+1}$ and $\varepsilon_t$ based on the definition of MAR, we have $\mathbb{E}\{\eta_{t+1}\boldsymbol{\xi}_t\varepsilon_t\} = \mathbb{E}\left\{\mathbb{E}\left(\eta_{t+1}\boldsymbol{\xi}_t\varepsilon_t | \mathcal{F}_t, \eta_t\right)\right\} = \mathbb{E}\left\{\boldsymbol{\xi}_t \mathbb{E}\left(\eta_{t+1} | \mathcal{F}_t, \eta_t\right) \mathbb{E}\left(\varepsilon_t | \mathcal{F}_t\right)\right\} = \mathbf{0}$. Similarly, for any $0 \leq t_1 < t_2 < T$, we obtain $\mathbb{E}\{\eta_{t_1+1}\eta_{t_2+1}\varepsilon_{t_1}\varepsilon_{t_2}\boldsymbol{\xi}_{t_1}^{\top}\boldsymbol{\xi}_{t_2}\} = 0$. In addition, by the independence assumption among trajectories, we have $\mathbb{E}\{\eta_{i_1,t_1+1}\eta_{i_2,t_2+1}\varepsilon_{i_1,t_1}\varepsilon_{i_2,t_2}\boldsymbol{\xi}_{i_1,t_1}^{\top}\boldsymbol{\xi}_{i_2,t_2}\} = 0$. It follows that $\mathbb{E}\|\sum_{i=1}^{n}\sum_{t=0}^{T-1}\eta_{i,t+1}\boldsymbol{\xi}_{i,t}\varepsilon_{i,t}\|_2^2 = \sum_{i=1}^{n}\sum_{t=0}^{T-1}\mathbb{E}\left\{\eta_{i,t+1}^2\varepsilon_{i,t}^2\boldsymbol{\xi}_{i,t}^{\top}\boldsymbol{\xi}_{i,t}\right\} = n\sum_{t=0}^{T-1}\mathbb{E}\left\{\eta_{t+1}^2\varepsilon_t^2\boldsymbol{\xi}_t^{\top}\boldsymbol{\xi}_t\right\}$. Together with (13) and Lemma G.2, we obtain

$$\mathbb{E}\left\|\sum_{i=1}^{n}\sum_{t=0}^{T-1}\eta_{i,t+1}\boldsymbol{\xi}_{i,t}\varepsilon_{i,t}\right\|_2^2 \leq \left(c_0 + 2c'\right)^2 n\sum_{t=0}^{T-1}\mathbb{E}\boldsymbol{\xi}_t^{\top}\boldsymbol{\xi}_t \leq \left(c_0 + 2c'\right)^2 nT\sup_{s \in \mathcal{S}}\|\Phi_L(s)\|_2^2 \preceq nTL.$$

By Markov inequality,

$$\frac{1}{nT}\sum_{i=1}^{n}\sum_{t=0}^{T-1}\eta_{i,t+1}\boldsymbol{\xi}_{i,t}\varepsilon_{i,t} = O_p\{\sqrt{L/(nT)}\}.$$

Combine with Lemma G.3 yields

$$\zeta_2 = O_p\{\sqrt{L/nT}\log(nT)\}O_p\{\sqrt{L/(nT)}\} = O_p\left\{L(nT)^{-1}\log(nT)\right\}. \tag{17}$$

**Error bound for $\|\zeta_1\|_2$.** Using similar arguments as bounding $\|\zeta_2\|_2$, we obtain

$$\zeta_1 = O_p\left\{L^{1/2}(nT)^{-1/2}\right\}. \tag{18}$$

Combining (16), (17), and (18), we have

$$\widehat{\boldsymbol{\beta}}_{\mathrm{CC}} - \boldsymbol{\beta}^* = O_p\left\{L^{1/2}(nT)^{-1/2}\right\} + O_p\left\{L(nT)^{-1}\log(nT)\right\} + O_p\left\{L^{-p/d}\right\}.$$

It follows from Condition A.1(e) that

$$\|\widehat{\boldsymbol{\beta}}_{\mathrm{CC}} - \boldsymbol{\beta}^*\|_2 = O_p\left(L^{-p/d}\right) + O_p\left\{L^{1/2}(nT)^{-1/2}\right\} = o_p(1).$$

Recall that $\widehat{V}_{\mathrm{CC}}^{\pi}(\mathbb{G}) = \left\{\int_s \boldsymbol{U}(s)\mathbb{G}(ds)\right\}^{\top}\widehat{\boldsymbol{\beta}}_{\mathrm{CC}}$, thus,

$$\left|\widehat{V}_{\mathrm{CC}}^{\pi}(\mathbb{G}) - V^{\pi}(\mathbb{G})\right| \leq \left\|\int_s \boldsymbol{U}(s)\mathbb{G}(ds)\right\|_2 \left\|\widehat{\boldsymbol{\beta}}_{\mathrm{CC}} - \boldsymbol{\beta}^*\right\|_2 = o_p(1), \tag{19}$$

that is, $\widehat{V}_{\mathrm{CC}}^{\pi}(\mathbb{G}) \xrightarrow{p} V^{\pi}(\mathbb{G})$ as $nT \to \infty$. Therefore, the value estimator $\widehat{V}_{\mathrm{CC}}^{\pi}(\mathbb{G})$ is still consistent when the dropout mechanism is MAR.

- **MNAR**

**Error bounds for $\|\zeta_1 + \zeta_2\|_2$.** Under nonignorable missingness, the conditional independence of $\eta_{t+1}$ and $\varepsilon_t$ no longer holds, so

$$\mathbb{E}\left\{\eta_{t+1}\boldsymbol{\xi}_t\varepsilon_t\right\} = \mathbb{E}\left\{\mathbb{E}\left\{\eta_{t+1}\boldsymbol{\xi}_t\varepsilon_t \mid \mathcal{F}_t, S_{t+1}, R_{t+1}, \eta_t\right\}\right\} = \mathbb{E}\left\{\boldsymbol{\xi}_t\varepsilon_t\mathbb{E}\left\{\eta_{t+1} \mid \mathcal{F}_t, S_{t+1}, R_{t+1}, \eta_t\right\}\right\}$$
$$= \mathbb{E}\left\{\boldsymbol{\xi}_t\varepsilon_t\eta_t(1 - \lambda(S_t, A_t, S_{t+1}, R_{t+1}))\right\} \neq \mathbf{0}.$$

We cannot bound the term $\frac{1}{nT}\sum_{i=1}^{n}\sum_{t=0}^{T-1}\eta_{i,t+1}\boldsymbol{\xi}_{i,t}\varepsilon_{i,t}$ as in the MAR case. As a result, $\|\zeta_1 + \zeta_2\| = o_p(1)$ may no longer holds. Therefore, $\widehat{V}^{\pi}(\mathbb{G})$ can be biased when the dropout mechanism is MNAR. $\qquad\square$

## G.2. Proof of Theorem 4.6

**Theorem** Assume conditions A.1 and A.2 hold. $\widehat{V}_{\text{IPW}}^{\pi}(\mathbb{G})$ is a consistent estimator for value, i.e., $\widehat{V}_{\text{IPW}}^{\pi}(\mathbb{G}) \xrightarrow{p} V^{\pi}(\mathbb{G})$ as either $n \to \infty$ or $T \to \infty$.

*Proof.* The steps in this proof will be very similar to the proof of Theorem 4.5, but now we incorporate the inverse weights. For simplicity, we use the notation

$$\omega_{i,t+1}(\boldsymbol{\psi}) := \eta_{i,t+1}\{1 - \lambda(S_{i,t}, A_{i,t}, R_{i,t+1}, S_{i,t+1}; \boldsymbol{\psi})\}^{-1}$$

to represent the weighting term, where $\boldsymbol{\psi} \in \mathbb{R}^k$ is the parameter of the dropout propensity model. Under assumption A.2(a) on the correct specification of the dropout propensity, there exists some $\boldsymbol{\psi}^*$ such that $\lambda(S_t, A_t, R_{t+1}, S_{t+1}) = \lambda(S_t, A_t, R_{t+1}, S_{t+1}; \boldsymbol{\psi}^*)$. It follows from the definition of dropout propensity that $\mathbb{E}(\eta_{t+1}|\mathcal{F}_t, R_{t+1}, S_{t+1}, \eta_t = 1) = 1 - \lambda(S_t, A_t, R_{t+1}, S_{t+1}; \boldsymbol{\psi}^*)$, therefore, $\mathbb{E}\{\omega_{t+1}(\boldsymbol{\psi}^*)|\mathcal{F}_t, R_{t+1}, S_{t+1}, \eta_t = 1\} = 1$.

Similar to the previous proof, $\widehat{\boldsymbol{\beta}}_{\text{IPW}} - \boldsymbol{\beta}^*$ can be decomposed as

$$\widehat{\boldsymbol{\beta}}_{\text{IPW}} - \boldsymbol{\beta}^* = \boldsymbol{\Sigma}^{-1} \underbrace{\left( \frac{1}{nT} \sum_{i=1}^{n} \sum_{t=0}^{T-1} \omega_{i,t+1}(\widehat{\boldsymbol{\psi}})\boldsymbol{\xi}_{i,t}\varepsilon_{i,t} \right)}_{\zeta_1}$$

$$+ \underbrace{\left( \widehat{\boldsymbol{\Sigma}}_{\text{IPW}}^{-1} - \boldsymbol{\Sigma}^{-1} \right) \left( \frac{1}{nT} \sum_{i=1}^{n} \sum_{t=0}^{T-1} \omega_{i,t+1}(\widehat{\boldsymbol{\psi}})\boldsymbol{\xi}_{i,t}\varepsilon_{i,t} \right)}_{\zeta_2}$$

$$+ \underbrace{\widehat{\boldsymbol{\Sigma}}_{\text{IPW}}^{-1} \left( \frac{1}{nT} \sum_{i=1}^{n} \sum_{t=0}^{T-1} \omega_{i,t+1}(\widehat{\boldsymbol{\psi}})\boldsymbol{\xi}_{i,t}r_{i,t} \right)}_{\zeta_3}.$$

To prove its consistency, it suffices to show $\|\zeta_1\|_2$, $\|\zeta_2\|_2$, and $\|\zeta_3\|_2$ are $o_p(1)$. To bound these terms, we first introduce the following lemma. This lemma is similar to Lemma G.3, but it is with respect to $\widehat{\boldsymbol{\Sigma}}_{\text{IPW}}$ instead of $\widehat{\boldsymbol{\Sigma}}$. The proof can be found in Appendix G.5.

**Lemma G.8.** *Suppose Assumption A.1-A.2 holds. We have* $\|\widehat{\boldsymbol{\Sigma}}_{IPW} - \boldsymbol{\Sigma}\|_2 = O_p\left\{L^{1/2}(nT)^{-1/2}\log(nT)\right\}$, $\|\widehat{\boldsymbol{\Sigma}}_{IPW}^{-1} - \boldsymbol{\Sigma}^{-1}\|_2 = O_p\left\{L^{1/2}(nT)^{-1/2}\log(nT)\right\}$ *and* $\|\widehat{\boldsymbol{\Sigma}}_{IPW}^{-1}\|_2 \leq 6\bar{c}^{-1}$ *with probability approaching 1, as either* $n \to \infty$ *or* $T \to \infty$.

Next, we will use Lemma G.8 to derive the error bounds for $\|\zeta_1\|_2$, $\|\zeta_2\|_2$, and $\|\zeta_3\|_2$.

**Error bound for $\|\zeta_3\|_2$.** It follows from condition A.2(b) that $\max_{1 \leq i \leq n, 0 \leq t < T} |\omega_{i,t}(\boldsymbol{\psi})| \leq c_{\lambda}^{-1}$. Using similar arguments in (15), we have

$$\left| \boldsymbol{a}^{\top} \left( \frac{1}{nT} \sum_{i=1}^{n} \sum_{t=0}^{T-1} \omega_{i,t+1}(\widehat{\boldsymbol{\psi}})\boldsymbol{\xi}_{i,t}r_{i,t} \right) \right| \leq \frac{2CL^{-p/d}}{c_{\lambda}} \left( \frac{1}{nT} \sum_{i=1}^{n} \sum_{t=0}^{T-1} \boldsymbol{a}^{\top}\boldsymbol{\xi}_{i,t}\boldsymbol{\xi}_{i,t}^{\top}\boldsymbol{a} \right)^{1/2}$$

for any $\boldsymbol{a} \in \mathbb{R}^{mL}$. Thus,

$$\left\| \frac{1}{nT} \sum_{i=1}^{n} \sum_{t=0}^{T-1} \omega_{i,t+1}(\widehat{\boldsymbol{\psi}})\boldsymbol{\xi}_{i,t}r_{i,t} \right\|_2 \leq \frac{2CL^{-p/d}}{c_{\lambda}}\lambda_{\max}^{1/2} \left( \frac{1}{nT} \sum_{i=1}^{n} \sum_{t=0}^{T-1} \boldsymbol{\xi}_{i,t}\boldsymbol{\xi}_{i,t}^{\top} \right) = O_p(L^{-p/d}).$$

By Lemma G.8, we obtain

$$\zeta_3 = O_p\left( L^{-p/d} \right). \tag{20}$$

**Error bounds for** $\|\zeta_2\|_2$**.** The RHS of $\zeta_1$ and $\zeta_2$ can be decomposed as follows

$$\frac{1}{nT} \sum_{i=1}^{n} \sum_{t=0}^{T-1} \omega_{i,t+1}(\widehat{\psi}) \boldsymbol{\xi}_{i,t} \varepsilon_{i,t} = \frac{1}{nT} \sum_{i=1}^{n} \sum_{t=0}^{T-1} \omega_{i,t+1}(\boldsymbol{\psi}^*) \boldsymbol{\xi}_{i,t} \varepsilon_{i,t}$$
$$+ \frac{1}{nT} \sum_{i=1}^{n} \sum_{t=0}^{T-1} \left( \omega_{i,t+1}(\widehat{\psi}) - \omega_{i,t+1}(\boldsymbol{\psi}^*) \right) \boldsymbol{\xi}_{i,t} \varepsilon_{i,t}. \tag{21}$$

We first show that

$$\frac{1}{nT} \sum_{i=1}^{n} \sum_{t=0}^{T-1} \omega_{i,t+1}(\boldsymbol{\psi}^*) \boldsymbol{\xi}_{i,t} \varepsilon_{i,t} = O_p\{\sqrt{L/(nT)}\}.$$

By the property of conditional expectation, we have

$$\mathbb{E}\left\{ \omega_{t+1}(\boldsymbol{\psi}^*) \boldsymbol{\xi}_t \varepsilon_t \right\} = \mathbb{E}\left\{ \mathbb{E}\left( \omega_{t+1}(\boldsymbol{\psi}^*) \mid \mathcal{F}_t, R_{t+1}, S_{t+1}, \eta_t \right) \boldsymbol{\xi}_t \varepsilon_t \right\} = \mathbb{E}\left\{ \eta_t \boldsymbol{\xi}_t \varepsilon_t \right\} = \mathbf{0}. \tag{22}$$

Using similar arguments in deriving $\zeta_2$'s error bound in Proof G.1, we show

$$\mathbb{E} \left\| \sum_{i=1}^{n} \sum_{t=0}^{T-1} \omega_{i,t+1}(\boldsymbol{\psi}^*) \boldsymbol{\xi}_{i,t} \varepsilon_{i,t} \right\|_2^2 = n \sum_{t=0}^{T-1} \mathbb{E}\left\{ \omega_{t+1}^2(\boldsymbol{\psi}^*) \varepsilon_t^2 \boldsymbol{\xi}_t^\top \boldsymbol{\xi}_t \right\}$$
$$\leq \frac{(c_0 + 2c')^2}{c_\lambda^2} n \sum_{t=0}^{T-1} \mathbb{E} \boldsymbol{\xi}_t^\top \boldsymbol{\xi}_t \leq \frac{(c_0 + 2c')^2}{c_\lambda^2} nT \sup_{s \in \mathcal{S}} \|\Phi_L(s)\|_2^2 \preceq nTL.$$

Then by the Markov inequality,

$$\frac{1}{nT} \sum_{i=1}^{n} \sum_{t=0}^{T-1} \omega_{i,t+1}(\boldsymbol{\psi}^*) \boldsymbol{\xi}_{i,t} \varepsilon_{i,t} = O_p\{\sqrt{L/(nT)}\}. \tag{23}$$

Next, we show that

$$\frac{1}{nT} \sum_{i=1}^{n} \sum_{t=0}^{T-1} \left( \omega_{i,t+1}(\widehat{\psi}) - \omega_{i,t+1}(\boldsymbol{\psi}^*) \right) \boldsymbol{\xi}_{i,t} \varepsilon_{i,t} = O_p\{(nT)^{-1/2}\}.$$

A mean value expansion of $\frac{1}{nT} \sum_{i=1}^{n} \sum_{t=0}^{T-1} \omega_{i,t+1}(\widehat{\psi})$ around $\boldsymbol{\psi}^*$ yields

$$\frac{1}{nT} \sum_{i=1}^{n} \sum_{t=0}^{T-1} \omega_{i,t+1}(\widehat{\psi}) \boldsymbol{\xi}_{i,t} \varepsilon_{i,t} = \frac{1}{nT} \sum_{i=1}^{n} \sum_{t=0}^{T-1} \omega_{i,t+1}(\boldsymbol{\psi}^*) \boldsymbol{\xi}_{i,t} \varepsilon_{i,t} + \boldsymbol{H}_1 (\widehat{\psi} - \boldsymbol{\psi}^*) + o_p(\|\widehat{\psi} - \boldsymbol{\psi}^*\|), \tag{24}$$

where $\boldsymbol{H}_1 = \mathbb{E}\left\{ \boldsymbol{\xi}_t \varepsilon_t \nabla_\psi \omega_{t+1}(\boldsymbol{\psi}^*)^\top \right\} \in \mathbb{R}^{m \times q}$. Similarly, we obtain

$$\frac{1}{nT} \sum_{i=1}^{n} \sum_{t=0}^{T-1} \omega_{i,t+1}(\widehat{\psi}) = \frac{1}{nT} \sum_{i=1}^{n} \sum_{t=0}^{T-1} \omega_{i,t+1}(\boldsymbol{\psi}^*) + \mathbb{E}\left\{ \nabla_\psi \omega_{t+1}(\boldsymbol{\psi}^*)^\top \right\} (\widehat{\psi} - \boldsymbol{\psi}^*) + o_p(\|\widehat{\psi} - \boldsymbol{\psi}^*\|).$$

Recall that in Equation (4), $\mathbb{E}[\boldsymbol{m}_t(\boldsymbol{\psi}^*)] = 0$. A mean value expansion of $\frac{1}{nT} \sum_{i=1}^{n} \sum_{t=0}^{T-1} \boldsymbol{m}_{i,t}(\widehat{\psi}) = 0$ yields

$$\frac{1}{nT} \sum_{i=1}^{n} \sum_{t=0}^{T-1} \boldsymbol{m}_{i,t}(\widehat{\psi}) = \frac{1}{nT} \sum_{i=1}^{n} \sum_{t=0}^{T-1} \boldsymbol{m}_{i,t}(\boldsymbol{\psi}^*) + \mathbb{E}\left\{ \nabla_\psi \boldsymbol{m}_t(\boldsymbol{\psi}^*) \right\} (\widehat{\psi} - \boldsymbol{\psi}^*) + o_p(\|\widehat{\psi} - \boldsymbol{\psi}^*\|),$$
$$\text{where } \nabla_\psi \boldsymbol{m}_t(\boldsymbol{\psi}^*) = \frac{\eta_{t+1} \boldsymbol{h}(S_t, A_t, Z_t)}{(1 - \lambda(S_t, A_t, R_{t+1}, S_{t+1}; \boldsymbol{\psi}^*))^2} \cdot \frac{\partial \lambda(S_t, A_t, R_{t+1}, S_{t+1}; \boldsymbol{\psi})}{\partial \boldsymbol{\psi}^\top} \Big|_{\boldsymbol{\psi} = \boldsymbol{\psi}^*}. \tag{25}$$

According to Central Limit Theorem (CLT) for M-estimators,

$$\sqrt{nT}(\widehat{\psi} - \boldsymbol{\psi}^*) = -\frac{1}{\sqrt{nT}} \sum_{i=1}^{n} \sum_{t=0}^{T-1} \boldsymbol{\phi}_{i,t+1} + o_p(1) \xrightarrow{d} \mathcal{N}(\mathbf{0}, \boldsymbol{\Sigma}_{\psi^*}), \tag{26}$$

where $\phi_{i,t+1} = \left[\mathbb{E}\left\{\nabla_{\psi}\boldsymbol{m}_t(\boldsymbol{\psi}^*)\right\}\right]^{-1}\boldsymbol{m}_t(\boldsymbol{\psi}^*)$, and $\boldsymbol{\Sigma}_{\psi^*} = \left[\mathbb{E}\left\{\nabla_{\psi}\boldsymbol{m}_t(\boldsymbol{\psi}^*)\right\}\right]^{-1}\mathrm{Var}(\boldsymbol{m}_t(\boldsymbol{\psi}^*))\left[\mathbb{E}\left\{\nabla_{\psi}\boldsymbol{m}_t(\boldsymbol{\psi}^*)\right\}\right]^{-1}$.

Plug (26) into (24) yields

$$\frac{1}{nT}\sum_{i=1}^{n}\sum_{t=0}^{T-1}\left(\omega_{i,t+1}(\widehat{\boldsymbol{\psi}}) - \omega_{i,t+1}(\boldsymbol{\psi}^*)\right)\boldsymbol{\xi}_{i,t}\varepsilon_{i,t} = \boldsymbol{H}_1(\widehat{\boldsymbol{\psi}} - \boldsymbol{\psi}^*) + o_p(\|\widehat{\boldsymbol{\psi}} - \boldsymbol{\psi}^*\|^2) = O_p\{(nT)^{-1/2}\}. \qquad (27)$$

Combining (23) and (27) yields

$$\frac{1}{nT}\sum_{i=1}^{n}\sum_{t=0}^{T-1}\omega_{i,t+1}(\widehat{\boldsymbol{\psi}})\boldsymbol{\xi}_{i,t}\varepsilon_{i,t} = O_p\{\sqrt{L/(nT)}\}.$$

Together with Lemma G.8, we obtain

$$\zeta_2 = O_p\{\sqrt{L/nT}\log(nT)\}O_p\{\sqrt{L/(nT)}\} = O_p\left\{L(nT)^{-1}\log(nT)\right\}. \qquad (28)$$

**Error bounds for $\|\zeta_1\|_2$.** Similarly, we obtain the error bound for $\zeta_1$ as follows

$$\zeta_1 = O_p\left\{L^{1/2}(nT)^{-1/2}\right\}. \qquad (29)$$

Combining (20), (28) and (29), we have

$$\widehat{\boldsymbol{\beta}}_{\mathrm{IPW}} - \boldsymbol{\beta}^* = O_p\left\{L^{1/2}(nT)^{-1/2}\right\} + O_p\left\{L^{-p/d}\right\} + O_p\left\{L(nT)^{-1}\log(nT)\right\} = o_p(1). \qquad (30)$$

Following similar arguments as (19), we show $|\widehat{V}_{\mathrm{IPW}}^{\pi}(\mathbb{G}) - V^{\pi}(\mathbb{G})| = o_p(1)$, therefore $\widehat{V}_{\mathrm{IPW}}^{\pi}(\mathbb{G})$ is a consistent value estimator.

$\square$

### G.3. Proof of Theorem 4.7

**Theorem** (Bidirectional Asymptotics) Assume conditions A.1-A.2 hold. As either $n \to \infty$ or $T \to \infty$, we have

$$\sqrt{nT}\widehat{\sigma}_{\pi,\mathrm{IPW}}^{-1}(\mathbb{G})\{\widehat{V}_{\mathrm{IPW}}^{\pi}(\mathbb{G}) - V^{\pi}(\mathbb{G})\} \xrightarrow{d} \mathcal{N}(0,1).$$

*Proof.* We first provide an outline for the proof, which consists of four steps. In the first step, we give the form of $\widehat{\sigma}_{\mathrm{IPW}}^2(s)$ and $\sigma_{\pi,\mathrm{IPW}}^2(s)$. In the second step, we show the linear representation $\sqrt{nT}\left\{\widehat{V}_{\mathrm{IPW}}^{\pi}(\mathbb{G}) - V^{\pi}(\mathbb{G})\right\}/\sigma_{\pi,\mathrm{IPW}}(\mathbb{G}) = \sqrt{nT}\left\{\int_s \boldsymbol{U}(s)\mathbb{G}(ds)\right\}^{\top}\zeta_1/\sigma_{\pi,\mathrm{IPW}}(\mathbb{G}) + o_p(1)$ holds. In the third step, we show $\sqrt{nT}\left\{\int_s \boldsymbol{U}(s)\mathbb{G}(ds)\right\}^{\top}\zeta_1/\sigma_{\pi,\mathrm{IPW}}(\mathbb{G}) \xrightarrow{d} \mathcal{N}(0,1)$ based on the martingale central limit theorem. In the last step, we show $\widehat{\sigma}_{\pi,\mathrm{IPW}}(\mathbb{G})/\sigma_{\pi,\mathrm{IPW}}(\mathbb{G}) \xrightarrow{p} 1$. A detailed proof is presented as follows. For succinctness, we will write $\omega_{t+1}(\widehat{\boldsymbol{\psi}})$ and $\omega_{t+1}(\boldsymbol{\psi}^*)$ as $\widehat{\omega}_{t+1}$ and $\omega_{t+1}^*$ respectively for the rest of the derivation, and use notation $\omega_{t+1,\psi}$ to replace $\omega_{t+1}(\boldsymbol{\psi})$. Also, we will use $\boldsymbol{m}_t^*, \widehat{\boldsymbol{m}}_t$ to represent $\boldsymbol{m}_t(\boldsymbol{\psi}^*)$, $\boldsymbol{m}_t(\widehat{\boldsymbol{\psi}})$.

**Step 1.** Derive $\widehat{\sigma}_{\pi,\mathrm{IPW}}^2(s)$ and $\sigma_{\pi,\mathrm{IPW}}^2(s)$.

In the previous proof, we have already shown that $\widehat{\boldsymbol{\beta}}_{\mathrm{IPW}} - \boldsymbol{\beta}^* = \zeta_1 + \zeta_2 + \zeta_3$, where

$$\zeta_1 = \boldsymbol{\Sigma}^{-1}\left(\frac{1}{nT}\sum_{i=1}^{n}\sum_{t=0}^{T-1}\widehat{\omega}_{i,t+1}\boldsymbol{\xi}_{i,t}\varepsilon_{i,t}\right) = O_p\left\{L^{1/2}(nT)^{-1/2}\right\}$$

$$\zeta_2 = \left(\widehat{\boldsymbol{\Sigma}}_{\mathrm{IPW}}^{-1} - \boldsymbol{\Sigma}^{-1}\right)\left(\frac{1}{nT}\sum_{i=1}^{n}\sum_{t=0}^{T-1}\widehat{\omega}_{i,t+1}\boldsymbol{\xi}_{i,t}\varepsilon_{i,t}\right) = O_p\left\{L(nT)^{-1}\log(nT)\right\}$$

$$\zeta_3 = \widehat{\boldsymbol{\Sigma}}_{\mathrm{IPW}}^{-1}\left(\frac{1}{nT}\sum_{i=1}^{n}\sum_{t=0}^{T-1}\widehat{\omega}_{i,t+1}\boldsymbol{\xi}_{i,t}r_{i,t}\right) = O_p\left(L^{-p/d}\right)$$

As long as the number of basis $L$ satisfies Assumption A.1(e), we have

$$\sqrt{nT}(\widehat{\boldsymbol{\beta}}_{\text{IPW}} - \boldsymbol{\beta}^*) = \sqrt{nT}\zeta_1 + o_p(1) = \boldsymbol{\Sigma}^{-1}\left(\frac{1}{\sqrt{nT}}\sum_{i=1}^n\sum_{t=0}^{T-1}\widehat{\omega}_{i,t+1}\boldsymbol{\xi}_{i,t}\varepsilon_{i,t}\right) + o_p(1).$$

Plug (26) into the mean expansion in (24) and define $\boldsymbol{H}_2 = \boldsymbol{H}_1\mathbb{E}[\nabla_{\boldsymbol{\psi}}\boldsymbol{m}_t(\boldsymbol{\psi}^*)]^{-1}$, we can express $\sqrt{nT}\zeta_1$ as follows

$$\sqrt{nT}\zeta_1 = \boldsymbol{\Sigma}^{-1}\left\{\frac{1}{\sqrt{nT}}\sum_{i=1}^n\sum_{t=0}^{T-1}\left(\omega_{i,t+1}^*\boldsymbol{\xi}_{i,t}\varepsilon_{i,t} - \boldsymbol{H}_1\boldsymbol{\phi}_{i,t+1}\right)\right\} + o_p(1)$$

$$= \boldsymbol{\Sigma}^{-1}\left\{\frac{1}{\sqrt{nT}}\sum_{i=1}^n\sum_{t=0}^{T-1}\left(\omega_{i,t+1}^*\boldsymbol{\xi}_{i,t}\varepsilon_{i,t} - \boldsymbol{H}_2\boldsymbol{m}_{i,t}(\boldsymbol{\psi}^*)\right)\right\} + o_p(1).$$

Let $\zeta_{i,t} := \omega_{i,t+1}^*\boldsymbol{\xi}_{i,t}\varepsilon_{i,t} - \boldsymbol{H}_2\boldsymbol{m}_{i,t}^*$, the expression for $\sqrt{nT}\zeta_1$ can be simplified as

$$\sqrt{nT}\zeta_1 = \boldsymbol{\Sigma}^{-1}\left\{\frac{1}{\sqrt{nT}}\sum_{i=1}^n\sum_{t=0}^{T-1}\zeta_{i,t}\right\} + o_p(1) \xrightarrow{d} \mathcal{N}\left(\boldsymbol{0}, \boldsymbol{\Sigma}^{-1}\boldsymbol{\Omega}_{\text{IPW}}(\boldsymbol{\Sigma}^\top)^{-1}\right), \tag{31}$$

where

$$\boldsymbol{\Omega}_{\text{IPW}} = \mathbb{E}\left(\zeta_{i,t}\zeta_{i,t}^\top\right) = \mathbb{E}\left\{(\omega_{i,t+1}^*\boldsymbol{\xi}_{i,t}\varepsilon_{i,t} - \boldsymbol{H}_2\boldsymbol{m}_{i,t}^*)(\omega_{i,t+1}^*\boldsymbol{\xi}_{i,t}\varepsilon_{i,t} - \boldsymbol{H}_2\boldsymbol{m}_{i,t}^*)^\top\right\}$$

*Remark* G.9. The estimator $\widehat{\boldsymbol{\Omega}}_{\text{IPW}}$ can be calculated using the empirical form.

$$\widehat{\boldsymbol{\Omega}}_{\text{IPW}} = \frac{1}{nT}\sum_{i=1}^n\sum_{t=0}^{T-1}\left\{(\widehat{\omega}_{i,t+1}\boldsymbol{\xi}_{i,t}\widehat{\varepsilon}_{i,t} - \widehat{\boldsymbol{H}}_2\widehat{\boldsymbol{m}}_{i,t})(\widehat{\omega}_{i,t+1}\boldsymbol{\xi}_{i,t}\widehat{\varepsilon}_{i,t} - \widehat{\boldsymbol{H}}_2\widehat{\boldsymbol{m}}_{i,t})^\top\right\}, \tag{32}$$

where

$$\widehat{\omega}_{i,t+1} = \eta_{i,t+1}/\{1 - \lambda(S_{i,t}, A_{i,t}, R_{i,t+1}, S_{i,t+1}; \widehat{\boldsymbol{\psi}}_{nT})\},$$

$$\widehat{\varepsilon}_{i,t} = R_{i,t+1} + \gamma\sum_{a\in\mathcal{A}}\Phi_L^\top(S_{i,t+1})\widehat{\boldsymbol{\beta}}_a\pi(a|S_{i,t+1}) - \Phi_L^\top(S_{i,t})\widehat{\boldsymbol{\beta}}_{A_{i,t}},$$

$$\widehat{\boldsymbol{H}}_2 = \left[\frac{1}{nT}\sum_{i=1}^n\sum_{t=0}^{T-1}\boldsymbol{\xi}_{i,t}\widehat{\varepsilon}_{i,t}\widehat{\nabla}_{\boldsymbol{\psi}}\omega_{t+1}(\widehat{\boldsymbol{\psi}}_{nT})^\top\right]\left[\frac{1}{nT}\sum_{i=1}^n\sum_{t=0}^{T-1}\boldsymbol{h}_{i,t}\widehat{\nabla}_{\boldsymbol{\psi}}\omega_{t+1}(\widehat{\boldsymbol{\psi}}_{nT})^\top\right]^{-1},$$

$$\widehat{\nabla}_{\boldsymbol{\psi}}\omega_{t+1}(\widehat{\boldsymbol{\psi}}_{nT})^\top = \frac{\eta_{t+1}}{(1 - \lambda(S_t, A_t, R_{t+1}, S_{t+1}; \widehat{\boldsymbol{\psi}}_{nT}))^2} \cdot \frac{\partial\lambda(S_t, A_t, R_{t+1}, S_{t+1}; \boldsymbol{\psi})}{\partial\boldsymbol{\psi}^\top}\Big|_{\boldsymbol{\psi}=\widehat{\boldsymbol{\psi}}_{nT}}.$$

For any parametric model of $\lambda(\boldsymbol{\psi})$, $\widehat{\omega}_{i,t+1}$ can be estimated directly by substituting the expressions for $\widehat{\omega}_{i,t+1}$, $\widehat{\varepsilon}_{i,t}$, $\widehat{\boldsymbol{H}}_2$, and $\widehat{\nabla}_{\boldsymbol{\psi}}\omega_{t+1}(\widehat{\boldsymbol{\psi}}_{nT})^\top$ as defined above. However, when using a semi-parametric model to estimate $\boldsymbol{\psi}$, as described in Appendix B.2, computing the explicit expression for $\widehat{\nabla}_{\boldsymbol{\psi}}\omega_{t+1}(\widehat{\boldsymbol{\psi}}_{nT})^\top$ becomes challenging, as noted by Shao & Wang (2016). Due to the complexity of $\widehat{\boldsymbol{\Omega}}_{\text{IPW}}$, we simplify by ignoring the uncertainty from dropout propensity estimation and retaining only the first term. Specifically, we approximate $\boldsymbol{\Omega}_{\text{IPW}}$ with $\widetilde{\boldsymbol{\Omega}}_{\text{IPW}}$ for variance calculation in the semi-parametric dropout model, as given by:

$$\widetilde{\boldsymbol{\Omega}}_{\text{IPW}} = \frac{1}{nT}\sum_{i=1}^n\sum_{t=0}^{T-1}\widehat{\omega}_{i,t+1}^2\widehat{\varepsilon}_{i,t}^2\boldsymbol{\xi}_{i,t}\boldsymbol{\xi}_{i,t}^\top$$

$$= \frac{1}{nT}\sum_{i=1}^n\sum_{t=0}^{T-1}\boldsymbol{\xi}_{i,t}\boldsymbol{\xi}_{i,t}^\top\left\{\frac{\eta_{i,t+1}}{1 - \lambda(S_{i,t}, A_{i,t}, R_{i,t+1}, S_{i,t+1}; \widehat{\boldsymbol{\psi}}_{nT})}\times\right. \tag{33}$$

$$\left.(R_{i,t+1} + \gamma\sum_{a\in\mathcal{A}}\Phi_L^\top(S_{i,t+1})\widehat{\boldsymbol{\beta}}_a\pi(a|S_{i,t+1}) - \Phi_L^\top(S_{i,t})\widehat{\boldsymbol{\beta}}_{A_{i,t}})\right\}^2.$$

The asymptotic variance of $\widehat{V}^\pi_{\mathrm{IPW}}(s)$ and its estimator are given by

$$\sigma^2_{\pi,\mathrm{IPW}}(s) = \boldsymbol{U}^\top_\pi(s)\boldsymbol{\Sigma}^{-1}\boldsymbol{\Omega}_{\mathrm{IPW}}(\boldsymbol{\Sigma}^\top)^{-1}\boldsymbol{U}_\pi(s),$$

$$\widehat{\sigma}^2_{\pi,\mathrm{IPW}}(s) = \boldsymbol{U}^\top_\pi(s)\widehat{\boldsymbol{\Sigma}}^{-1}_{\mathrm{IPW}}\widehat{\boldsymbol{\Omega}}_{\mathrm{IPW}}(\widehat{\boldsymbol{\Sigma}}^\top_{\mathrm{IPW}})^{-1}\boldsymbol{U}_\pi(s),$$

where $\widehat{\boldsymbol{\Omega}}_{\mathrm{IPW}}$ can be replaced by $\widetilde{\boldsymbol{\Omega}}_{\mathrm{IPW}}$ as defined in Equation (33) for semi-parametric dropout estimation.

It then follows that

$$\sigma^2_{\pi,\mathrm{IPW}}(\mathbb{G}) = \left\{\int_s \boldsymbol{U}(s)\mathbb{G}(ds)\right\}^\top \boldsymbol{\Sigma}^{-1}\boldsymbol{\Omega}_{\mathrm{IPW}}(\boldsymbol{\Sigma}^\top)^{-1}\left\{\int_s \boldsymbol{U}(s)\mathbb{G}(ds)\right\},$$

$$\widehat{\sigma}^2_{\pi,\mathrm{IPW}}(\mathbb{G}) = \left\{\int_s \boldsymbol{U}(s)\mathbb{G}(ds)\right\}^\top \widehat{\boldsymbol{\Sigma}}^{-1}_{\mathrm{IPW}}\widehat{\boldsymbol{\Omega}}_{\mathrm{IPW}}(\widehat{\boldsymbol{\Sigma}}^\top_{\mathrm{IPW}})^{-1}\left\{\int_s \boldsymbol{U}(s)\mathbb{G}(ds)\right\} \tag{34}$$

Again, for semi-parametric dropout model, asymptotic variance can be approximated by $\widetilde{\sigma}^2_{\pi,\mathrm{IPW}}(\mathbb{G})$, given by

$$\widetilde{\sigma}^2_{\pi,\mathrm{IPW}}(\mathbb{G}) = \left\{\int_{s\in\mathcal{S}} \boldsymbol{U}_\pi(s)\mathbb{G}(ds)\right\}^\top \widehat{\boldsymbol{\Sigma}}^{-1}_{\mathrm{IPW}}\widetilde{\boldsymbol{\Omega}}_{\mathrm{IPW}}(\widehat{\boldsymbol{\Sigma}}^\top_{\mathrm{IPW}})^{-1}\left\{\int_{s\in\mathcal{S}} \boldsymbol{U}_\pi(s)\mathbb{G}(ds)\right\}. \tag{35}$$

**Step 2.** Show the following linear representation holds

$$\frac{\sqrt{nT}\left\{\widehat{V}^\pi_{\mathrm{IPW}}(\mathbb{G}) - V^\pi(\mathbb{G})\right\}}{\sigma_{\pi,\mathrm{IPW}}(\mathbb{G})} = \frac{\sqrt{nT}\left\{\int_s \boldsymbol{U}(s)\mathbb{G}(ds)\right\}^\top \zeta_1}{\sigma_{\pi,\mathrm{IPW}}(\mathbb{G})} + o_p(1). \tag{36}$$

Using arguments similar to step 2 of Theorem 1's proof in Shi et al. (2021b), we have

$$\left|\widehat{V}^\pi_{\mathrm{IPW}}(\mathbb{G}) - V^\pi(\mathbb{G}) - \left\{\int_s \boldsymbol{U}(s)\mathbb{G}(ds)\right\}^\top \zeta_1\right| \leq \left\|\int_s \boldsymbol{U}(s)\mathbb{G}(ds)\right\|_2 \left\|\widehat{\boldsymbol{\beta}}_{\mathrm{IPW}} - \boldsymbol{\beta}^* - \zeta_1\right\|_2 + CL^{-p/d}. \tag{37}$$

Here we introduce the following lemma.

**Lemma G.10.** *Suppose Assumption A.1-A.2 holds. Then there exist $C_{\Omega,1}$ such that $\lambda_{\min}(\boldsymbol{\Omega}_{IPW}) \geq C_{\Omega,1}$ with probability approaching 1. Besides, $\lambda_{\max}(\boldsymbol{\Omega}_{IPW}) = O_p(1)$.*

Lemma G.10 can be shown by noting that

$$\lambda_{\min}\left(\frac{1}{T}\sum_{t=0}^{T-1}\mathbb{E}\left\{\omega^{*2}_{i,t+1}\varepsilon^2_{i,t}\boldsymbol{\xi}_{i,t}\boldsymbol{\xi}^\top_{i,t}\right\}\right) = \lambda_{\min}\left(\frac{1}{T}\sum_{t=0}^{T-1}\mathbb{E}\left\{\frac{1}{1-\lambda^*_{i,t}}\varepsilon^2_{i,t}\boldsymbol{\xi}_{i,t}\boldsymbol{\xi}^\top_{i,t}\right\}\right)$$

$$\geq \lambda_{\min}\left(\frac{1}{T}\sum_{t=0}^{T-1}\mathbb{E}\left\{\varepsilon^2_{i,t}\boldsymbol{\xi}_{i,t}\boldsymbol{\xi}^\top_{i,t}\right\}\right) \geq c_0^{-1}\lambda_{\min}\left(\frac{1}{T}\sum_{t=0}^{T-1}\mathbb{E}\left\{\boldsymbol{\xi}_{i,t}\boldsymbol{\xi}^\top_{i,t}\right\}\right) \geq \frac{\overline{c}}{3c_0} := C_{\Omega,1},$$

$$\lambda_{\max}(\boldsymbol{A}) \leq c_\lambda^{-1}(c_0 + 2c')^2\lambda_{\max}\left(\frac{1}{T}\sum_{t=0}^{T-1}\mathbb{E}\left\{\boldsymbol{\xi}_{i,t}\boldsymbol{\xi}^\top_{i,t}\right\}\right) = O_p(1).$$

By Lemma G.10, the lower bound of $\sigma^2_{\pi,\mathrm{IPW}}(\mathbb{G})$ satisfies

$$\sigma^2_{\pi,\mathrm{IPW}}(\mathbb{G}) \geq C_{\Omega,1}\left\{\int_s \boldsymbol{U}(s)\mathbb{G}(ds)\right\}^\top \boldsymbol{\Sigma}^{-1}\left(\boldsymbol{\Sigma}^\top\right)^{-1}\left\{\int_s \boldsymbol{U}(s)\mathbb{G}(ds)\right\}. \tag{38}$$

According to Lemma G.3, we have $\lambda_{\max}(\boldsymbol{\Sigma}^\top\boldsymbol{\Sigma}) = O(1)$. This implies that $\lambda_{\min}\{\boldsymbol{\Sigma}^{-1}(\boldsymbol{\Sigma}^\top)^{-1}\} \geq \bar{C}$ for some constant $\bar{C} > 0$, hence

$$\sigma^2_{\pi,\mathrm{IPW}}(\mathbb{G}) \geq C_{\Omega,1}\bar{C}\left\|\int_s U(s)\mathbb{G}(ds)\right\|^2_2 \tag{39}$$

Combining Equation (39) together with Equation (37) yields that

$$\frac{1}{\sigma_{\pi,\mathrm{IPW}}(\mathbb{G})} \left| \widehat{V}_{\mathrm{IPW}}^{\pi}(\mathbb{G}) - V^{\pi}(\mathbb{G}) - \left\{ \int_s \boldsymbol{U}(s)\mathbb{G}(ds) \right\}^{\top} \zeta_1 \right|$$

$$\leq \frac{1}{\sqrt{C_{\Omega,1}\overline{C}}} \left\| \widehat{\boldsymbol{\beta}}_{\mathrm{IPW}} - \boldsymbol{\beta}^* - \zeta_1 \right\|_2 + \frac{CL^{-p/d}}{\sqrt{C_{\Omega,1}\overline{C}} \left\| \int_s \boldsymbol{U}(s)\mathbb{G}(ds) \right\|_2}.$$

According to the previous proof, we have

$$\widehat{\boldsymbol{\beta}}_{\mathrm{IPW}} - \boldsymbol{\beta}^* = \zeta_1 + O_p\left\{ L(nT)^{-1}\log(nT) \right\} + O_p\left( L^{-p/d} \right)$$

Together with the condition that $L \ll \sqrt{nT}/\log(nT)$ and $L^{2p/d} \gg nT\left\{ 1 + \left\| \int_s \boldsymbol{U}(s)\mathbb{G}(ds) \right\|_2^{-2} \right\}$, we obtain

$$\frac{\sqrt{nT}\{\widehat{V}_{\mathrm{IPW}}^{\pi}(\mathbb{G}) - V^{\pi}(\mathbb{G})\}}{\sigma_{\pi,\mathrm{IPW}}(\mathbb{G})} = \frac{\sqrt{nT}\left\{ \int_s \boldsymbol{U}(s)\mathbb{G}(ds) \right\}^{\top} \zeta_1}{\sigma_{\pi,\mathrm{IPW}}(\mathbb{G})} + o_p(1). \tag{40}$$

This completes the second step of the proof.

**Step 3.** Show

$$\frac{\sqrt{nT}\left\{ \int_s \boldsymbol{U}(s)\mathbb{G}(ds) \right\}^{\top} \zeta_1}{\sigma_{\pi,\mathrm{IPW}}(\mathbb{G})} \xrightarrow{d} \mathcal{N}(0,1).$$

In this step, we first construct a martingale and then apply the martingale central limit theorem. For any integer $1 \leq g \leq nT$, let $i(g)$ and $t(g)$ be the quotient and the remainder of $g + T - 1$ divided by $T$, that is, $g = \{i(g) - 1\} \cdot T + t(g) + 1$, $1 \leq i(g) \leq n, 0 \leq t(g) < T$. Let $\mathcal{F}^{(0)} = \{S_{1,0}, A_{1,0}\}$, then iteratively define $\{\mathcal{F}^{(g)}\}_{1 \leq g \leq nT}$ as follows:

$$\mathcal{F}^{(g)} = \mathcal{F}^{(g-1)} \cup \left\{ R_{i(g),t(g)+1}, \eta_{i(g),t(g)+1}, S_{i(g),t(g)+1}, A_{i(g),t(g)+1} \right\}, \quad \text{if } t(g) < T - 1$$

$$\mathcal{F}^{(g)} = \mathcal{F}^{(g-1)} \cup \left\{ R_{i(g),T}, \eta_{i(g),T}, S_{i(g),T}, S_{i(g)+1,0}, A_{i(g)+1,0} \right\}, \quad \text{otherwise.}$$

Use $\boldsymbol{\xi}^{(g)}, \boldsymbol{m}^{(g)}, \varepsilon^{(g)}, \omega_{\psi}^{(g)}$ to represent $\boldsymbol{\xi}_{i(g),t(g)}, \boldsymbol{m}_{i(g),t(g)}, \varepsilon_{i(g),t(g)}$, and $\omega_{i(g),t(g)+1,\psi}$, respectively. It follows from (31) that

$$\sqrt{nT}\frac{\left\{ \int_s \boldsymbol{U}(s)\mathbb{G}(ds) \right\}^{\top} \zeta_1}{\sigma_{\pi,\mathrm{IPW}}(\mathbb{G})} = \sum_{g=1}^{nT} \frac{\left\{ \int_s \boldsymbol{U}(s)\mathbb{G}(ds) \right\}^{\top} \boldsymbol{\Sigma}^{-1}\zeta^{(g)}}{\sqrt{nT}\sigma_{\pi,\mathrm{IPW}}(\mathbb{G})} + o_p(1), \tag{41}$$

where $\zeta^{(g)} = \omega^{*(g)}\boldsymbol{\xi}^{(g)}\varepsilon^{(g)} - \boldsymbol{H}_2\boldsymbol{m}^{*(g)}$. Using similar arguments as (22), we can show $\mathbb{E}\{\omega^{*(g)}\boldsymbol{\xi}^{(g)}\varepsilon^{(g)} \mid \mathcal{F}^{(g-1)}\} = 0$. Meanwhile, $\mathbb{E}\{\boldsymbol{m}^{*(g)} \mid \mathcal{F}^{(g-1)}\} = 0$ holds as a result of $\mathbb{E}\{\omega^{*(g)} \mid \mathcal{F}^{(g-1)}\} = 1$. Therefore, $\mathbb{E}\{\zeta^{(g)} \mid \mathcal{F}^{(g-1)}\} = \boldsymbol{0}$, the first term of the RHS of (41) forms a martingale with respect to the filtration $\{\sigma(\mathcal{F}^{(g)})\}_{g \geq 0}$, where $\sigma(\mathcal{F}^{(g)})$ stands for the $\sigma$-algebra generated by $\mathcal{F}^{(g)}$.

We can then use a martingale central limit theorem for triangular arrays (Corollary 2.8 of McLeish (1974)) to show the asymptotic normality. This requires to verify the following two conditions:

(a) $\max_{1 \leq g \leq nT} \left| \left\{ \int_s \boldsymbol{U}(s)\mathbb{G}(ds) \right\}^{\top} \boldsymbol{\Sigma}^{-1}\zeta^{(g)} \right| / \{\sqrt{nT}\sigma_{\pi,\mathrm{IPW}}(s)\} \xrightarrow{P} 0.$

(b) $(nT)^{-1}\sum_{g=1}^{nT} \left| \left\{ \int_s \boldsymbol{U}(s)\mathbb{G}(ds) \right\}^{\top} \boldsymbol{\Sigma}^{-1}\zeta^{(g)} \right|^2 / \{\sigma_{\pi,\mathrm{IPW}}^2(s)\} \xrightarrow{P} 1.$

First, we verify condition (a). It follows from Cauchy-Schwarz inequality that

$$\left| \frac{\left\{ \int_s \boldsymbol{U}(s)\mathbb{G}(ds) \right\}^{\top} \boldsymbol{\Sigma}^{-1}\zeta^{(g)}}{\sqrt{nT}\sigma_{\pi,\mathrm{IPW}}(s)} \right| \leq \frac{\left\| \left\{ \int_s \boldsymbol{U}(s)\mathbb{G}(ds) \right\}^{\top} \boldsymbol{\Sigma}^{-1} \right\|_2 \left\| \zeta^{(g)} \right\|_2}{\sqrt{nT}\sigma_{\pi,\mathrm{IPW}}(s)}.$$

Notice that

$$
\begin{aligned}
\left\| \zeta^{(g)} \right\|_2 &= \left\| \omega^{*(g)} \boldsymbol{\xi}^{(g)} \varepsilon^{(g)} - \boldsymbol{H}_2 \boldsymbol{m}^{*(g)} \right\|_2 \leq \left\| \omega^{*(g)} \boldsymbol{\xi}^{(g)} \varepsilon^{(g)} \right\|_2 + \left\| \boldsymbol{H}_2 \boldsymbol{m}^{*(g)} \right\|_2 \\
&\leq |\omega^{*(g)}| \left\| \boldsymbol{\xi}^{(g)} \right\|_2 |\varepsilon^{(g)}| + \|\boldsymbol{H}_2\|_2 \left\| \boldsymbol{m}^{*(g)} \right\|_2 \\
&\leq \frac{(c_0 + 2c')}{c_\lambda} \sup_s \|\Phi_L(s)\|_2 + \|\boldsymbol{H}_2\|_2 \|\boldsymbol{m}^{*(g)}\|_2 \\
&\leq \frac{(c_0 + 2c') c^*}{c_\lambda} \sqrt{L} + \left( \frac{1}{c_\lambda} - 1 \right) \|\boldsymbol{H}_2\|_2 \leq C_\zeta \sqrt{L}, \text{ for some constant } C_\zeta.
\end{aligned}
$$

Together with Equation (39), we have

$$
\left| \frac{\left\{ \int_s \boldsymbol{U}(s) \mathbb{G}(ds) \right\}^\top \boldsymbol{\Sigma}^{-1} \zeta^{(g)}}{\sqrt{nT} \sigma_{\pi,\mathrm{IPW}}(s)} \right| \leq \frac{C_\zeta}{\sqrt{C_{\Omega,1}C}} \frac{\sqrt{L}}{\sqrt{nT}}.
$$

Since $L \ll \sqrt{nT}/\log(nT)$, condition (a) is proven.

Next, we verify condition (b). Notice that

$$
\left| \frac{1}{nT} \sum_{g=1}^{nT} \frac{\left| \left\{ \int_s \boldsymbol{U}(s) \mathbb{G}(ds) \right\}^\top \boldsymbol{\Sigma}^{-1} \zeta^{(g)} \right|^2}{\sigma_{\pi,\mathrm{IPW}}^2(s)} - 1 \right|
$$

$$
= \frac{1}{\sigma_{\pi,\mathrm{IPW}}^2(s)} \times \left| \left\{ \int_s \boldsymbol{U}(s) \mathbb{G}(ds) \right\}^\top \boldsymbol{\Sigma}^{-1} \left( \frac{1}{nT} \sum_{g=1}^{nT} \zeta^{(g)} \zeta^{(g)\top} - \boldsymbol{\Omega}_{\mathrm{IPW}} \right) \left( \boldsymbol{\Sigma}^\top \right)^{-1} \left\{ \int_s \boldsymbol{U}(s) \mathbb{G}(ds) \right\} \right|,
$$

where

$$
\boldsymbol{\Omega}_{\mathrm{IPW}} = \frac{1}{T} \sum_{t=0}^{T-1} \mathbb{E} \left\{ \zeta^{(g)} \zeta^{(g)\top} \right\}
$$

In view of Equation (38), it suffices to show

$$
\left\| \frac{1}{nT} \sum_{g=1}^{nT} \zeta^{(g)} \zeta^{(g)\top} - \boldsymbol{\Omega}_{\mathrm{IPW}} \right\|_2 = o_p(1). \tag{42}
$$

This can be proven using similar arguments in bounding $\|\widehat{\boldsymbol{\Sigma}} - \boldsymbol{\Sigma}\|_2$ in the proof of Lemma G.3. Therefore,

$$
\sqrt{nT} \frac{\left\{ \int_s \boldsymbol{U}(s) \mathbb{G}(ds) \right\}^\top \zeta_1}{\sigma_{\pi,\mathrm{IPW}}(\mathbb{G})} = \sum_{g=1}^{nT} \frac{\left\{ \int_s \boldsymbol{U}(s) \mathbb{G}(ds) \right\}^\top \boldsymbol{\Sigma}^{-1} \zeta^{(g)}}{\sqrt{nT} \sigma_{\pi,\mathrm{IPW}}(\mathbb{G})} \xrightarrow{d} \mathcal{N}(0,1).
$$

It follows from (40) and Slutsky's theorem that,

$$
\frac{\sqrt{nT} \{\widehat{V}^\pi(\mathbb{G}) - V^\pi(\mathbb{G})\}}{\sigma_{\pi,\mathrm{IPW}}(\mathbb{G})} \xrightarrow{d} \mathcal{N}(0,1).
$$

**Step 4.** Show $\widehat{\sigma}_\pi(\mathbb{G})/\sigma_{\pi,\mathrm{IPW}}(\mathbb{G}) \xrightarrow{p} 1$.

Using similar arguments in verifying condition (b), it suffices to show $\|\widehat{\boldsymbol{\Sigma}}_{\mathrm{IPW}}^{-1} \widehat{\boldsymbol{\Omega}}_{\mathrm{IPW}} (\widehat{\boldsymbol{\Sigma}}_{\mathrm{IPW}}^\top)^{-1} - \boldsymbol{\Sigma}^{-1} \boldsymbol{\Omega}_{\mathrm{IPW}} (\boldsymbol{\Sigma}^\top)^{-1}\|_2 = o_p(1)$. Lemma G.10 indicates that $\|\boldsymbol{\Omega}_{\mathrm{IPW}}\|_2 = O_p(1)$. This together with Lemma G.8 and the condition $L \ll \sqrt{nT}/\log(nT)$ yields that

$$
\begin{aligned}
\|\widehat{\boldsymbol{\Sigma}}_{\mathrm{IPW}}^{-1} \boldsymbol{\Omega}_{\mathrm{IPW}} (\widehat{\boldsymbol{\Sigma}}_{\mathrm{IPW}}^\top)^{-1} - \boldsymbol{\Sigma}^{-1} \boldsymbol{\Omega}_{\mathrm{IPW}} \left( \boldsymbol{\Sigma}^\top \right)^{-1}\|_2 &\leq \|\widehat{\boldsymbol{\Sigma}}_{\mathrm{IPW}}^{-1} - \boldsymbol{\Sigma}^{-1}\|_2 \|\boldsymbol{\Omega}_{\mathrm{IPW}}\|_2 \|\widehat{\boldsymbol{\Sigma}}_{\mathrm{IPW}}^{-1}\|_2 \\
&\quad + \|\boldsymbol{\Sigma}_{\mathrm{IPW}}^{-1}\|_2 \|\boldsymbol{\Omega}_{\mathrm{IPW}}\|_2 \|\widehat{\boldsymbol{\Sigma}}_{\mathrm{IPW}}^{-1} - \boldsymbol{\Sigma}^{-1}\|_2 \\
&= O_p \left\{ L^{1/2} (nT)^{-1/2} \log(nT) \right\} = o_p(1).
\end{aligned}
$$

Thus, it remains to show $\|\widehat{\boldsymbol{\Sigma}}_{\mathrm{IPW}}^{-1}\widehat{\boldsymbol{\Omega}}_{\mathrm{IPW}}(\widehat{\boldsymbol{\Sigma}}_{\mathrm{IPW}}^{\top})^{-1} - \widehat{\boldsymbol{\Sigma}}_{\mathrm{IPW}}^{-1}\boldsymbol{\Omega}_{\mathrm{IPW}}(\widehat{\boldsymbol{\Sigma}}_{\mathrm{IPW}}^{\top})^{-1}\|_2 = o_p(1)$, or

$$\|\widehat{\boldsymbol{\Omega}}_{\mathrm{IPW}} - \boldsymbol{\Omega}_{\mathrm{IPW}}\|_2 = o_p(1). \tag{43}$$

In view of (42), we only need to show $\|\widehat{\boldsymbol{\Omega}}_{\mathrm{IPW}} - \frac{1}{nT}\sum_{g=1}^{nT}\zeta^{(g)}\zeta^{(g)\top}\|_2 = o_p(1)$. Notice that

$$\widehat{\boldsymbol{\Omega}}_{\mathrm{IPW}} - \frac{1}{nT}\sum_{g=1}^{nT}\zeta^{(g)}\zeta^{(g)\top} = \frac{1}{nT}\sum_{g=1}^{nT}\left\{\widehat{\zeta}^{(g)}\widehat{\zeta}^{(g)\top} - \zeta^{(g)}\zeta^{(g)\top}\right\}$$

$$= \frac{1}{nT}\sum_{g=1}^{nT}\left\{\left(\widehat{\zeta}^{(g)} - \zeta^{(g)}\right)\widehat{\zeta}^{(g)\top} + \zeta^{(g)}\left(\widehat{\zeta}^{(g)} - \zeta^{(g)}\right)^{\top}\right\}.$$

By the triangle inequality, we have

$$\left\|\widehat{\boldsymbol{\Omega}}_{\mathrm{IPW}} - \frac{1}{nT}\sum_{g=1}^{nT}\zeta^{(g)}\zeta^{(g)\top}\right\|_2 \le \left\|\frac{1}{nT}\sum_{g=1}^{nT}\left(\widehat{\zeta}^{(g)} - \zeta^{(g)}\right)\widehat{\zeta}^{(g)\top}\right\|_2 + \left\|\frac{1}{nT}\sum_{g=1}^{nT}\zeta^{(g)}\left(\widehat{\zeta}^{(g)} - \zeta^{(g)}\right)^{\top}\right\|_2. \tag{44}$$

It suffices to show

$$\left\|\frac{1}{nT}\sum_{g=1}^{nT}(\widehat{\zeta}^{(g)} - \zeta^{(g)})\widehat{\zeta}^{(g)\top}\right\|_2 = o_p(1) \text{ and } \left\|\frac{1}{nT}\sum_{g=1}^{nT}\zeta^{(g)}(\widehat{\zeta}^{(g)} - \zeta^{(g)})^{\top}\right\|_2 = o_p(1).$$

Recall that $\widehat{\zeta}^{(g)} = \widehat{\omega}^{(g)}\widehat{\varepsilon}^{(g)}\boldsymbol{\xi}^{(g)} - \widehat{\boldsymbol{H}}_2\widehat{\boldsymbol{m}}^{(g)}$, where

$$\widehat{\varepsilon}^{(g)} = R_{i(g),t(g)+1} + \gamma\sum_{a\in\mathcal{A}}\pi\left(a \mid S_{i(g),t(g)+1}\right)\Phi_L^{\top}\left(S_{i(g),t(g)+1}\right)\widehat{\boldsymbol{\beta}}_a - \Phi_L^{\top}\left(S_{i(g),t(g)}\right)\widehat{\boldsymbol{\beta}}_{A_{i(g),t(g)}},$$

and $\widehat{\omega}^{(g)}, \widehat{\boldsymbol{H}}_2, \widehat{\boldsymbol{m}}^{(g)}$ are obtained by plugging in $\widehat{\boldsymbol{\psi}}$.

We first show $\|\frac{1}{nT}\sum_{g=1}^{nT}\zeta^{(g)}(\widehat{\zeta}^{(g)} - \zeta^{(g)})^{\top}\|_2 = \|\frac{1}{nT}\sum_{g=1}^{nT}(\widehat{\zeta}^{(g)} - \zeta^{(g)})\zeta^{(g)\top}\|_2 = o_p(1)$, the other statement can be shown using similar arguments. By definition, $\widehat{\zeta}^{(g)} - \zeta^{(g)}$ can be expressed as

$$\widehat{\zeta}^{(g)} - \zeta^{(g)} = \widehat{\omega}^{(g)}\widehat{\varepsilon}^{(g)}\boldsymbol{\xi}^{(g)} - \widehat{\boldsymbol{H}}_2\widehat{\boldsymbol{m}}^{(g)} - \omega^{*(g)}\varepsilon^{(g)}\boldsymbol{\xi}^{(g)} + \boldsymbol{H}_2\boldsymbol{m}^{*(g)}$$

$$= \underbrace{\left(\widehat{\omega}^{(g)}\widehat{\varepsilon}^{(g)} - \omega^{*(g)}\varepsilon^{(g)}\right)\boldsymbol{\xi}^{(g)}}_{\boldsymbol{E}_1^{(g)}} - \underbrace{(\widehat{\boldsymbol{H}}_2\widehat{\boldsymbol{m}}^{(g)} - \boldsymbol{H}_2\boldsymbol{m}^{*(g)})}_{\boldsymbol{E}_2^{(g)}},$$

and $\zeta^{(g)}$ can be expressed as

$$\zeta^{(g)} = \underbrace{\omega^{*(g)}\varepsilon^{(g)}\boldsymbol{\xi}^{(g)}}_{\boldsymbol{E}_3^{(g)}} - \underbrace{\boldsymbol{H}_2\boldsymbol{m}^{*(g)}}_{\boldsymbol{E}_4^{(g)}}.$$

By the triangle inequality,

$$\left\|\frac{1}{nT}\sum_{g=1}^{nT}(\widehat{\zeta}^{(g)} - \zeta^{(g)})\zeta^{(g)\top}\right\|_2 \le \left\|\frac{1}{nT}\sum_{g=1}^{nT}\boldsymbol{E}_1^{(g)}\boldsymbol{E}_3^{(g)\top}\right\|_2 + \left\|\frac{1}{nT}\sum_{g=1}^{nT}\boldsymbol{E}_1^{(g)}\boldsymbol{E}_4^{(g)\top}\right\|_2$$

$$+ \left\|\frac{1}{nT}\sum_{g=1}^{nT}\boldsymbol{E}_2^{(g)}\boldsymbol{E}_3^{(g)\top}\right\|_2 + \left\|\frac{1}{nT}\sum_{g=1}^{nT}\boldsymbol{E}_2^{(g)}\boldsymbol{E}_4^{(g)\top}\right\|_2.$$

Thus, it suffices to show $\|\frac{1}{nT}\sum_{g=1}^{nT}\boldsymbol{E}_i^{(g)}\boldsymbol{E}_j^{(g)\top}\|_2 = o_p(1)$ for all $i \in \{1, 2\}, j \in \{3, 4\}$.

We first show $\|\frac{1}{nT}\sum_{g=1}^{nT}\boldsymbol{E}_1^{(g)}\boldsymbol{E}_3^{(g)\top}\|_2 = o_p(1)$. It is equivalent to showing

$$\sup_{\boldsymbol{a}\in\mathbb{S}^{mL-1}}\left|\frac{1}{nT}\sum_{g=1}^{nT}\boldsymbol{a}^{\top}\boldsymbol{\xi}^{(g)}\boldsymbol{\xi}^{(g)\top}\boldsymbol{a}\left(\widehat{\omega}^{(g)}\widehat{\varepsilon}^{(g)} - \omega^{*(g)}\varepsilon^{(g)}\right)\omega^{*(g)}\varepsilon^{(g)}\right| = o_p(1),$$

where $\mathbb{S}^{mL-1}$ denotes the unit sphere $\{\boldsymbol{a} \in \mathbb{R}^{mL} : \|\boldsymbol{a}\|_2 = 1\}$. According to Lemma G.4, $\sup_{\boldsymbol{a} \in \mathbb{S}^{mL-1}} \frac{1}{nT} \sum_{g=1}^{nT} \boldsymbol{a}^\top \boldsymbol{\xi}^{(g)} \boldsymbol{\xi}^{(g)\top} \boldsymbol{a} = O_p(1)$, hence we only need to show $\max_{1 \leq g \leq nT} |(\widehat{\omega}^{(g)} \widehat{\varepsilon}^{(g)} - \omega^{*(g)} \varepsilon^{(g)}) \widehat{\omega}^{(g)} \widehat{\varepsilon}^{(g)}| = o_p(1)$. The bound in (13) indicates that $\varepsilon^{(g)}$'s are uniformly bounded, together with the bound for $\omega^{*(g)}$ given in condition A.2(b), we obtain $\max_{1 \leq g \leq nT} |\omega^{*(g)} \varepsilon^{(g)}| = O_p(1)$. Therefore, it remains to show

$$\max_{1 \leq g \leq nT} \left| \widehat{\omega}^{(g)} \widehat{\varepsilon}^{(g)} - \omega^{*(g)} \varepsilon^{(g)} \right| = o_p(1).$$

Note that the term can be decomposed as $\widehat{\omega}^{(g)} \widehat{\varepsilon}^{(g)} - \omega^{*(g)} \varepsilon^{(g)} = (\widehat{\omega}^{(g)} - \omega^{*(g)}) \widehat{\varepsilon}^{(g)} + \omega^{*(g)} (\widehat{\varepsilon}^{(g)} - \varepsilon^{(g)})$. Using similar arguments in showing (E.50) in Shi et al. (2021b), we have $\max_{1 \leq g \leq nT} |\varepsilon^{(g)} - \widehat{\varepsilon}^{(g)}| = o_p(1)$. On the other hand, the consistency of dropout propensity model, $\widehat{\boldsymbol{\psi}} \xrightarrow{p} \boldsymbol{\psi}^*$, indicates that $\widehat{\omega}^{(g)} \xrightarrow{P} \omega^{*(g)}$ for any $g$, thus $\max_{1 \leq g \leq nT} |\omega^{(g)} - \widehat{\omega}^{(g)}| = o_p(1)$. Combine them together yields

$$\begin{aligned}
\max_{1 \leq g \leq nT} |\widehat{\omega}^{(g)} \widehat{\varepsilon}^{(g)} - \omega^{*(g)} \varepsilon^{(g)}| &= \max_{1 \leq g \leq nT} |(\widehat{\omega}^{(g)} - \omega^{*(g)}) \widehat{\varepsilon}^{(g)} + \omega^{*(g)} (\widehat{\varepsilon}^{(g)} - \varepsilon^{(g)})| \\
&\leq \max_{1 \leq g \leq nT} |\widehat{\omega}^{(g)} - \omega^{*(g)}| |\widehat{\varepsilon}^{(g)}| + \max_{1 \leq g \leq nT} |\omega^{*(g)}| |\widehat{\varepsilon}^{(g)} - \varepsilon^{(g)}| = o_p(1).
\end{aligned} \tag{45}$$

This completes the proof for $\|\frac{1}{nT} \sum_{g=1}^{nT} \boldsymbol{E}_1^{(g)} \boldsymbol{E}_3^{(g)\top}\|_2 = o_p(1)$.

Next, we show $\|\frac{1}{nT} \sum_{g=1}^{nT} \boldsymbol{E}_2^{(g)} \boldsymbol{E}_4^{(g)\top}\|_2 = o_p(1)$. Using similar arguments in bounding $\|\widehat{\boldsymbol{\Sigma}} - \boldsymbol{\Sigma}\|_2$ in the proof of Lemma G.3, we obtain $\|\widehat{\boldsymbol{H}}_2 - \boldsymbol{H}_2\|_2 = o_p(1)$ as well as $\left\| \frac{1}{nT} \sum_{g=1}^{nT} \boldsymbol{m}^{*(g)} (\boldsymbol{m}^{*(g)})^\top - \mathbb{E}\left\{ \boldsymbol{m}^* (\boldsymbol{m}^*)^\top \right\} \right\|_2 = o_p(1)$.

It follows from $\widehat{\boldsymbol{\psi}} \xrightarrow{p} \boldsymbol{\psi}^*$ that $\frac{1}{nT} \sum_{g=1}^{nT} \left\| \widehat{\boldsymbol{m}}^{(g)} - \boldsymbol{m}^{*(g)} \right\|_2^2 \xrightarrow{p} 0$. Therefore,

$$\begin{aligned}
&\left\| \frac{1}{nT} \sum_{g=1}^{nT} \widehat{\boldsymbol{m}}^{(g)} (\boldsymbol{m}^{*(g)})^\top - \frac{1}{nT} \sum_{g=1}^{nT} (\boldsymbol{m}^{*(g)}) (\boldsymbol{m}^{*(g)})^\top \right\|_2 \\
&\leq \frac{1}{nT} \sum_{g=1}^{nT} \left\| \widehat{\boldsymbol{m}}^{(g)} (\boldsymbol{m}^{*(g)})^\top - \boldsymbol{m}^{*(g)} (\boldsymbol{m}^{*(g)})^\top \right\|_2 \leq \frac{1}{nT} \sum_{g=1}^{nT} \left\| \widehat{\boldsymbol{m}}^{(g)} - \boldsymbol{m}^{*(g)} \right\|_2 \left\| \boldsymbol{m}^{*(g)} \right\|_2 \\
&\leq \left( \frac{1}{nT} \sum_{g=1}^{nT} \left\| \widehat{\boldsymbol{m}}^{(g)} - \boldsymbol{m}^{*(g)} \right\|_2^2 \right) \left( \frac{1}{nT} \sum_{g=1}^{nT} \left\| \boldsymbol{m}^{*(g)} \right\|_2^2 \right) \xrightarrow{p} 0,
\end{aligned}$$

where the last step holds since the convergence of the diagonal elements of $\frac{1}{nT} \sum_{g=1}^{nT} \boldsymbol{m}^{*(g)} (\boldsymbol{m}^{*(g)})^\top$ implies that $\frac{1}{nT} \sum_{g=1}^{nT} \left\| \boldsymbol{m}^{*(g)} \right\|_2^2$ is bounded in probability. Based on the aforementioned results, we can show

$$\begin{aligned}
\left\| \frac{1}{nT} \sum_{g=1}^{nT} \boldsymbol{E}_2^{(g)} \boldsymbol{E}_4^{(g)\top} \right\|_2 &= \left\| \frac{1}{nT} \sum_{g=1}^{nT} \left\{ \widehat{\boldsymbol{H}}_2 \widehat{\boldsymbol{m}}^{(g)} (\boldsymbol{m}^{*(g)})^\top \boldsymbol{H}_2^\top - \boldsymbol{H}_2 \boldsymbol{m}^{*(g)} (\boldsymbol{m}^{*(g)})^\top \boldsymbol{H}_2^\top \right\} \right\|_2 \\
&\leq \left\| (\widehat{\boldsymbol{H}}_2 - \boldsymbol{H}_2) \left\{ \frac{1}{nT} \sum_{g=1}^{nT} \widehat{\boldsymbol{m}}^{(g)} (\boldsymbol{m}^{*(g)})^\top \right\} \boldsymbol{H}_2^\top \right\|_2 \\
&\quad + \left\| \boldsymbol{H}_2 \left\{ \frac{1}{nT} \sum_{g=1}^{nT} \widehat{\boldsymbol{m}}^{(g)} (\boldsymbol{m}^{*(g)})^\top - \boldsymbol{m}^{*(g)} (\boldsymbol{m}^{*(g)})^\top \right\} \boldsymbol{H}_2^\top \right\|_2 = o_p(1).
\end{aligned}$$

It remains to show $\|\frac{1}{nT} \sum_{g=1}^{nT} \boldsymbol{E}_1^{(g)} \boldsymbol{E}_4^{(g)\top}\|_2 = o_p(1)$ and $\|\frac{1}{nT} \sum_{g=1}^{nT} \boldsymbol{E}_2^{(g)} \boldsymbol{E}_3^{(g)\top}\|_2 = o_p(1)$. They can be shown in similar

ways, here we only prove $\|\frac{1}{nT} \sum_{g=1}^{nT} \boldsymbol{E}_1^{(g)} \boldsymbol{E}_4^{(g)\top}\|_2 = o_p(1)$ for brevity. Notice that

$$
\begin{aligned}
\left\| \frac{1}{nT} \sum_{g=1}^{nT} \boldsymbol{E}_1^{(g)} \boldsymbol{E}_4^{(g)\top} \right\|_2 &= \left\| \left\{ \frac{1}{nT} \sum_{g=1}^{nT} \left( \widehat{\omega}^{(g)} \widehat{\varepsilon}^{(g)} - \omega^{*(g)} \varepsilon^{(g)} \right) \boldsymbol{\xi}^{(g)} (\boldsymbol{m}^{*(g)})^\top \right\} \boldsymbol{H}_2^\top \right\|_2 \\
&\leq \left\| \frac{1}{nT} \sum_{g=1}^{nT} \left( \widehat{\omega}^{(g)} \widehat{\varepsilon}^{(g)} - \omega^{*(g)} \varepsilon^{(g)} \right) \boldsymbol{\xi}^{(g)} (\boldsymbol{m}^{*(g)})^\top \right\|_2 \|\boldsymbol{H}_2\|_2 \\
&\leq \frac{1}{nT} \sum_{g=1}^{nT} \left\| \left( \widehat{\omega}^{(g)} \widehat{\varepsilon}^{(g)} - \omega^{*(g)} \varepsilon^{(g)} \right) \boldsymbol{\xi}^{(g)} \right\|_2 \left\| \boldsymbol{m}^{*(g)} \right\|_2 \|\boldsymbol{H}_2\|_2 \\
&\leq \left( \frac{1}{nT} \sum_{g=1}^{nT} \left\| \left( \widehat{\omega}^{(g)} \widehat{\varepsilon}^{(g)} - \omega^{*(g)} \varepsilon^{(g)} \right) \boldsymbol{\xi}^{(g)} \right\|_2^2 \right) \left( \frac{1}{nT} \sum_{g=1}^{nT} \left\| \boldsymbol{m}^{*(g)} \right\|_2^2 \right) \|\boldsymbol{H}_2\|_2 .
\end{aligned}
$$

Since $\frac{1}{nT} \sum_{g=1}^{nT} \left\| \boldsymbol{m}^{*(g)} \right\|_2^2$ and $\|\boldsymbol{H}_2\|_2$ are bounded in probability, it suffices to show $\frac{1}{nT} \sum_{g=1}^{nT} \left\| \left( \widehat{\omega}^{(g)} \widehat{\varepsilon}^{(g)} - \omega^{*(g)} \varepsilon^{(g)} \right) \boldsymbol{\xi}^{(g)} \right\|_2^2 = o_p(1)$. We have already prove $\max_{1 \leq g \leq nT} |\widehat{\omega}^{(g)} \widehat{\varepsilon}^{(g)} - \omega^{*(g)} \varepsilon^{(g)}| = o_p(1)$ in (45), hence $\max_{1 \leq g \leq nT} (\widehat{\omega}^{(g)} \widehat{\varepsilon}^{(g)} - \omega^{*(g)} \varepsilon^{(g)})^2 = o_p(1)$. Meanwhile, by Lemma G.4, we have $\sup_{\boldsymbol{a} \in \mathbb{S}^{mL-1}} \frac{1}{nT} \sum_{g=1}^{nT} \boldsymbol{a}^\top \boldsymbol{\xi}^{(g)} \boldsymbol{\xi}^{(g)\top} \boldsymbol{a} = O_p(1)$. Thus,

$$
\sup_{\boldsymbol{a} \in \mathbb{S}^{mL-1}} \left| \frac{1}{nT} \sum_{g=1}^{nT} \boldsymbol{a}^\top \boldsymbol{\xi}^{(g)} \boldsymbol{\xi}^{(g)\top} \boldsymbol{a} \left( \widehat{\omega}^{(g)} \widehat{\varepsilon}^{(g)} - \omega^{*(g)} \varepsilon^{(g)} \right)^2 \right| = o_p(1),
$$

equivalently, $\frac{1}{nT} \sum_{g=1}^{nT} \left\| \left( \widehat{\omega}^{(g)} \widehat{\varepsilon}^{(g)} - \omega^{*(g)} \varepsilon^{(g)} \right) \boldsymbol{\xi}^{(g)} \right\|_2^2 = o_p(1)$. This completes the proof for $\|\frac{1}{nT} \sum_{g=1}^{nT} \boldsymbol{E}_1^{(g)} \boldsymbol{E}_4^{(g)\top}\|_2 = o_p(1)$. We can also show $\|\frac{1}{nT} \sum_{g=1}^{nT} \boldsymbol{E}_2^{(g)} \boldsymbol{E}_3^{(g)\top}\|_2 = o_p(1)$ using similar steps.

Combine these results together yields $\|\frac{1}{nT} \sum_{g=1}^{nT} \zeta^{(g)} (\widehat{\zeta}^{(g)} - \zeta^{(g)})^\top\|_2 = o_p(1)$. Similarly we can show $\|\frac{1}{nT} \sum_{g=1}^{nT} (\widehat{\zeta}^{(g)} - \zeta^{(g)}) \widehat{\zeta}^{(g)\top}\|_2 = o_p(1)$. Therefore, $\|\widehat{\boldsymbol{\Omega}}_{\text{IPW}} - \frac{1}{nT} \sum_{g=1}^{nT} \zeta^{(g)} \zeta^{(g)\top}\|_2 = o_p(1)$, and hence (43) is proven.

$\square$

### G.4. Proof of Theorem F.1

*Proof.* Similar to the proof of Theorem 4.5, we define $\varepsilon_{i,t}$ as follows:

$$
\varepsilon_{i,t} = R_{i,t+1} + \gamma \sum_{a \in \mathcal{A}} Q^\pi(S_{i,t+1}, a) \pi(a|S_{i,t+1}) - Q^\pi(S_{i,t}, A_{i,t}),
$$

and use $\mathcal{F}_t = \{(S_j, A_j, R_{j+1})\}_{0 \leq j < t} \cup \{S_t, A_t\}$ to denote the past information up to time $t$. Based on Assumption 3.1, 3.2, and Bellman equation, $\varepsilon_{i,t}$ satisfies $\mathbb{E}(\varepsilon_t|\mathcal{F}_t) = \mathbb{E}(\varepsilon_t|S_t, A_t) = 0$. For simplicity, we use $\widehat{\omega}_{\pi,i,t}$ and $\omega_{\pi,i,t}$ to represent

$\widehat{\omega}_{\pi,nT}(S_{i,t}, A_{i,t})$ and $\omega_{\pi}(S_{i,t}, A_{i,t})$ respectively. The value estimation error $\widehat{V}_{\mathrm{CC}}^{\pi}(\mathbb{G}) - V^{\pi}(\mathbb{G})$ can be decomposed as

$$
\begin{aligned}
&\widehat{V}_{\mathrm{CC}}^{\pi}(\mathbb{G}) - V^{\pi}(\mathbb{G}) \\
=&\frac{1}{1-\gamma}\frac{1}{nT}\sum_{i=1}^{n}\sum_{t=0}^{T-1}\eta_{i,t+1}\widehat{\omega}_{\pi,i,t}R_{i,t+1} - \mathbb{E}_{S_0\sim\mathbb{G}}\left\{\sum_{a\in\mathcal{A}}\pi(a|S_0)Q^{\pi}(S_0,a)\right\} \\
=&\frac{1}{1-\gamma}\frac{1}{nT}\sum_{i=1}^{n}\sum_{t=0}^{T-1}\eta_{i,t+1}\widehat{\omega}_{\pi,i,t}\left[\varepsilon_{i,t+1} + Q^{\pi}(S_{i,t},A_{i,t}) - \gamma\sum_{a\in\mathcal{A}}Q^{\pi}(S_{i,t+1},a)\pi(a|S_{i,t+1})\right] \\
&- \mathbb{E}_{S_0\sim\mathbb{G}}\left\{\sum_{a\in\mathcal{A}}\pi(a|S_0)Q^{\pi}(S_0,a)\right\} \\
=&-\frac{1}{1-\gamma}\mathcal{L}_{nT}(\widehat{\omega}_{\pi},Q^{\pi})\cdots\cdots\text{(I)} \\
&+\frac{1}{1-\gamma}\frac{1}{nT}\sum_{i=1}^{n}\sum_{t=0}^{T-1}\eta_{i,t+1}\widehat{\omega}_{\pi,i,t}\varepsilon_{i,t+1}\cdots\cdots\text{(II)}
\end{aligned}
$$

For (I), note that $\mathcal{L}_{nT}(\widehat{\omega}_{\pi}, Q^{\pi})$ captures the difference between two sides of equation (8) under the estimated density ratio and the true Q-function. This loss term is dependent on the specific algorithm, the choice of function class $\mathcal{Q}$, and the computation procedure. Under Assumption F.1(b), this term converges to 0. For (II), by applying the Cauchy inequality and the boundedness condition in Assumption F.1(a), we have

$$
\begin{aligned}
&\mathbb{E}\left\{\frac{1}{nT}\sum_{i=1}^{n}\sum_{t=0}^{T-1}\eta_{i,t+1}\widehat{\omega}_{\pi,i,t}\varepsilon_{i,t}\right\}^2 \\
\leq&\mathbb{E}\left(\frac{1}{nT}\sum_{i=1}^{n}\sum_{t=0}^{T-1}\widehat{\omega}_{\pi,i,t}^2\right)\left(\frac{1}{nT}\sum_{i=1}^{n}\sum_{t=0}^{T-1}\eta_{i,t+1}^2\varepsilon_{i,t}^2\right) \\
\leq&c_{\omega}^2\cdot\mathbb{E}\left(\frac{1}{nT}\sum_{i=1}^{n}\sum_{t=0}^{T-1}\eta_{i,t+1}^2\varepsilon_{i,t}^2\right).
\end{aligned}
\tag{46}
$$

The last inequality follows from the boundedness of $\widehat{\omega}_{\pi,i,t}$.

Next, we derive the bound for $\mathbb{E}\left((nT)^{-1}\sum_{i=1}^{n}\sum_{t=0}^{T-1}\eta_{i,t+1}^2\varepsilon_{i,t}^2\right)$. Under the MAR assumption, $\eta_{t+1}$ and $\varepsilon_t$ are conditionally independent, it follows that

$$
\mathbb{E}\{\eta_{t+1}\varepsilon_t\} = \mathbb{E}\{\mathbb{E}(\eta_{t+1}\varepsilon_t|\mathcal{F}_t,\eta_t)\} = \mathbb{E}\{\mathbb{E}(\eta_{t+1}|\mathcal{F}_t,\eta_t)\mathbb{E}(\varepsilon_t|\mathcal{F}_t)\} = \mathbf{0}.
$$

Similarly, for any $0 \leq t_1 < t_2 < T$, we obtain $\mathbb{E}\{\eta_{t_1+1}\eta_{t_2+1}\varepsilon_{t_1}\varepsilon_{t_2}\} = 0$. In addition, by the independence assumption among trajectories, we have

$$
\mathbb{E}\{\eta_{i_1,t_1+1}\eta_{i_2,t_2+1}\varepsilon_{i_1,t_1}\varepsilon_{i_2,t_2}\} = 0.
$$

Applying the bound for $\varepsilon_t$ derived from (13) yields

$$
\begin{aligned}
\mathbb{E}\left\{\frac{1}{nT}\sum_{i=1}^{n}\sum_{t=0}^{T-1}\eta_{i,t+1}\varepsilon_{i,t}\right\}^2 &= \frac{1}{(nT)^2}\sum_{i=1}^{n}\sum_{t=0}^{T-1}\mathbb{E}\{\eta_{i,t+1}^2\varepsilon_{i,t}^2\} = \frac{1}{(nT)^2}\cdot n\sum_{t=0}^{T-1}\mathbb{E}\{\eta_{t+1}^2\varepsilon_t^2\} \\
&\leq \frac{1}{nT}(c_0 + 2c')^2.
\end{aligned}
\tag{47}
$$

Combine (46) and (47), we have

$$
\mathbb{E}\left\{\frac{1}{nT}\sum_{i=1}^{n}\sum_{t=0}^{T-1}\eta_{i,t+1}\widehat{\omega}_{\pi,i,t}\varepsilon_{i,t}\right\}^2 \leq \frac{1}{nT}(c_0 + 2c')^2 c_{\omega}^2.
$$

By Markov inequality,

$$\frac{1}{nT} \sum_{i=1}^{n} \sum_{t=0}^{T-1} \eta_{i,t+1} \widehat{\omega}_{\pi,i,t} \varepsilon_{i,t} = O_p\{(nT)^{-1/2}\}.$$

As a result, $\widehat{V}_{\mathrm{CC}}^{\pi}(\mathbb{G}) \xrightarrow{p} V^{\pi}(\mathbb{G})$ as $nT \to \infty$, indicating that $\widehat{V}_{\mathrm{CC}}^{\pi}(\mathbb{G})$ is a consistent estimator of $V^{\pi}(\mathbb{G})$ under ignorable missingness.

However, when the missingness is nonignorable, the conditional independence between $\eta_{t+1}$ and $\varepsilon_t$ no longer holds. As a result, the convergence of (II) to 0 is not guaranteed, and the complete-case value estimator $\widehat{V}_{\mathrm{CC}}^{\pi}(\mathbb{G})$ will be biased from $V^{\pi}(\mathbb{G})$. $\qquad\square$

### G.5. Proof of Lemma G.8

**Lemma** Suppose Assumption A.1-A.2 holds. We have $\|\widehat{\Sigma}_{\mathrm{IPW}} - \Sigma\|_2 = O_p\left\{L^{1/2}(nT)^{-1/2}\log(nT)\right\}$, $\|\widehat{\Sigma}_{\mathrm{IPW}}^{-1} - \Sigma^{-1}\|_2 = O_p\left\{L^{1/2}(nT)^{-1/2}\log(nT)\right\}$ and $\|\widehat{\Sigma}_{\mathrm{IPW}}^{-1}\|_2 \le 6\bar{c}^{-1}$ with probability approaching 1, as either $n \to \infty$ or $T \to \infty$.

*Proof.* Recall that

$$\widehat{\Sigma}_{\mathrm{IPW}} = \frac{1}{nT} \sum_{i=1}^{n} \sum_{t=0}^{T-1} \widehat{\omega}_{i,t+1} \boldsymbol{\xi}_{i,t} \left(\boldsymbol{\xi}_{i,t} - \gamma \boldsymbol{U}_{\pi,i,t+1}\right)^{\top}$$

$$\Sigma = \mathbb{E}\left\{\omega_{i,t+1}^{*} \boldsymbol{\xi}_{i,t}(\boldsymbol{\xi}_{i,t} - \gamma \boldsymbol{U}_{\pi,i,t+1})\right\} = \mathbb{E}\left\{\boldsymbol{\xi}_{i,t}(\boldsymbol{\xi}_{i,t} - \gamma \boldsymbol{U}_{\pi,i,t+1})\right\}.$$

It follows that

$$\widehat{\Sigma}_{\mathrm{IPW}} - \Sigma = \frac{1}{nT} \sum_{i=1}^{n} \sum_{t=0}^{T-1} \widehat{\omega}_{i,t+1} \boldsymbol{\xi}_{i,t} \left(\boldsymbol{\xi}_{i,t} - \gamma \boldsymbol{U}_{\pi,i,t+1}\right)^{\top} - \Sigma$$

$$= \left(\frac{1}{nT} \sum_{i=1}^{n} \sum_{t=0}^{T-1} (\widehat{\omega}_{i,t+1} - \omega_{i,t+1}^{*}) \boldsymbol{\xi}_{i,t} \left(\boldsymbol{\xi}_{i,t} - \gamma \boldsymbol{U}_{\pi,i,t+1}\right)^{\top}\right)$$

$$+ \left(\frac{1}{nT} \sum_{i=1}^{n} \sum_{t=0}^{T-1} \omega_{i,t+1}^{*} \boldsymbol{\xi}_{i,t} \left(\boldsymbol{\xi}_{i,t} - \gamma \boldsymbol{U}_{\pi,i,t+1}\right)^{\top} - \Sigma\right)$$

Using similar arguments in proving Lemma G.3, we obtain

$$\left\|\frac{1}{nT} \sum_{i=1}^{n} \sum_{t=0}^{T-1} \omega_{i,t+1}^{*} \boldsymbol{\xi}_{i,t} \left(\boldsymbol{\xi}_{i,t} - \gamma \boldsymbol{U}_{\pi,i,t+1}\right)^{\top} - \Sigma\right\|_2 = O_p\left\{L^{1/2}(nT)^{-1/2}\log(nT)\right\}.$$

On the other hand, using a similar technique as in (24), we have

$$\frac{1}{\sqrt{nT}} \sum_{i=1}^{n} \sum_{t=0}^{T-1} \left(\widehat{\omega}_{i,t+1} - \omega_{i,t+1}^{*}\right) \boldsymbol{\xi}_{i,t} \left(\boldsymbol{\xi}_{i,t} - \gamma \boldsymbol{U}_{\pi,i,t+1}\right)^{\top} = \sqrt{nT} \boldsymbol{H}_3(\widehat{\boldsymbol{\psi}} - \boldsymbol{\psi}^{*}) + o_p(1),$$

for some tensor $\boldsymbol{H}_3$. By convergence of $\widehat{\boldsymbol{\psi}}$ given in (26), we have

$$\left\|\frac{1}{nT} \sum_{i=1}^{n} \sum_{t=0}^{T-1} \left(\widehat{\omega}_{i,t+1} - \omega_{i,t+1}^{*}\right) \boldsymbol{\xi}_{i,t} \left(\boldsymbol{\xi}_{i,t} - \gamma \boldsymbol{U}_{\pi,i,t+1}\right)^{\top}\right\|_2 = O_p\{(nT)^{-1/2}\}.$$

Therefore,

$$
\begin{aligned}
\left\| \widehat{\boldsymbol{\Sigma}}_{\mathrm{IPW}} - \boldsymbol{\Sigma} \right\|_2 &\leq \left\| \frac{1}{nT} \sum_{i=1}^{n} \sum_{t=0}^{T-1} \omega_{i,t+1}^* \boldsymbol{\xi}_{i,t} \left( \boldsymbol{\xi}_{i,t} - \gamma \boldsymbol{U}_{\pi,i,t+1} \right)^\top - \boldsymbol{\Sigma} \right\|_2 \\
&\quad + \left\| \frac{1}{nT} \sum_{i=1}^{n} \sum_{t=0}^{T-1} \left( \widehat{\omega}_{i,t+1} - \omega_{i,t+1}^* \right) \boldsymbol{\xi}_{i,t} \left( \boldsymbol{\xi}_{i,t} - \gamma \boldsymbol{U}_{\pi,i,t+1} \right)^\top \right\|_2 \\
&= O_p \left\{ L^{1/2} (nT)^{-1/2} \log(nT) \right\} + O_p \{ (nT)^{-1/2} \} \\
&= O_p \left\{ L^{1/2} (nT)^{-1/2} \log(nT) \right\}.
\end{aligned}
$$

Based on this result, we can follow similar steps in the proof for Lemma G.3 to show $\|\widehat{\boldsymbol{\Sigma}}_{\mathrm{IPW}}^{-1}\|_2 \leq 6\bar{c}^{-1}$ and $\|\widehat{\boldsymbol{\Sigma}}_{\mathrm{IPW}}^{-1} - \boldsymbol{\Sigma}^{-1}\|_2 = O_p \left\{ L^{1/2}(nT)^{-1/2} \log(nT) \right\}$. The proof is hence completed.

$\square$

