# OpenReview forum: "Off-Policy Evaluation under Nonignorable Missing Data"
_ICML.cc/2025/Conference — ICML 2025 poster_

### Official Review · Reviewer_59Sd · 2025-03-11

**Overall Recommendation:** 4

**Summary:**

The authors study and propose OPE for RL under monotone MNAR missing.
Specifically, they construct an IPW-based correction of value-based OPE and show that, unlike an uncorrected method, the proposed method is unbiased with MNAR missing process under the existence of a shadow variable.
They also conducted synthetic/real/semi-real numerical experiments to verify the effectiveness of the proposed method.

**Claims And Evidence:**

Yes.

**Essential References Not Discussed:**

None that I am aware of.

**Experimental Designs Or Analyses:**

Yes.

**Methods And Evaluation Criteria:**

Yes.

**Other Comments Or Suggestions:**

- L110: $p:S\times A\times S\to [0,1]$ only makes sense with discrete state space. Need some update.
- Please provide the definitions of B-spline and wavelet for the completeness.

**Other Strengths And Weaknesses:**

It is a solid contribution to the OPE literature and deserves to be accepted, as far as I can tell.

One thing I have noticed is that the Assumption A.1 is a bit unorganized with mixed levels of details. Can you factor out the details in (a), (b) and (e)?

**Questions For Authors:**

It seems Assuption A.1 (d) (i) poses a concentrability condition on the feature map $\Phi_L$,
but how easy/difficult is it to satisfy this condition, especially if $\Phi_L$ is B-spline or wavelet?
Since $\Phi_L$ is fixed to one of these two, can you translate the condition in terms of the behavior distribution, e.g., to the boundedness of $\mu$ from below?

**Relation To Broader Scientific Literature:**

It provides a solution to a practical problem related to missing values that OPE methods will be faced with.

**Theoretical Claims:**

Partly yes.

---

> ### Author Rebuttal · Authors · 2025-04-01
>
> Thanks for your thoughtful questions and the time you spent reviewing our paper. We really appreciate your insights and are happy to discuss any further ideas or questions you may have.
>
> **Regarding Assumptions (a), (b), and (e):**
>
> Below is a more detailed explanation of what each assumption means and how it is used in the proof. We hope this will help improve the clarity of the assumptions, and we will incorporate these explanations in the final version of our paper.
>
> * Assumption (a):
> We assume smoothness of the transition kernel $\mathcal{P}$. Specifically, when $0 < p \leq 1$, $\lfloor p \rfloor = 0$, and the condition becomes equivalent to assuming $h$ satisfies $\sup_{x, y} \frac{|h(x) - h(y)|}{\\|x - y\\|_2^p} \leq c$, which is a form of Hölder continuity. Under this assumption, it can be shown that there exists a constant $c' > 0$ such that $Q(\pi; \cdot, a) \in \Lambda(p, c')$ for any policy $\pi$ and $a \in \mathcal{A}$. This ensures that the $Q$-function has bounded derivatives up to order $\lfloor p \rfloor$, which is critical when deriving inference for the value function.
> * Assumption (b):
> This assumption is more of a claim or explanation rather than a strict assumption. Here we consider two types of basis functions, which are commonly used sieve basis functions. These are standard choices for such problems and serve to simplify the analysis while ensuring general applicability.
> * Assumption (e): Here, $L$ controls the smoothness of the basis function, which in turn determines how closely the linear sieve basis function can approximate the true function. This assumption is used to ensure the $Q$ function is well approximated. In the proof of asymptotic normality, we rely on this condition on $L$ to establish that $\sup_{s \in \mathcal{S}, a \in \mathcal{A}} |Q(\pi; s, a) - \Phi_L^\top(x) \beta^*_{\pi, a}| = O(L^{-p/d})$, which guarantees the consistency and asymptotic behavior of $\hat{\beta}$ and the value estimates.
>
> **Regarding "Other Comments Or Suggestions”:**
> * The reviewer mentioned that *Line 110 is only valid for a discrete state space*. We would like to clarify that our system does not impose any constraints on whether the state space is discrete or continuous. In fact, we explicitly allow a multi-dimensional continuous state space, i.e., $S\in \mathbb{R}^d$, as stated in Line 115-116. We recognize that the definition in Line 110 might be somewhat misleading, as $p$ actually refers to the transition kernel in the MDP and is not a probability mass function. We believe it would be clearer to revise the current statement to $p:\mathcal{S} \times\mathcal{A}\rightarrow\mathcal{S}$, and we hope this revision will help clarify the matter.
> * We sincerely appreciate the reviewer's feedback, and will include detailed definitions of B-splines and wavelets in the final version of our paper for completeness.
>
> **Regarding Assuption A.1 (d) (i):**
>
> We appreciate the reviewer’s question regarding the ease of satisfying this assumption. A special case that may help clarify the strength of this assumption is when $\pi$ is deterministic, $b$ is the $\epsilon$-greedy policy with respect to $\pi$ satisfying $\epsilon \leq 1 - \gamma^2$, and $\mu=\nu_0$. In this case, Assumption A.1 (d)(i) can be shown to be naturally satisfied (see Sec C.1 of the supplementary material of Shi et al. (2021b) for proof details).
>
> We recognize that this is a somewhat non-intuitive assumption, and the verification may require some calculations. We also appreciate the reviewer’s suggestion to explore whether and how this assumption could be simplified by incorporating the specific form of $\Phi$ and the moving behavior distribution $b(a|s)$. Checking whether the LHS of this assumption is positive-definite (or finding an easier-to-understand sufficient condition for it) essentially requires us to quantify the positive definiteness of the matrix part, i.e., $\xi\xi^\top -\gamma^2 u_\pi u_\pi^\top$. The multiplication by $b(a|s)$ and $\mu(s)$ is more of a weighting method to determine whether the matrix, after taking the expectation, remains positive-definite, which is not the main focus. Thus, deriving a boundedness assumption for $\mu$ from below seems implausible. We would greatly appreciate any further thoughts or ideas you may have on this aspect and look forward to your suggestions.
>
> Once again, we sincerely appreciate your time reviewing our paper and welcome any further questions or discussions.

---

### Official Review · Reviewer_EbSe · 2025-03-13

**Overall Recommendation:** 3

**Summary:**

The paper analyzes the problem of policy evaluation in the presence of missing data. The authors distinguish between two types of missing data:
- **Missing at Random (MAR):** Data is missing independently of unobserved factors.
- **Missing Not at Random (MNAR):** Data is missing due to a hidden cause, introducing bias in the value function and preventing the actor from completing the trajectory.

### Methodology

To address the policy evaluation problem under MNAR conditions, the authors propose an importance sampling (IS) correction method. Their estimator, **V-IPW**, is shown theoretically to be consistent (converging to the true value) and unbiased, whereas the standard estimator **V-CC** suffers from bias.

The importance sampling estimator relies on a **dropout propensity parameter**, which represents the probability of data being missing given the current state. The authors also propose a principled method to estimate this parameter when it is not known a priori.

### Experimental Evaluation

The authors evaluate their estimator, **V-IPW**, against the baseline **V-CC** in two different settings:

1. **Simulated Environment:**
   - The dropout propensity is known.
   - A single target policy is evaluated.

2. **Real-World Environment:**
   - The dropout propensity is unknown.
   - Four different target policies are evaluated: Behavior, DQN, Dueling DQN, and BCQ, all learned via offline reinforcement learning (RL) algorithms.

### Results

The experimental results demonstrate that the **V-IPW estimator** corrects the bias introduced in **V-CC**, effectively compensating for the dropout effect.

**Claims And Evidence:**

While the theoretical analysis of **V-IPW** and **V-CC** is comprehensive and well-structured, the empirical evaluation is somewhat limited.

In the **real-world environment**, there is no ground truth available for the value function. As a result, the comparison between **V-IPW** and **V-CC** relies on the assumption that **V-CC is negatively biased**—but the extent of this bias remains unclear. A more detailed analysis quantifying the potential bias in **V-CC** would strengthen the empirical claims.

**Essential References Not Discussed:**

The related work section provides a comprehensive overview of the literature on off-policy evaluation.

**Experimental Designs Or Analyses:**

The experiments demonstrate the effectiveness of the proposed algorithm. However, there are several limitations in the evaluation:

- **Low-Dimensional Settings:** All experiments are conducted in low-dimensional environments, making it unclear how well the method generalizes to more complex, high-dimensional settings.
- **Propensity Accuracy Analysis:** The synthetic experiments lack an analysis of the accuracy of the estimated propensity scores, which is crucial for understanding the reliability of the importance sampling correction.
- **Limited Environment Diversity:** The algorithm is tested on only a few environments, making it difficult to assess its robustness and general applicability. A broader evaluation across diverse environments would strengthen the claims.
- **Analysis of the important samples requirements:** The algorithm relies on the importance sampling to add weight to rare samples, however, an analysis of how many such samples are needed is missing, this analysis has significant ramifications on the applicability of the proposed algorithm.

**Methods And Evaluation Criteria:**

The purpose method makes sense as Importance sampling is widely used in the RL literature.

**Other Comments Or Suggestions:**

No

**Other Strengths And Weaknesses:**

### Strengths
1. An algorithm that tackles an issue that was not referenced in the literature before
2. The theoretical analysis and presentation is thorough
### Weaknesses

1. Limited Scope of the Algorithm:
   - The proposed method focuses on bias correction in policy evaluation but does not claim to improve the convergence rate over **V-CC**.
   - Enhancing convergence could potentially reduce data collection constraints, making the approach more practical for real-world applications.

2. Dropout Propensity Estimation is Deferred to the Appendix:
   - The estimation of **dropout propensity** is a critical component for applying the algorithm in real-world settings.
   - Given its importance, this section should be discussed in greater detail in the main text rather than being relegated to the appendix.

**Questions For Authors:**

No

**Relation To Broader Scientific Literature:**

The reinforcement learning setting and problems discussed in this paper are closely related to causal machine learning and uplift modeling.

**Theoretical Claims:**

The claims and arguments in the theoretical analysis of V-IPW and V-CC make a lot of sense, but I did not validate the correctness of their proofs.

---

> ### Author Rebuttal · Authors · 2025-04-01
>
> **Response to "Summary-Claims And Evidence”:**
>
> The reviewer raised a concern that *the comparison between V-IPW and V-CC relies on the assumption that V-CC is negatively biased.* Actually, Our real-world experiment consists of two parts: the first (Table 2) is based on the original sepsis dataset, while the second (Table 3) is derived from a quasi-real dataset where we controlled only the dropout hazard function, ensuring it follows a known form. We would like to emphasize the following points:
> * In the offline dataset, where the true dropout pattern is unknown, there is no ground truth for the true value function under no dropout. This means we cannot directly assess the bias in V-CC or definitively validate the extent to which V-IPW corrects it. Our inference in Table 2 is based on common knowledge and the intuition that V-CC may underestimate the value function due to early discharge, though we acknowledge that this inference lacks definitive empirical support. We appreciate the reviewer’s attention to this issue, which motivated our quasi-real data analysis using MIMIC-III, as presented in Table 3.
> * In Table 3, we explicitly controlled the dropout hazard model using the function $\lambda(\cdot)$, specified above Sec 7, which follows a functional form inspired by prior MIMIC-III studies (Kramer & Zimmerman, 2010; McWilliams et al., 2019). This setup allows us to obtain an unbiased estimate of the value function under no dropout (reported in the first row), serving as a ground truth for comparison. The results in Table 3 consistently show that V-IPW outperforms V-CC in correcting the underestimation bias caused by missing not at random (MNAR) dropout.
>
> **Response to "Summary-Experimental Designs Or Analyses”:**
>
> 1. Regarding the "Propensity Accuracy Analysis", we have provided additional support to validate the accuracy of the dropout model estimation in https://anonymous.4open.science/r/OPE_MNAR_ICML_rebuttal-832D/, where the first plot shows small MSEs close to zero, and the second plot compares the estimated parameters to their true values. Additionally, as all V-IPW experiments under MNAR require prior propensity score estimation, Figure 2 and the last two rows of Table 1 (MNAR (P) and MNAR (SP)) in the main paper confirm the accuracy of these estimations through bias, standard error, and empirical coverage probability. Without stable estimation results, the statistical properties of the value function would not hold. We hope the added accuracy analysis regarding the propensity score, along with the existing tables and figures in the main paper, will help address your concerns regarding the dropout function estimation performance.
>
> 2. Regarding the "Low-Dimensional Settings" and "Limited Environment Diversity": we sincerely appreciate the reviewer's suggestion on trying more diverse simulation settings and environments to possibly test the wider applicability of our approach. Due to time limit, we might not able to provide another group of setting and detailed analysis during this rebuttal time, but we will incorporate more settings to the final version of our paper.
>
> 3. Regarding the "Analysis of the important samples requirements": We appreciate the reviewer's question regarding the sample requirements in order to guarantee a stable estimate of the value function. Based on the asymptotic results in Theorem 4.7, the sample size requirement actually can be derived with a few steps. Here due to space limit we provide a brief analysis about the requirement, and we will add it to the final version of our paper to help with it. As $\hat{V}^{\pi}\sim \mathcal{N}(V^{\pi},\hat{\sigma}^2/(nT))$ (for simplicity we omit $\mathbb{G}$ in parenthesis and the subcript 'IPW'),  applying Gaussian tail bounds we have $\mathbb{P}(|\hat{V}^{\pi}-V^{\pi}|\geq \epsilon)\leq 2\exp\\{-nT\epsilon^2/(2\sigma^2)\\}$. Therefore, to achieve an error bound $\epsilon$ with probability at least $1-\delta$, the required sample size would be $n\geq \log(2/\delta)\cdot 2\sigma^2_{\pi}/(\epsilon^2T)$. For example, by setting $\delta = \epsilon = 0.01$, $T = 100$, and $\sigma^2_{\pi} = 0.5$ (a value comparable to the simulation setting), we find that $n \geq 530$ trajectories would suffice.
>
> **Response to Weakness 1:**
>
> Although we did not explicitly state the convergence rate of V-IPW, it is provided by the asymptotic normality result in Theorem 4.7, which shows that $\hat{V}^{\pi}$ converges to the true value at a rate of $O(n^{-1/2}T^{-1/2})$. This ensures fast bi-directional convergence in both $n$ and $T$, and this rate matches the parametric convergence rate, leaving no room to further improve the order of convergence.
>
> **Response to Weakness 2:**
>
> Thank you for your suggestion. We agree and will reintegrate the dropout propensity section into the main text to enhance readability and clarity.
>
> Once again, we appreciate your time and effort in reviewing our paper and look forward to your questions for further discussion.

---

### Official Review · Reviewer_JCcV · 2025-03-18

**Overall Recommendation:** 4

**Summary:**

This paper studies OPE when trajectories are truncated/missing and the missingness is non-ignorable. A new estimator based on inverse probability weighting is proposed, with theoretical justification for its unbiasedness and consistency properties. Experiments were conducted on a synthetic and a semi-synthetic problem to show how proposed approach compares with baseline approach that assumes complete data.

## Update after rebuttal
I am maintaining my already positive recommendation.

**Claims And Evidence:**

Yes.

**Essential References Not Discussed:**

Not to my knowledge.

Other suggested citations:
- Ji et al. Trajectory Inspection: A Method for Iterative Clinician-Driven Design of Reinforcement Learning Studies. AMIA 2021. This paper also investigated the early discharge / early termination issue in MIMIC dataset.

**Experimental Designs Or Analyses:**

Yes, the experiments are described with necessary details.

**Methods And Evaluation Criteria:**

Yes.

**Other Comments Or Suggestions:**

None. See comments above.

**Other Strengths And Weaknesses:**

- Overall the paper is well written.
- Perhaps in the introduction it should be made clearer what type of missingness is being addressed, especially in an MDP setting. L14-right "offline data is often incomplete due to different types of missingness" - at this point, it's unclear what is missing, is it the states/observations, actions, or rewards, or some combination.

**Questions For Authors:**

None. See comments above.

**Relation To Broader Scientific Literature:**

This paper addresses an important issue in OPE on real data which is often overlooked by assuming complete data.

As it touches on multiple areas, terminologies from different areas are being used. For example, MAR/MNAR, ignorable/nonignorable misingness. I think it is important to clarify these upfront, and also acknowledge other possible interpretations of the same phenomenon: e.g. trajectory truncation (as an alternative phrase for monotone missingness), or censoring (from survival analysis, example would be early ICU discharge).

**Theoretical Claims:**

I looked at the main text theorems and they appear to make sense. I did not read Sec 4.3 shadow variables carefully as I am not too familiar with that concept.

---

> ### Author Rebuttal · Authors · 2025-04-01
>
> Thanks for your thoughtful questions and the time you spent reviewing our paper. We really appreciate your insights and are
> happy to discuss any further ideas or questions you may have.
>
> **Clarification on Terminology:**
>
> We sincerely appreciate the reviewers' feedback on clarifying terminology across different fields, particularly regarding the comments on *Relation to Broader Scientific Literature*. We will provide thorough definitions of concepts such as monotone missingness and censoring in the introduction or at their first occurrence to enhance readability and coherence throughout the paper.
>
> **Regarding Essential References Not Discussed:**
>
> Thank you for highlighting the related work by Ji et al. (2021) on early discharge in the MIMIC-III dataset. After a careful review, we notice that their study primarily applies RL methods to identify potential issues with the existing vasopressor treatment policy. Notably, part of their findings aligns well with our problem motivation, which demonstrates that early discharge can introduce significant modeling bias, thereby impacting evaluation. We will incorporate this reference into the introduction, as it further substantiates the motivation behind our work.
>
>
> **Clarification on Types of Missingness in the Introduction:**
>
> We acknowledge that Line 14 may be somewhat vague in discussing different types of missingness, which could be unclear to readers. Here, the missingness types (as introduced in the subsequent paragraph) include both MAR and MNAR, specifically concerning the observation status of the reward being evaluated. Within the RL framework, this missingness affects both the reward and next-state, denoted as $(R_{t+1}, S_{t+1})$. Given the presence of non-ignorable missingness, the absence of $(R_{t+1}, S_{t+1})$ naturally leads to cascading missingness in subsequent state-action-reward sequences. While a detailed explanation may not be feasible in the introduction, we will clarify in Line 14 that the missingness primarily pertains to the reward in general settings. We hope this could ensure greater clarity for readers.
>
> **Finally**, We sincerely appreciate your time and effort in reviewing our paper and look forward to your questions and further discussion.

---

### Official Review · Reviewer_FCgQ · 2025-04-06

**Overall Recommendation:** 2

**Summary:**

The paper proposes the challenge of non-ignorable missing data in policy evaluation in reinforcement learning, which is the type of missingness that has dependency with the reward and next state value. The intuition for the significance of the problem and its applications in practice is well explained. Moreover, a method for estimating the value function in the presence of such an issue is proposed, and its unbiasedness is proved. The method's effectiveness is also justified by numerical and practical experiments. The main idea is fairly simple and easy to understand, and appears effective in the case of the non-ignorable missingness.

**Claims And Evidence:**

The major claims of the study are as follows:
1. The concept of ignorable/non-ignorable missingness in OPE and the unbiasedness of the simple Monte-Carlo average for value estimation in the case of ignorable missingness.
2. A value estimation method that remains unbiased even when the missing data is non-ignorable.
3. Unbiasedness and asymptotic distribution of the proposed estimation.
4. Empirical experiments on both synthetic and real-world datasets that support the effectiveness of the proposed method.
5. There are claims for *traditional* OPE methods compared to the proposed method, such as the biasedness in MNAR and unbiasedness in MAR. *The traditional methods* in the paper, however, are limited to CC. There is a vast range of off-policy evaluation methods in the literature, at least 7 of them I'm aware of (PM, ES, IX, OS, LSE, IPS-TR, LS). I suggest that the authors explicitly define that their analysis and comparison on only on the CC and IPW(proposed) methods.

**Essential References Not Discussed:**

The following paper investigates missing-not-at-random data in ope.

[1] Takahashi, Tatsuki, Chihiro Maru, and Hiroko Shoji. "Off-Policy Evaluation for Recommendations with Missing-Not-At-Random Rewards." arXiv preprint arXiv:2502.08993 (2025).

[2] Yang, Longqi, et al. "Unbiased offline recommender evaluation for missing-not-at-random implicit feedback." Proceedings of the 12th ACM conference on recommender systems. 2018.

There are also other papers that investigate missing-not-at-random data like the following,

[3] Wang, Zifeng, et al. "Information theoretic counterfactual learning from missing-not-at-random feedback." Advances in Neural Information Processing Systems 33 (2020): 1854-1864.

**Experimental Designs Or Analyses:**

The comparison is limited to CC method, which is a very trivial and naive method. Other OPE methods should also be included in the experiments. These are some, but not all of them:

[1] Dudík, Miroslav, John Langford, and Lihong Li. "Doubly robust policy evaluation and learning." arXiv preprint arXiv:1103.4601 (2011).

[2] Metelli, Alberto Maria, Alessio Russo, and Marcello Restelli. "Subgaussian and differentiable importance sampling for off-policy evaluation and learning." Advances in neural information processing systems 34 (2021): 8119-8132.

[3] Behnamnia, Armin, et al. "Batch Learning via Log-Sum-Exponential Estimator from Logged Bandit Feedback". ICML 2024 Workshop: Aligning Reinforcement Learning Experimentalists and Theorists, 2024, https://openreview.net/forum?id=dT6pUWzSZM.

[4] Wang, Yu-Xiang, Alekh Agarwal, and Miroslav Dudık. "Optimal and adaptive off-policy evaluation in contextual bandits." International Conference on Machine Learning. PMLR, 2017.

[5] Sakhi, Otmane, et al. "Logarithmic smoothing for pessimistic off-policy evaluation, selection and learning." arXiv preprint arXiv:2405.14335 (2024).

**Methods And Evaluation Criteria:**

The evaluation in synthetic experiments is based on ECP and MSE, which are standard and appropriate.
For the real-world experiments, the real value function is not known, and the true value is hence not identifiable. This is especially more problematic in the presence of non-ignorable missing data. Therefore, in the setting provided in these experiments, a discussion on the estimated value is apparently the most that we can do.
I'm not sure if it is possible or not, but an experimental setting where the missingness happened naturally (not synthetically), but the best policy is trivial, and hence the true value function is known, can be added to provide a more confident evidence of the model's effectiveness on arbitrary, complex, unknown missed-data mechanisms.

**Other Comments Or Suggestions:**

I stated my suggestions in the Strengths and Weaknesses section.

**Other Strengths And Weaknesses:**

1. There is an assumption that the first sample of the trajectory is never missing. This limits the application of the proposed method in OPE for the bandits, as a bandit problem is an RL problem with trajectories of length 1. So, the method cannot be used for missing data in the bandit learning and evaluation problems.
2. There are some implicit assumptions at are not discussed. They should be implicitly stated as assumption, and justified by evidence and discussion, these are the ones that I found doubtful, implicit and ignored.
* uniform boundedness on reward values
* Action-state linear separability in the model of the Q-function.
3. The proposed challenge is very significant, and the proposed method is also intuitive and easy to accept as a working method. There are also some initial theoretical findings and base experiments. However, it appears that it has much work to do to become a complete study that meets ICML standards.
  * Comprehensive comparisons with the OPE methods in the literature are required.
  * The form of the missing mechanism can be investigated, and more real-world experiments with complex, unknown missing data procedures should be added that can be actually validated. Also, more settings for synthetic experiments can make the paper richer and the claims more persuasive.
  * The idea of IPW is used in many different contexts in OPE and OPL (for example, in average treatment effect literature), hence either the extensive practical or theoretical analysis can solidify the contributions of the paper.
  * A learning-based model-free method is necessary for large-scale applications and complex environments.
  * The estimation of $\lambda$, the dropout probability, is not fully handled, and it's not clear how it can be done in an arbitrary problem. It needs more investigation and analysis. Also, theoretical analysis in the presence of estimated, inaccurate $\lambda$ should be added to justify the method in practice.

**Questions For Authors:**

1. What is the advantage of the proposed method compared to other methods that handle missing data, such as the following,

[1] Takahashi, Tatsuki, Chihiro Maru, and Hiroko Shoji. "Off-Policy Evaluation for Recommendations with Missing-Not-At-Random Rewards." arXiv preprint arXiv:2502.08993 (2025).

**Relation To Broader Scientific Literature:**

Missing data in OPE is a key challenge and in practice, it occurs very often. In the medical field, the fact that patients' status is mostly not completely tracked makes almost every subject (trajectory of treatment) incomplete. In recommendation systems, the delayed reward observation is the major source of missing data.

**Theoretical Claims:**

The theoretical claims are sound and proven mathematically, and I don't find any fault in this part.

---

### Decision · Program_Chairs · 2025-05-01

**Decision:**

Accept (poster)

**Comment:**

This paper investigates a novel case in off-policy evaluation where non-ignorable missingness can occur (missing-not-at-random). Initially the paper received three positive reviews where the reviewers praised the significance of the contribution, both in terms of the problem studied as well as the presented solution. There were some minor concerns which were largely resolved during the author-reviewer discussion phase. Later near the end of the discussion phase, a late review came in, during which the authors could no longer post official rebuttal. I tried to pass messages between the authors and the last reviewer in the hope of initiating some discussion but no further interaction happened. I myself happen to be reasonably familiar with some of the literature (as part of the missing comparison) brought up by the last reviewer and I think the authors' response addresses the concern adequately. One thing that is worth mentioning is that, due to the fact that this paper touches multiple related but still somewhat distant fields (causality, RL, missing-data theory), most of the reviewers admit that they are not proficient with all the topics covered. Therefore, the confidence is slightly lower than normal. Overall I am still in favoring of accepting this paper, but I am also OK bumping it down to weak accept.